# Unraveling phenological and stomatal responses to flash drought and implications for water and carbon budgets

Nicholas K. Corak[1,2], Jason A. Otkin[3], Trent W. Ford[4], and Lauren E. Lowman[1,2]

[1]Department of Physics, Wake Forest University, Winston-Salem, NC, USA
[2]Department of Engineering, Wake Forest University, Winston-Salem, NC, USA
[3]Cooperative Institute for Meteorological Satellite Studies, Space Science and Engineering Center, University of Wisconsin-Madison, Madison, WI, USA
[4]Illinois State Water Survey, Prairie Research Institute, University of Illinois at Urbana-Champaign, Urbana-Champaign, IL, USA

**Correspondence:** Lauren E. L. Lowman (lowmanle@wfu.edu)

**Abstract.** In recent years, extreme drought in the United States increased in frequency and severity underlining a need to improve our understanding of vegetation resilience and adaptation. Flash droughts are extreme events marked by rapid dry down of soils due to lack of precipitation, high temperatures, and dry air. These events are also associated with reduced preparation, response, and management time windows before and during drought which exacerbate their detrimental impacts on people and food systems. Improvements in actionable information for flash drought management are informed by atmospheric and land surface processes, including responses and feedbacks from vegetation. Phenologic state, or growth stage, is an important metric for modeling how vegetation modulates land-atmosphere interactions. Reduced stomatal conductance during drought leads to cascading effects on carbon and water fluxes. We investigate how uncertainty in vegetation phenology and stomatal regulation propagates through vegetation responses during drought and non-drought periods by coupling a land-surface hydrology model to a predictive phenology model. We also assess the role of vegetation in the partitioning of carbon, water, and energy fluxes during flash drought and compare against drought and non-drought periods. We selected study sites in Kansas, USA that were impacted by the flash drought of 2012, and where AmeriFlux eddy covariance towers provide ground observations to validate and compare against model estimates. Results show the compounding effects of reduced precipitation and high vapor pressure deficit (VPD) on vegetation distinguish flash drought from other drought and non-drought periods. High VPD during flash drought shuts down modeled stomatal conductance resulting in rates of evapotranspiration (ET), gross primary productivity (GPP), and water-use efficiency (WUE) falling below average drought conditions. Decreases in uncertainty from ensemble estimates of GPP and ET during the flash drought period reduce to winter levels implying variability in plant life stage and functionality during drought periods are similar to those of dormant months. These results have implications for improving predictions of drought impacts on vegetation.

# 1 Introduction

Frequency and severity of extreme droughts are predicted to increase within the next century (Dai, 2013). Flash droughts are a particular type of extreme drought characterized by their rapid intensification (Svoboda et al., 2002; Ford and Labosier, 2017; Otkin et al., 2018, 2022). The flash drought of 2012 that impacted the Central United States amplified the need to understand and predict flash droughts because of its estimated $30 billion of impacts to agriculture (Otkin et al., 2018). Work over the last decade has improved methods for identifying flash droughts based on development time and concurrent meteorological conditions (see Lisonbee et al., 2021, for a summary of flash drought definitions and indicators). Many studies have examined the drivers (e.g., lack of precipitation, greater atmospheric demand for water, above average temperatures) and impacts (e.g., soil moisture deficits and damages to agriculture) of flash drought (e.g., Lowman et al., 2023; Christian et al., 2023, 2022; Jin et al., 2019; Otkin et al., 2018) while others have examined vegetation-atmosphere interactions (Hosseini et al., 2022; Chen et al., 2021; Zhang and Yuan, 2020; Gerken et al., 2018; Otkin et al., 2016) and stomatal functioning (Novick et al., 2016; Roman et al., 2015). This study addresses the need to bring together the physical mechanisms driving flash drought and the resulting vegetation responses that inform land-atmosphere interactions.

Further assessment of vegetation-atmosphere feedback mechanisms may help improve identification of flash drought onset (Qing et al., 2022). Gross primary productivity (GPP), or carbon assimilation by plants during photosynthesis, is one such vegetation-atmospheric interaction impacted by drought (Zeng et al., 2023). Large reductions in GPP due to soil moisture and temperature anomalies can be used to mark the beginning and duration of flash drought events (Poonia et al., 2022; Zhang and Yuan, 2020), as seen in the 2012 flash drought (Jin et al., 2019). Flash droughts can intensify through land-atmosphere feedbacks (Basara et al., 2019); for example, vegetation expediting water stress by pulling water from deeper soil layers and further drying soils (Qing et al., 2022). Otkin et al. (2016) studied the evolution of soil moisture and vegetation conditions during the 2012 event, finding that changes in soil moisture and evaporative stress indicators preceded rapid drought intensification in the US Drought Monitor (USDM, Svoboda et al. (2002)). Chen et al. (2019) found declines in evapotranspiration (ET), another interaction between the vegetation and the atmosphere, to be a major sign of flash drought intensification.

Interactions between vegetation and the atmosphere are altered during flash drought events, thus it is necessary to consider vegetation state when studying the effects of flash drought (Chen et al., 2021). Additionally, capturing differences across plant types is essential for modeling vegetation response to drought. Failure to account for differential responses across plant functional types (PFTs) could result in underestimating the plant's ability to maintain its function under water stress (Zhou et al., 2013). Roman et al. (2015) showed that tree species in a forested region behaved differently during drought, with some species exhibiting isohydric tendencies, whereas others were more anisohydric. Isohydric plants are more conservative with their water-use strategies when under stress and tend to regulate their stomatal conductance making them less susceptible to hydraulic failure (Konings and Gentine, 2017). These tendencies dictate how much photosynthesis occurs and thus how much carbon is exchanged (Roman et al., 2015). However, Garcia-Forner et al. (2017) cautions against making links between carbon assimilation and water potential regulation by showing similar rates of carbon assimilation under controlled drought simulations between two species of Mediterranean trees with opposing drought responses (one isohydric and one anisohydric). For

some species, stomatal regulation exists on a spectrum and can shift between isohydric and anisohydric in response to atmo-spheric and water conditions (Wu et al., 2021; Guo et al., 2020) leading to variation and uncertainties in water-use strategies (Kannenberg et al., 2022). Ecosystem scale modeling may be able to incorporate the plant level spatial and temporal variability in water-use strategies (Giardina et al., 2023; Konings and Gentine, 2017) by taking into account concurrent meteorological and environmental conditions that influence plant water-use tendencies beyond the specie's physiological characteristics (Hochberg et al., 2018).

Vegetation type and growth stage can plan an important role in determining whether and how an area experiences changes in carbon uptake during flash drought. There is evidence connecting vegetation changes in response to flash drought to lower plant production (Zhang et al., 2020; Jin et al., 2019; He et al., 2018; Otkin et al., 2016; Hunt et al., 2014). Jin et al. (2019) and He et al. (2018) found that croplands, grasslands, and shrublands experienced the majority of loss to carbon uptake rates during the droughts of 2011 and 2012 across the central US and similar rates of ET were found in croplands in the US northern plain flash drought of 2017 (He et al., 2019; Kimball et al., 2019). Chen et al. (2021) showed increases in LAI led to increased ET and that in a low moisture regime the amount of latent heat released due to ET was sensitive to changes in LAI. Hunt et al. (2014) showed that maize experienced decreases in stomatal conductance, which led to declines in GPP and ET, during a flash drought. Roman et al. (2015) show that species specific stomatal control can lead to different drought responses implying that some plants that exhibit more drought tolerant behavior might be accessing deeper stores of water (Giardina et al., 2023).

Previous studies have used remotely sensed or ground measurements as indicators to study vegetation responses to flash drought (e.g., Christian et al., 2022; Zhang et al., 2020; Basara et al., 2019). In contrast, Chen et al. (2021) used an earth system model to gauge plant behavior during flash drought while Hosseini et al. (2022) used models with different phenological forcing to investigate impacts on the water and carbon cycles during drought. Remotely sensed and eddy covariance data provide snapshots of the state of the system at point-scale or gridded spatial resolutions, and fixed temporal resolutions, while models can scale in space and time. Inherently simplified due to the complexity of systems, numerical models incorporate physical and biological processes and statistical techniques to make predictions based on current states and their uncertainties (Dietze, 2017). Data assimilation procedures and Bayesian inference allow modelers to incorporate observations while also identifying sources of uncertainty in both processes and scale (Dietze, 2017; Dietze et al., 2013).

Accurately capturing plant phenology has implications for estimating photosynthetic activity (Lowman and Barros, 2018, 2016; Stöckli et al., 2008; Jolly et al., 2005), which will influence the water, carbon, and energy fluxes coupled between the land and atmosphere. We use two versions of the Duke Coupled Land-Surface Hydrology Model (DCHM) that incorporate routines for photosynthesis (Garcia-Quijano and Barros, 2005; Gebremichael and Barros, 2006) and predictive phenology, or plant life stage (Lowman and Barros, 2018, 2016) to more closely investigate if and how vegetation water-use strategies accelerate or decelerate dry down before and during flash drought. Data assimilation techniques allow us to capture model uncertainty around processes controlling vegetation activity, and in particular, assimilating vegetation phenology can improve the detection of drought (Mocko et al., 2021). We investigate whether plants exhibit anisohydric tendencies thereby exacerbating the dry down, or whether they regulate their water intake to preserve soil moisture to mitigate the effects of flash drought. In turn, we also investigate if plant behavior can be altered during periods of water stress by predicting phenology model parameters from

hydrologic model outputs in dry and wet periods. We hypothesize that simulated transpiration and carbon uptake rates will taper during flash drought due to limited soil water availability and increased atmospheric demand and that the phenological changes are directly related to changes in transpiration rates and GPP (Figure 1). Our specific hypotheses are:

**H1** During flash drought, there is an increase in days between precipitation events leading to larger reductions in total precipitation and infiltration as compared to non-flash drought events.

**H2** Lower total infiltration and higher atmospheric demand for water observed during flash drought reduces soil water available for root water uptake. This decreases stomatal conductance, subsequently leading to reduced rates of transpiration, carbon uptake, and water-use efficiency as compared to non-flash drought within a subseasonal time frame.

**H3** In response to decreased water availability during flash drought, vegetation phenological states will be diminished as compared to non-flash drought years exacerbating the reduction of transpiration and carbon uptake.

Here we use phenological responses of fraction of photosynthetically active radiation (FPAR) and leaf area index (LAI) to examine how flash droughts affect vegetation state and ultimately impact the surface fluxes governing the movement of water and carbon between the land and atmosphere. We use the well-studied flash drought of 2012 to compare vegetation growth state and water-use strategies during flash drought and non-drought periods to better understand how plants modulate water and interact with the atmosphere when under stress. Specifically, the model is used to explore how phenological state and stomatal regulation are altered by flash drought and subsequently affect vegetation productivity. We compare our model results with eddy covariance and remotely sensed values of vegetation state and atmospheric interactions. Discrepancies between observations and models with predictive versus forced phenology illuminate physical processes dictating plant water-use strategies (e.g., suppressing transpiration by closing stomata and limiting carbon intake). This study extends previous research on the water and carbon movement between plants and the atmosphere during flash drought by simulating the propagation of uncertainty after implementing a predictive phenology routine to understand how variability in the representation of vegetation state within a modeling framework impacts land-atmosphere exchanges during extreme drought events.

## 2 Methods and Data

### 2.1 Overview of Modeling Approach

Remotely sensed or ground observations of land and atmospheric responses to flash drought are useful in identifying changes in plant phenology, soil moisture, and evaporation rates, among others, but observations alone are unable to fully explain the mechanisms driving ecological responses and water-use strategies. Physically-based models can help fill the gaps in understanding what drives these changes by identifying key processes in the land-atmosphere interactions. For example, decreases in ground-based or satellite-derived GPP do not illuminate what processes caused the change, whereas a process based model might be able to signal that changes in root water uptake lead to decreased transpiration rates, which ultimately lead to decreased photosynthesis and carbon assimilation.

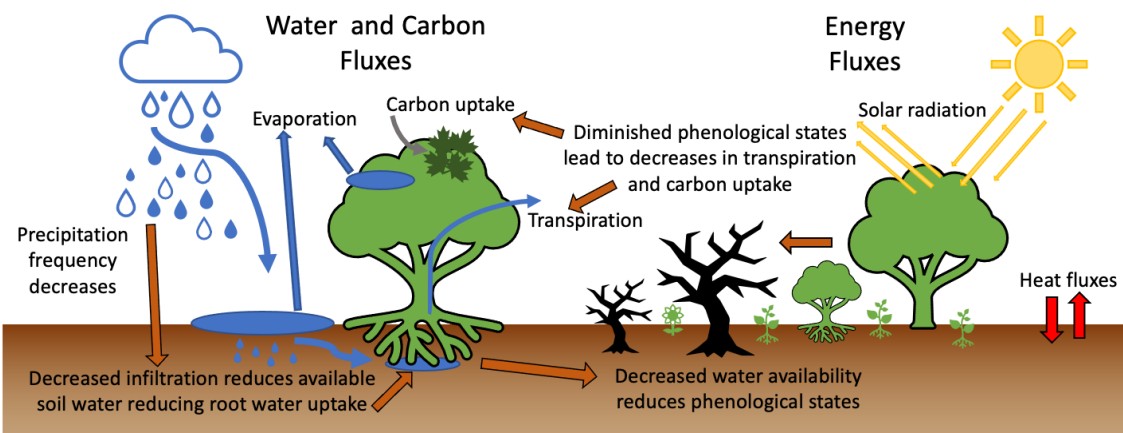

**Figure 1.** Schematic of water, carbon, and energy fluxes with hypotheses about ecological response to flash drought indicated with orange arrows. Decreased frequency of precipitation events leads to decreased infiltration and less water available for plant use during flash drought as compared to non-flash drought periods. During flash drought, the cascading effects of decreased water availability, exacerbated by the reduced phenological states and stomatal conductance, include rapid reductions in transpiration and atmospheric carbon uptake to levels below other drought periods.

120    Within physical models, changes in land-surface variables (e.g., soil moisture, root uptake, evaporation rates, etc.) are dependent upon meteorological conditions, either forced or dynamic (Sellers et al., 1997). Water-use strategies are dictated by vegetation phenological states (Hu et al., 2008) and stomatal regulation (Novick et al., 2016) and strongly influence GPP and ET (Beer et al., 2009). Therefore, physical, process-based models are able to adapt to changing meteorological conditions and capture mechanistic changes in vegetation-atmosphere interactions. Our goal is to identify vegetation responses that occur as a
125    result of flash drought and associate those changes with the physical processes represented in a land-surface hydrology model.

To identify physical mechanisms driving plant responses to flash drought intensification, we use two configurations of the physically based Duke Coupled surface-subsurface Hydrology Model (DCHM) with dynamic Vegetation (DCHM-V) and Predictive Vegetation (DCHM-PV). The DCHM-V provides baseline estimates soil moisture (SM), root uptake (RU), ET, and GPP using forced phenology from the from Moderate Resolution Imaging Spectroradiometer (MODIS) fraction photosynthetically
130    active radiation (FPAR) and LAI products. Instead of using forced phenology, the DCHM-PV uses a prognostic vegetation (i.e. phenological) model to predict the vegetation states of FPAR and LAI using parameters that correspond to seasonality (e.g., temperature and photoperiod), water availability (e.g., soil and vapor pressure deficit), and local vegetation characteristics (Lowman and Barros, 2018; Kim et al., 2015; Caldararu et al., 2014; Stöckli et al., 2008; Moradkhani et al., 2005). An ensemble Kalman filter (EnKF) data assimilation procedure following Lowman and Barros (2018) is used to estimate ensem-
135    bles of parameters for use in the predictive phenology model. Monte Carlo simulations of the DCHM-PV with the ensembles of predictive phenology parameters from the data assimilation step are used to explore the propagation of error and uncertainty.

We validate model simulations against ground observations, remotely sensing data, and other modeled products. A summary of the data sets used to force or validate both configurations of the DCHM is provided in Table 1.

**Table 1.** Summary of data products and uses

| Dataset | Variable(s) | Spatial Resolution | Temporal Resolution | Use | Reference |
|---|---|---|---|---|---|
| Stage-IV | Precipitation | 4 km | hourly | Forcing | Baldwin and Mitchell (1998) Du (2011) |
| NLDAS-2 Forcing File A | Atmospheric | 0.125° | hourly | Forcing/Data Assimilation | Mitchell et al. (2004) |
| NLDAS-2 Mosaic | Vegetation Fraction/ Albedo | 0.125° | hourly | Forcing/Data Assimilation | Xia et al. (2012) |
| MODIS MOD15A2H | LAI/FPAR | 500 m | 8 day | Forcing/Data Assimilation | Myneni et al. (2015) |
| MODIS MOD12Q1 | Land Cover | 500 m | yearly | Forcing | Friedl and Sulla-Menashe (2015) |
| STATSGO | Soil Texture and Porosity | 30 arcsec | fixed | forcing | Miller and White (1998) |
| AmeriFlux | GPP, latent heat, SM | point | 30 min. | Validation | Baldocchi et al. (2001) |
| MODIS MOD17A2H | GPP | 500 m | 8 day | Validation | Running et al. (2015) |
| NLDAS-2 | SM | 0.125° | hourly | Validation | Xia et al. (2012) |
| SMERGE | SM | 0.125° | hourly | Validation | Tobin et al. (2019) |

## 2.2 Forcing Data Sets for DCHM

### 140 2.2.1 Meteorological

The 1-D DCHM-V and -PV spatial and temporal resolution is set to the same scale as the highest quality precipitation forcing data available. For this study, the model uses the native resolution of the Stage-IV precipitation forcing from the National Oceanic and Atmospheric Administration (NOAA) National Centers for Environmental Prediction (NCEP) (Baldwin and Mitchell, 1998; Du, 2011). The Stage-IV dataset has 4 km spatial resolution and 1 h temporal resolution and with a record

beginning in 2002. All forcing data sets were interpolated to the Stage-IV resolution for the entire continental US (CONUS) before study site specific data were extracted. Atmospheric forcing data (downward short and long wave radiation, air temperature, specific humidity, surface pressure, wind velocity) used in the DCHM are from the North America Land Data Assimilation System Phase 2 (NLDAS-2) Forcing File A. NLDAS-2 is a combination of observational and reanalysis data sets (Mitchell et al., 2004) intended for use in land surface models like the DCHM. The data are available at 0.125 degree spatial resolution and 1 h temporal resolution. They are spatially interpolated to the 4 km Stage-IV grid. No temporal interpolation was necessary.

### 2.2.2 Land Cover

The land surface albedo and fraction of vegetation cover used in the DCHM-V and -PV come from the NLDAS-2 Mosaic Land Surface Model L4 dataset at 0.125 degree spatial resolution and 1 h temporal resolution (Xia et al., 2012; Mitchell et al., 2004). NASA's MODIS Land Cover (MCD12Q1) remotely sensed satellite land cover classification product is used to determine land cover type within the DCHM. In particular, we use the University of Maryland classification scheme (Sulla-Menashe and Friedl, 2018). Within the model, land cover type is updated yearly. The native spatial resolution of this data set is 500 m and it is interpolated to the 4 km resolution using a nearest neighbor approach.

### 2.2.3 Soil Texture and Porosity

Soil texture and porosity data was acquired from Soil Information for Environmental Modeling and Ecosystem Management CONUS-Soil (Miller and White, 1998). The CONUS-Soil spatial resolution is 1 km with 11 layers. We upscaled the raw soil texture and porosity data to the 4-km Stage-IV grid using two different methods. By averaging over the top 100 cm, we avoid averaging layers interpolated as bedrock, and thus near zero porosity. We approximate soil porosity by averaging the top eight layers (100 cm) and we represent texture using the texture mode across each grid cell and layer.

### 2.2.4 Vegetation

MODIS LAI and FPAR data were obtained for all of CONUS at the native 500-m spatial and 8-day temporal resolution. Before linearly interpolating the data to the Stage-IV grid and timestep, the data for each pixel were smoothed using a Savitsky-Golay filter (Savitzky and Golay, 1964) algorithm following Chen et al. (2004) in order to preserve seasonality and reduce noise in the data from cloud contamination and other atmospheric disturbances that may alter surface reflectance observations (Cihlar et al., 1997; Tanré et al., 1997). We use m=6 scaling window and d=4 degree for the interpolating polynomial as (Chen et al., 2004; Lowman and Barros, 2016).

### 2.3 Data Sets Used for Model Comparison

We assess vegetation responses to the Kansas flash drought of 2012 by comparing model results of land surface, sub-surface and atmospheric carbon and water fluxes (e.g., SM, GPP, ET) to multiple ground and remotely sensed observations. Modeled SM fluxes from the DCHM-V and -PV are compared to SoilMERGE (SMERGE), NLDAS-2 NOAH model output, and AmeriFlux

eddy covariance. SMERGE is a 0.125 degree root-zone (0-40 cm) SM product obtained from 'merging' NLDAS-2 outputs with European Space Agency Climate Change Initiative surface satellite data that can predict vegetation health anomalies (Tobin et al., 2019). Because SMERGE only provides root-zone SM, we only compare it to the DCHM middle layer SM output. We also validate SM estimates against NLDAS-2 estimates Noah land-surface model (LSM) for all three soil layers used in DCHM (Xia et al., 2012)). When AmeriFlux SM data is available, we compare with modeled soil moisture from the top layer

since most AmeriFlux SM sensors are in the top few centimeters of soil. DCHM-V and -PV estimates of GPP are compared to MODIS (MOD17A2H) GPP product and AmeriFlux eddy covariance outputs of GPP. We also compare DCHM estimates of ET to AmeriFlux eddy covariance flux tower estimates by dividing observed latent heat flux by the latent heat of vaporization of water ($\lambda_w = 2.5$ MJ kg$^{-1}$, Dingman, 2015)).

## 2.4  Description of Study Sites

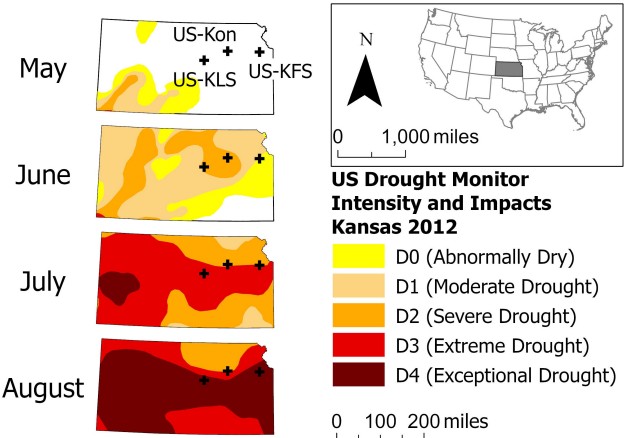

**Figure 2.** Evolution of the 2012 flash drought from May - Aug in the US Drought Monitor with the three AmeriFlux tower study sites (US-KFS, US-KLS, and US-Kon).

This study focuses on three AmeriFlux sites in Kansas (US-KFS, US-KLS, US-Kon, Figure 2 and Table 2), chosen because of the availability of GPP and latent heat (converted to ET) data during the flash drought year of 2012 and at least one wet year after 2012. When available, we used gap-filled FLUXNET FULLSET data for US-KFS and US-Kon (Pastorello et al., 2020). All three sites are classified as grasslands according to the International Geosphere-Biosphere Programme (IGBP) land cover and all three sites have Cfa (humid, subtropical) Köppen Climate Classifications (Brunsell, 2020a, 2021, 2020b). US-KFS is

located within a grassland-deciduous forest boundary area and receives 1014 mm of precipitation annually (Brunsell, 2020a). US-KLS is a perennial agricultural study sties receiving 812 mm of rainfall each year (Brunsell, 2021). US-Kon is part of the Konza Prairie Long-term Ecological Research (LTER), recieves 867 mm of precipitation, and is burned annually. Static characteristics of PFT, soil texture and porosity, and geographic information for the study sites are shown in Table 2. According

to the MODIS land cover classification product (MCD12Q1), each site had a unique vegetation cover type (savanna, grassland, cropland, Table 2). The PFT is a result of interpolating MODIS MCD12Q1 Land Cover Type 2 to the 4-km grid and does not align with the land cover from AmeriFlux in all cases. The soil texture and porosity are interpolated CONUS-Soil (Miller and White, 1998) values.

**Table 2.** AmeriFlux study sites contained within Stage-IV pixels.

| Site | Latitude | Longitude | PFT | Soil Texture | Soil Porosity | Mean Precipitation [mm yr$^{-1}$] | Reference |
|------|----------|-----------|-----|--------------|---------------|-----------------------------------|-----------|
| US-KFS | 39.0561 | -95.1907 | SAV | silty clay loam | 0.4225 | 1012 | Brunsell (2020a) |
| US-KLS | 38.7754 | -97.5684 | CRO | silt loam | 0.4812 | 812 | Brunsell (2021) |
| US-Kon | 39.0824 | -96.5603 | GRA | silty clay loam | 0.4588 | 867 | Brunsell (2020b) |

Plant functional type (PFT), soil texture, and soil porosity determined after interpolation to the Stage-IV grid. Abbreviations: SAV = Savanna, CRO = Cropland, GRA = grassland. Precipitation totals listed as AmeriFlux annual mean.

## 2.5 Description of Modeling Work

### 2.5.1 Land-Surface Hydrology Model

We employ two 1-D versions of the DCHM coupled land-surface hydrology model that accounts for water and energy exchanges between three soil layers, the surface, and the atmosphere (Lowman and Barros, 2018, 2016; Tao and Barros, 2014, 2013; Yildiz et al., 2009; Yildiz and Barros, 2007, 2005; Gebremichael and Barros, 2006; Garcia-Quijano and Barros, 2005; Devonec and Barros, 2002; Barros, 1995). A 4-km grid resolution and 1-hr timestep were chosen to run the model to match the native spatial resolution of the Stage-IV precipitation data, as precipitation is the main source of uncertainty when modeling drought (Trenberth et al., 2014). We use 80 mm for the top layer soil depth to ensure model stability, but middle and deep layers were selected to best match the USDA Kansas soil profile (Soil Survey Staff). The this yields three soil layers: top (0-80 mm), middle (80-890 mm) and bottom (890-1830 mm). Rooting depth and density, which are used to determine the total root water uptake in the DCHM, are calculated using empirical exponential root distribution functions that vary by PFT (Lowman and Barros, 2016; Zeng, 2001; Lai and Katul, 2000; Jackson et al., 1996; Clausnitzer and Hopmans, 1994). Soil layer and rooting depths align with the different combinations of soil textures and PFTs found in Thornthwaite and Mather (1957).

The DCHM water balance includes subroutines for evaporation from the different components of the land surface (i.e., bare soil, and vegetation), ponding and groundwater runoff, snow accumulation and melt, and root water uptake while energy balance routines solve for net radiation, and sensible, latent heat, and ground heat fluxes (Lowman and Barros, 2018, 2016; Tao and Barros, 2014, 2013; Yildiz and Barros, 2007, 2005; Garcia-Quijano and Barros, 2005; Devonec and Barros, 2002; Barros, 1995). The water and energy balances both influence photosynthesis, which is simulated using the Farquhar model (Lowman and Barros, 2016; Garcia-Quijano and Barros, 2005; Farquhar and Caemmerer, 1982; Farquhar et al., 1980).

### 2.5.2 Predictive Phenology

The key difference between the two versions of the DCHM used for this study is that within the DCHM-V vegetative phenology is forced using the MODIS MOD15A2H FPAR and LAI products, while the DCHM-PV predicts phenology for the next day based on the current day conditions. Establishing differences in the outputs from DCHM-V and -PV illuminates changes in plant growth strategies. MODIS is a passive sensor and uses only the red (648 nm) and near-infrared (NIR, 858 nm) spectral bands to estimate values of LAI (Myneni et al., 2015). Within the DCHM-PV, the dynamic canopy biophysical properties (DCBP) model predicts plant life stage based on climatological properties of water availability, air temperature, and evaporative demand (Lowman and Barros, 2018). FPAR and LAI are dynamically estimated instead of forced using MODIS observations to evaluate impacts on estimates of ET and GPP (Lowman and Barros, 2018; Kim et al., 2015; Caldararu et al., 2014).

The DCBP is the predictive phenology model that determines future plant growth based on differences between current and potential phenological states. The growing season index (GSI) determines potential phenological state based on current climate conditions (Jolly et al., 2005; Stöckli et al., 2008). Specifically it is a function of temperature, photoperiod, soil water potential, and VPD (Lowman et al., 2023; Lowman and Barros, 2018). Lowman and Barros (2018) adapted the framework to incorporate soil water parameters that affect predictions of plant growth stage. The DCBP is implemented within the DCHM-PV to estimate phenologic state with the land-surface hydrology model. However, in order to implement the predictive phenology model within the DCHM-PV, we first must estimate parameters that determine plant growth rates and sensitivity to meteorological and soil conditions.

A Bayesian hierarchical approach is used to estimate the parameters for the DCBP. Specifically, a dual state-parameter ensemble Kalman filter (EnKF) is used to jointly estimate the phenologic states of FPAR and LAI and the eleven other parameters within the DCBP (Table 4 Lowman et al., 2023; Lowman and Barros, 2018). This method was described by Moradkhani et al. (2005) as a way of simultaneously predicting states and parameters in hydrologic models, and later implemented by Stöckli et al. (2008) to assimilate remotely sensed observations of LAI and FPAR into a predictive phenology model.

The parameter estimation procedure first consists of creating a prior distribution by sampling each state and parameter from a Gaussian distribution. This generates N=2000 ensemble members. Phenological states and input parameters are updated at every timestep for the duration of the data assimilation period using the EnKF. We assimilate MODIS LAI and FPAR every 8 days (the native MODIS temporal resolution) to reduce error and ensure that phenological state predictions do not stray too far from observations (Lowman et al., 2023; Lowman and Barros, 2018).

### 2.6 Model Simulations

We run both the DCHM-V and -PV from 2002-2019 at a 1 h timestep and 4 km spatial resolution, spinning-up 2002 three times to allow for model stabilization (Lowman and Barros, 2016, 2018). The DCHM-V simulations provide a baseline for changes in water, energy, and carbon exchange using forced phenology from MODIS while the DCHM-PV simulations implements a predictive phenology scheme that allows us to investigate how dynamic changes in plant growth strategy impact the aforementioned fluxes.

In order to run the DCHM-PV, we first generate phenology model parameters for the predictive phenology routine. Specifically, we use 2003 (DRY), 2005 (WET), and 2003-2005 (3YR), as the data assimilation periods in the DCBP to generate parameters that correspond to wet, dry, or average precipitation regimes (Table 3). We use three different assimilation periods in order to capture the sensitivity of phenology model parameters to the meteorological conditions. It has been shown under varied climatological conditions plants can be highly adaptable, transitioning from isohydric to anisohydric in a single season

(Guo et al., 2020). Lowman and Barros (2018) showed that assimilation period can determine the water stress adaptations for the modeled vegetation state. Broadly speaking vegetation model parameters predicted using data from years with minimal rainfall represent plants that are accustomed to drier conditions and therefore exhibit more regulation in their water-use tendencies (Lowman and Barros, 2018; Sade et al., 2012).

To incorporate uncertainty from the phenology parameter estimation step into the DCHM-PV simulation, we run the model

as Monte Carlo simulations with N=2000 members. Each ensemble member is sampled from a Gaussian distribution using the final mean and standard deviation of the parameter estimates from each of the assimilations period. In our results, we focus on analyzing model output from 2006-2019 to omit from our analysis the 2003-2005 period used in the data assimilation step.

**Table 3.** Summary of precipitation conditions during assimilation periods.

| Year(s) | Assimilation Period* | Stage-IV Annual Precipitation Accumulation [mm] | | |
| --- | --- | --- | --- | --- |
| | | US-KFS | US-KLS | US-Kon |
| 2003-2005 | 3YR | 1066 | 770 | 847 |
| 2003 | DRY | 804 | 756 | 670 |
| 2005 | WET | 1242 | 806 | 956 |

*The data assimilation periods: 3YR represents a period with average annual precipitation, WET and DRY are periods with above and below average annual precipitation, respectively.

## 2.7   Analysis of Model Outputs

In this manuscript, we are interested in exploring whether land-surface, subsurface, and atmospheric interactions are distinct

in flash drought compared to drought and non-drought periods. We focus on results from the three AmeriFlux sites for 2012 (flash drought), 2018 (drought), and 2019 (non-drought) to draw conclusions about plant response during flash drought and how they differ from drought and non-drought years. We also evaluate model outputs from 2006-2019 to assess the differences between the DCHM-V and DCHM-PV model configurations during drought and non-drought years compared to a flash drought year. During this time period, we identified drought years as 2006, 2011, 2013, 2014, 2018 and non-drought years as 2007-

2010, 2015-2017, 2019 using the USDM for the Central and East Central Kansas climate regions (Svoboda et al., 2002). Drought years were determined by whether parts of the region reached the D2 "Severe Drought" classification or higher. When computing drought and non-drought averages, we use the years listed here. In many time series results, we display the water year (April-October) rather than the entire year because plants are largely dormant outside of the water year in a temperate

region (Dai et al., 2016; Wang et al., 2003; Towne and Owensby, 1984). Transpiration is calculated from total root water uptake
through the three soil layers and total evaporation is computed from summing evaporation from ground and canopy surfaces
allowing us to partition ET into evaporation and transpiration (Lowman and Barros, 2018; Lai and Katul, 2000). Water-use
efficiency is represented as the ratio of GPP and ET (WUE = GPP/ET, Beer et al. (2009)). We highlight differences between
the DCHM-V and DCHM-PV model simulations and compare outputs to remotely sensed and in situ observations where
available.

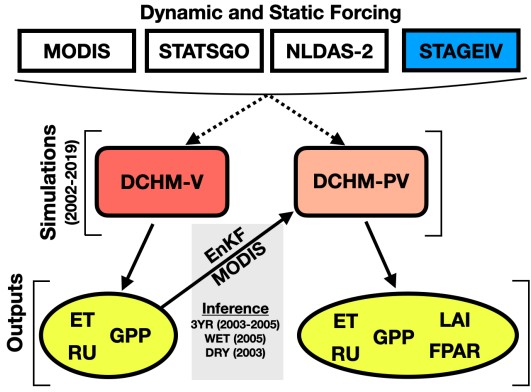

**Figure 3.** Schematic of modeling workflow. Spatial and temporal resolutions of all forcing data are interpolated to match the resolution of the
Stage IV precipitation (4 km and 1 h) which is the resolution used for the DCHM in this study. Land cover, soil properties, and atmospheric
forcing inputs come from MODIS, STATSGO, and NLDAS-2, respectively. Simulations are run from 2002-2019. Three ensembles of pa-
rameters for the predictive phenology routine in the DCHM-PV are generated using an ensemble Kalman filter (EnKF) with simulated soil
water potential and vapor pressure deficit from the DCHM-V, MODIS MOD15A2H FPAR/LAI, and concurrent meteorological conditions
from 2003 (DRY), 2005 (WET), and 2003-2005 (3YR). DCHM-V outputs of interest include evapotranspiration (ET), Root water uptake
(RU), and gross primary productivity (GPP). Additional DCHM-PV outputs include predicted fraction of photosynthetically active radiation
(FPAR) and leaf area index (LAI).

## 3 Results

### 3.1 Phenology

#### 3.1.1 Growth Rate Parameter

The growth rate parameter, $\gamma$, dictates how much phenological state (i.e. FPAR and LAI) can change in a given time step
(Lowman and Barros, 2018; Stöckli et al., 2008). The uncertainty in $\gamma$ shows the variability in vegetation responses to changing
phenological states. Lower uncertainty in $\gamma$ establishes the 3YR assimilation period, with a mixture of wet and dry years, as
the preferred choice for running the DCHM-PV (Figure 4). This finding is in agreement with Lowman and Barros (2018)

who found that using assimilation periods with both wet and dry conditions has the effect of capturing adaptive plant water-use strategies. This lower uncertainty propagates through the DCBP in the DCHM-PV, leading to lower uncertainty in the predictions of FPAR and LAI (Figures 5 and 6). The values of $\gamma$ vary by site due to a combination of local climate and vegetation type. US-KFS, modeled as a savanna, has the lowest mean and standard deviation of $\gamma$ (Table 4). The smaller magnitudes of the growth parameters indicates that vegetation is less likely to make abrupt changes and exhibit more resilience when faced with extreme dry down. Other parameter estimation outputs used to generate ensembles from the 3YR assimilation period can be found in Table 4.

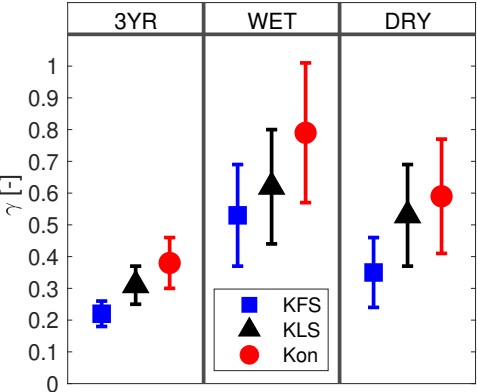

**Figure 4.** Ensemble means and one standard deviation of the growth rate parameter, $\gamma$, for each site and all three data assimilation periods: 3YR (2003-2005), WET (2005), DRY (2003).

### 3.1.2 Fraction of Photosynthetically Active Radiation

Overall, DCHM-PV simulated FPAR tends to follow the same patterns as MODIS throughout the growing season, irrespective of choice of parameters. Results indicate slower senescence and reduced variance using the 3YR assimilation parameters as compared to the WET and DRY parameters during late June and early July 2012 across all three sites (Figure 5 a,d,g). This aligns with the known period of flash drought that occurred across Kansas (Lisonbee et al., 2021). The predicted values of FPAR at US-KFS and US-KLS are slightly higher than the MODIS values during the 2012 growing season. The predicted values of FPAR match well against MODIS for the US-Kon site, especially during the decline in late June through July. During the flash drought period, there is a notable decrease in variance, or uncertainty, across the Monte Carlo simulations.

For US-KFS across the three simulations, the simulation using the WET parameters achieves a higher FPAR during the flash drought and holds its peak throughout the month of May, with declines beginning in June and bottoming in early July before rising again in the latter part of the growing season. FPAR decreases from 0.77 to 0.41 for the WET parameters while reductions from the time of peak FPAR to early July in the simulations using DRY and 3YR parameters are from 0.73 to 0.47 and 0.76 to 0.53, respectively. The decreases in FPAR observed from mid-May to mid-July in 2012 are more pronounced than during the growing season of the drought year 2018 when fluctuations in FPAR were smaller. Results from an above

**Table 4.** Ensemble mean and one standard deviation of predictive phenology model parameters from the 3YR assimilation period

| Parameter | Description | Units | Mean parameter estimates ± one standard deviation | | |
| --- | --- | --- | --- | --- | --- |
| | | | US-KFS | US-KLS | US-Kon |
| $T_{min_{min}}$ | Minimum value of daily minimum temperature | °C | $-5.5 \pm 3.1$ | $0.1 \pm 2.4$ | $-2.3 \pm 3.2$ |
| $T_{min_{max}}$ | Maximum value of daily minimum temperature | °C | $14.0 \pm 1.8$ | $16.5 \pm 1.8$ | $15.8 \pm 2.0$ |
| $Pht_{min}$ | Minimum daily exposure to sunlight | h | $10.0 \pm 0.4$ | $9.8 \pm 0.6$ | $10.7 \pm 0.6$ |
| $Pht_{max}$ | Maximum daily exposure to sunlight | h | $14.3 \pm 0.3$ | $14.2 \pm 0.4$ | $14.3 \pm 0.4$ |
| $VPD_{avg_{min}}$ | Minimum daily average vapor pressure deficit | mb | $17.1 \pm 1.3$ | $16.6 \pm 1.4$ | $16.9 \pm 1.4$ |
| $VPD_{avg_{max}}$ | Maximum daily average vapor pressure deficit | mb | $58.7 \pm 2.3$ | $55.8 \pm 2.2$ | $55.6 \pm 2.3$ |
| $\psi_{soil,avg_{min}}$ | Minimum daily average soil water potential | J kg$^{-1}$ | $-42.1 \pm 5.6$ | $-37.2 \pm 5.8$ | $16.9 \pm 5.5$ |
| $\psi_{soil,avg_{max}}$ | Maximum daily average soil water potential | J kg$^{-1}$ | $-7.4 \pm 1.3$ | $-7.0 \pm 1.4$ | $-6.9 \pm 1.4$ |
| $FPAR_{min}$ | Minimum fraction of photosynthetically active radiation | - | $0.31 \pm 0.01$ | $0.35 \pm 0.01$ | $0.31 \pm 0.01$ |
| $LAI_{max}$ | Maximum leaf area index | m$^2$ m$^{-2}$ | $6.36 \pm 0.15$ | $6.51 \pm 0.17$ | $6.65 \pm 0.18$ |
| $\gamma$ | growth rate | day$^{-1}$ | $0.22 \pm 0.04$ | $0.31 \pm 0.06$ | $0.38 \pm 0.08$ |

For an in depth description of the predictive phenology routine within dynamic canopy biophysical properties (DCBP) model see Lowman et al. (2023) and Lowman and Barros (2018).

average precipitation year (2019) show a steady increase, a longer peak growing season, and a decrease in line with fall senescence across all simulations. However, using WET and DRY parameters at US-KLS lead to ~0.2 reduction in FPAR in July 2019, opposed to ~0.1 reduction from the 3YR parameters. The larger reduction is likely due to the below average July precipitation and the larger WET and DRY values of $\gamma$ leading to more rapid phenological changes. Similar to the 2012 results, 2019 simulations using phenology parameters from the 3YR assimilation period showed slower late-season declines in FPAR than simulations using parameters from the WET or DRY assimilation periods. This can be seen from the 3YR parameter simulations for US-KLS and US-Kon which show higher FPAR through July.

### 3.1.3 Leaf Area Index

Predicted values of LAI are similar to MODIS LAI with small relative differences (Figure 6). During the flash drought year of 2012, a steep decline in modeled LAI can be seen in late June and early July across the three sites. LAI declines almost 1 m$^2$ m$^{-2}$ in a few weeks during summer 2012 compared to steadier values during the drought of 2018. Growing season LAI was ~0.5 m$^2$ m$^{-2}$ lower in 2012 compared to 2018. DCHM-PV model outputs of LAI during 2019 match MODIS but are 1-2 m$^2$ m$^{-2}$ higher during June, July, and early August at US-KFS and US-KLS, and slightly lower than MODIS at US-Kon.

Simulated LAI values vary slightly across the three sites. For US-KFS, simulations using the WET year parameters achieve higher values in LAI than the other two simulations (Figure 6 a-c). For US-KLS, and US-Kon, the growing season LAI has

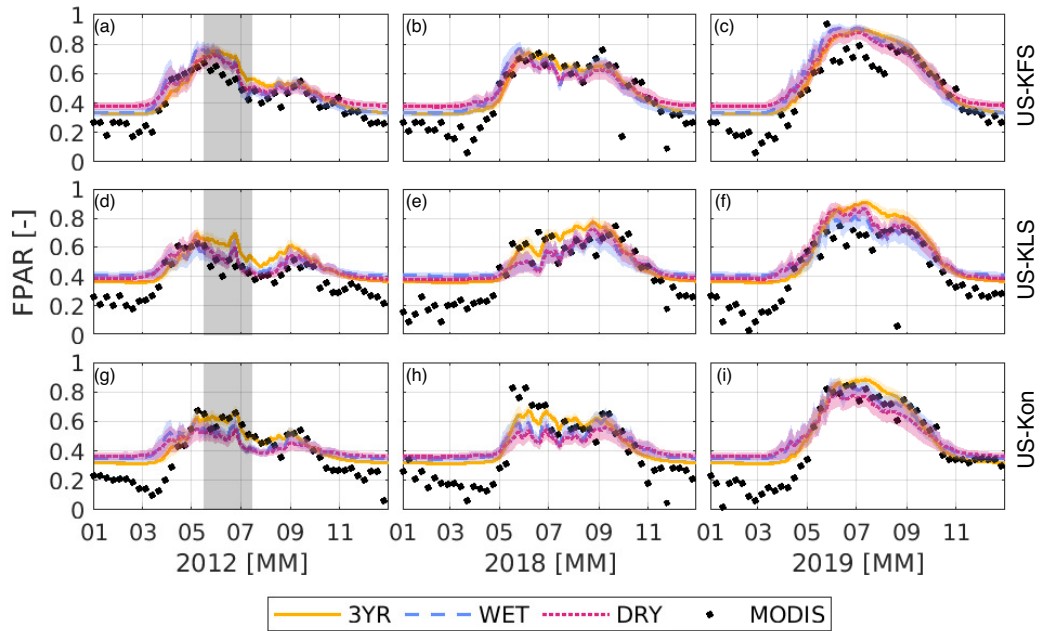

**Figure 5.** Fraction of photosynthetically active radiation (FPAR) predicted from the DCHM-PV for the flash drought year (2012), a drought year (2018), and a non-drought year (2019). Colors indicate the different data assimilation periods: 3YR (yellow), WET (blue), DRY (red). Corresponding shaded regions represent one standard deviation of model outputs from the 2000 ensemble members. The 8-day MODIS MOD15A2H LAI is shown as black dots. The gray shaded regions in the left most panels highlights the 2012 flash drought period.

the highest peaks in the simulations using the 3YR parameters (Figure 6 d-i). With more rainfall in May and June 2019, the simulations using the WET parameters result in lower LAI than the simulations using the DRY parameters.

The most consistent similarities across the phenology results is that the simulations using the 3YR parameters generally show a slower decline in LAI in flash and non-flash drought years for all sites. Additionally, the simulations using WET and DRY parameters are more similar to each other than to the simulations using 3YR parameters. This result is commensurate with the values of the means and variances of $\gamma$ resulting from the different assimilation periods. Simulations using the 3YR assimilation period result in LAI remaining high for a longer period of time with a decrease in response to flash drought

developing slower than the other two simulations. This is also apparent for US-KLS and US-Kon in the 2019 3YR simulations in which leaf growth continues through June and peaks in the middle of July, while in the WET and DRY simulations new growth tends to slow from the beginning of June through mid-July.

    Generally, the predictive phenology model compares favorably with the seasonal changes observed in MODIS FPAR and LAI (Figures 5 and 6) in both flash drought and non-flash drought periods. In the summer, at US-KFS and US-KLS during

2019, the model tends to predict FPAR and LAI values higher than MODIS. In 2019, at US-KFS, MODIS observed a steady decline in FPAR from 0.8 to 0.6 throughout July followed by an increase to 0.8 over an 8-day period at the beginning of August (Figure 5c). The DCHM-PV results do not show the same decline. Similarly, MODIS observes a drop in LAI (Figure 6c) before

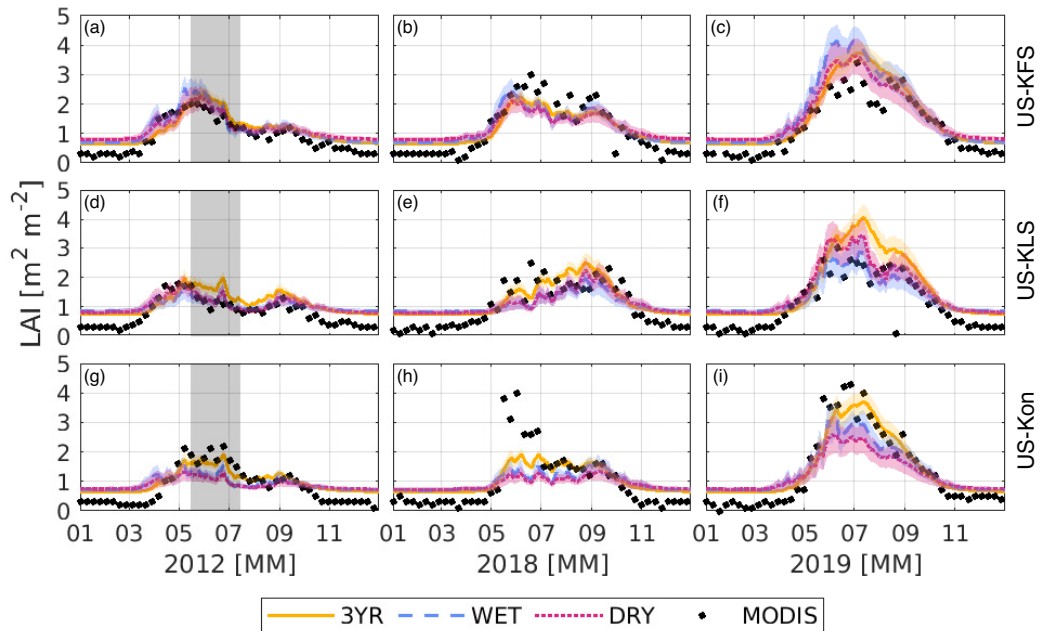

**Figure 6.** Leaf area index (LAI) predicted from DCHM-PV for the flash drought year (2012), a drought year (2018), and a non-drought year (2019). Colors indicate the different data assimilation periods (yellow - 3YR (2003-2005), blue - WET (2005), red - DRY (2003)), with corresponding shaded regions representing one standard deviation of model outputs from the 2000 ensemble simulations. The 8-day MODIS MOD15A2H LAI is shown in black markers. The gray shaded regions in the left most panels highlights the 2012 flash drought period.

an abrupt increase while model estimates remain higher than MODIS. Yet, in June 2019 at US-Kon, the DCHM-PV estimates are lower than MODIS LAI.

The bulk of the following results and analysis compares vegetation responses during flash drought and non-flash drought periods rather than an inter model comparison across the different assimilation strategies. Estimates from the WET and DRY simulations tend to be in agreement with results from the 3YR simulations. From this point forward, we only show results from the 3YR simulations.

## 3.2   Sub-surface Water

### 3.2.1   Infiltration

During non-drought years, monthly infiltration accumulations are above or near 100 mm per month, on average, from April to July with the highest amounts in May (Figure 7). During drought years, infiltration between April-July is less than non-drought years. Furthermore, monthly accumulated infiltration is lower during the flash drought year compared to both drought and non-drought years, suggesting there is less water available for plant use during the growing season. At US-KFS from April-

October of 2012, monthly infiltration is slightly below that observed during drought years. A large decline in May infiltration

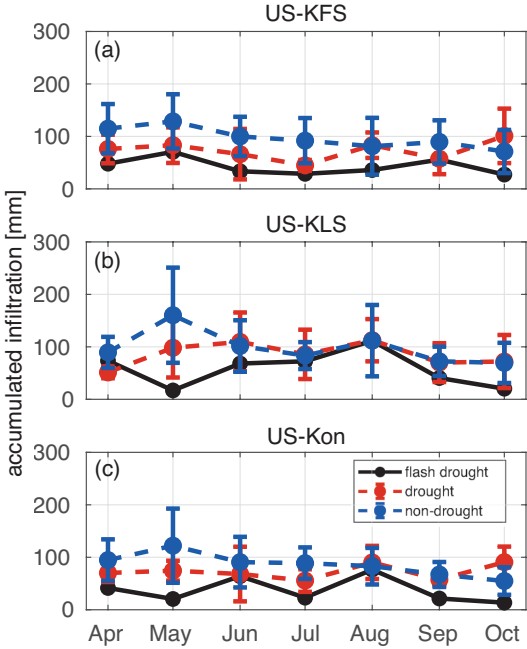

**Figure 7.** DCHM-PV 3YR ensemble means of monthly infiltration accumulations for drought (red dashed line) and non-drought (blue dashed line) years compared to 2012 (black solid line) for all three study sites. Monthly sums are computed from the ensemble means of the 2000 Monte Carlo simulations then averaged across drought or non-drought years. Error bars represent one standard deviation across drought and non-drought years.

at US-KLS and US-Kon led to infiltration accumulations that are 1-2 standard deviations below average drought conditions. All sites had infiltration rates below 100 mm for all months during 2012 with the exception of US-KLS in August 2012.

Low monthly infiltration amounts during the flash drought year are likely due to lower precipitation accumulations (Figure S4) coupled with an increase in the number of days between precipitation events (Figure 8) and an increase in atmospheric demand for water (Figure S5). During drought and non-drought years, the average number of days between rainfall events within a month ranges from 1 to 7 days, while the lower end for the flash drought year is higher at 2.5 days. Here, we consider a rainfall event to be any day with recorded precipitation. Additionally, during drought and non-drought years, monthly infiltration exceeds 150 mm, but in 2012 remains at or below 75 mm for all sites aside from August 2012 at US-KLS where monthly infiltration is ∼110 mm. In 2012, all three sites averaged over four days between rainfall events during May, June, and July with US-KFS averaging over six days between rainfall events during both May and June and more than five days in July (Figure 8a). Across all three sites from April-October 2012, there were more than four days between precipitation events 80% percent of the time compared to just 20% of the time in non-flash drought years.

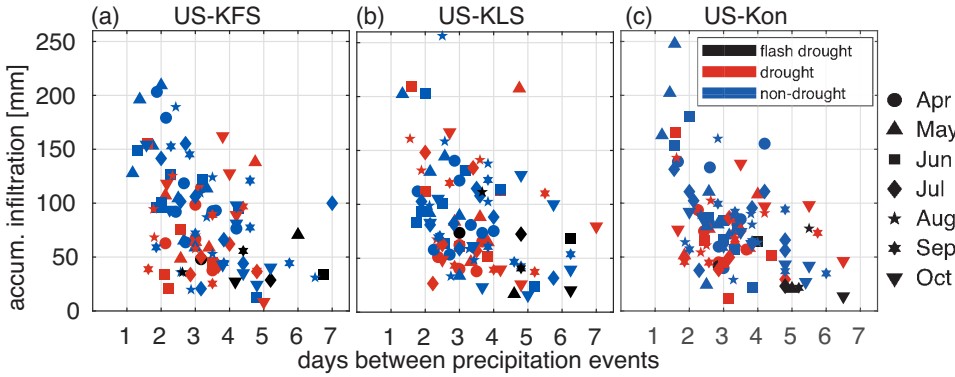

**Figure 8.** Monthly infiltration accumulation vs average days between precipitation events within a single month for (a) US-KFS, (b) US-KLS, and (c) US-KON. Each shape indicates one month over which the averaging occurred and colors distinguish flash drought (black) from drought (red) and non-drought (blue) years.

### 3.2.2 Soil Moisture

Soil moisture analysis and comparison to other soil moisture products is similar for all three study sites. Figures for soil
moisture at US-KFS for all three soil layers are available in supplemental material. Top layer soil moisture reaches the wilting point several times throughout the flash drought period of 2012 (Figure S1a). During peak flash drought, at the end of June and beginning of July, moisture content remains at wilting point for many days. Daily soil moisture agrees with AmeriFlux soil moisture observations in the top layer during 2012 at US-KFS. Discrepancies exist in 2018 when AmeriFlux observations fall to levels just above 0 $m^3$ $m^{-3}$.

Fluctuations in soil moisture match favorably with NLDAS-2 estimates across the top two layers in 2012, 2018, and 2019. However, middle layer soil moisture from DCHM estimates is about 0.05 $m^3$ $m^{-3}$ higher than NLDAS-2 and SMERGE by the late growing season of the flash drought year (Figure S2). DCHM estimates remain fairly steady in the deep layer during 2012, while NLDAS-2 soil moisture estimates continue to fall throughout the rest of the growing season (Figure S3). The steady DCHM soil moisture levels during flash drought may be indicative of the modeling stunting root water uptake during the same
time, preserving soil water content.

### 3.2.3 Root Water Uptake

Root water uptake is above non-flash drought levels in 2012 before the onset of flash drought in June. Then it remains lower than non-flash drought levels for the remainder of the growing season (Figure S6). The middle soil layer is responsible for up to four times more root water uptake than the other layers. Thus, a major decline in root water uptake through the middle
layer is informative of how plant water-use is altered during drought. While root water uptake starts out in 2012 at levels above average non-drought years, it falls to more than one standard deviation below drought averages by July. This drastic shift is

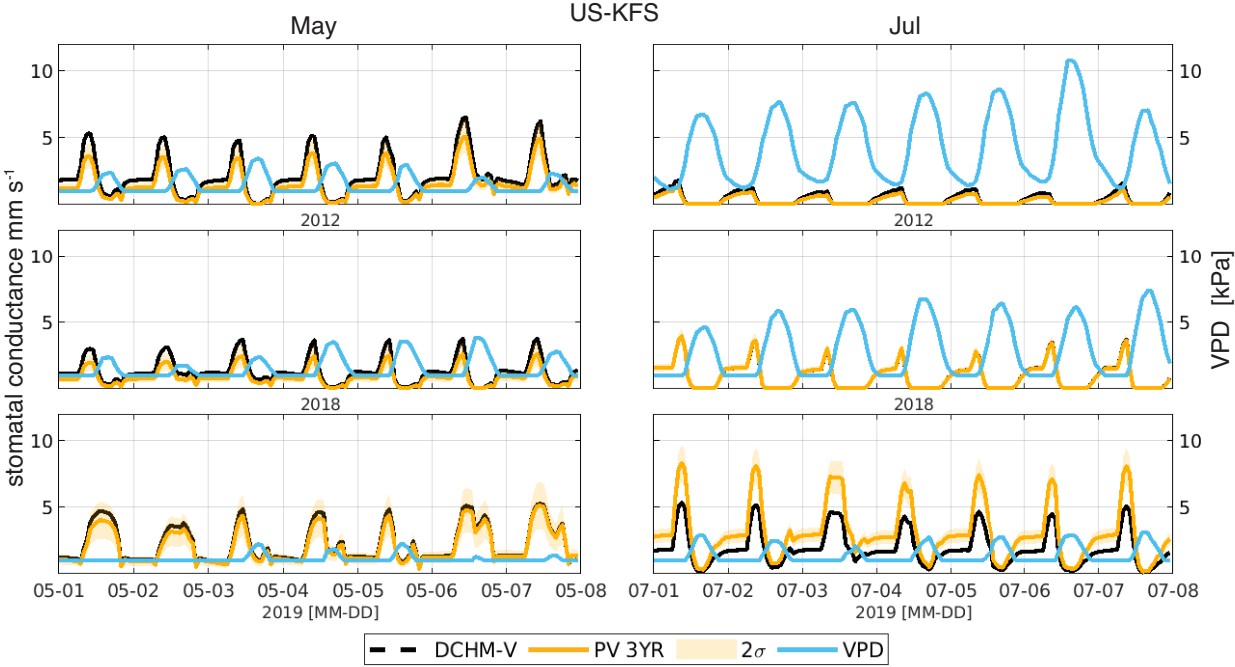

**Figure 9.** Stomatal conductance [mm s$^{-1}$] and vapor pressure deficit (VPD, kPa) for one week in May and July of 2012, 2018, and 2019 for US-KFS.

likely due to lower infiltration (Figure 7) and drives down rates of transpiration within the DCHM-V and -PV over the same period.

## 3.3 Plant-Atmosphere Interactions

### 3.3.1 Sub-daily Stomatal Conductance

Sub-daily estimates of stomatal conductance highlight how VPD can drive stomatal activity within the DCHM. In 2012, stomatal conductance in the first week of May was as high or higher than in 2019, a non-drought year at US-KFS (Figure 9). But by July, major differences in 2012 and 2019 stomatal conductance coincide with changes to VPD. In July 2012, high VPD shuts down midday stomatal conductance whereas lower values of VPD allow for higher rates of stomatal conductance during the same time in 2019. The large reduction in stomatal conductance from the first week of May to the first week of July during the flash drought year of 2012 is unlike that seen in a drought year like 2018 where stomatal conductance rates are similar in May and July.

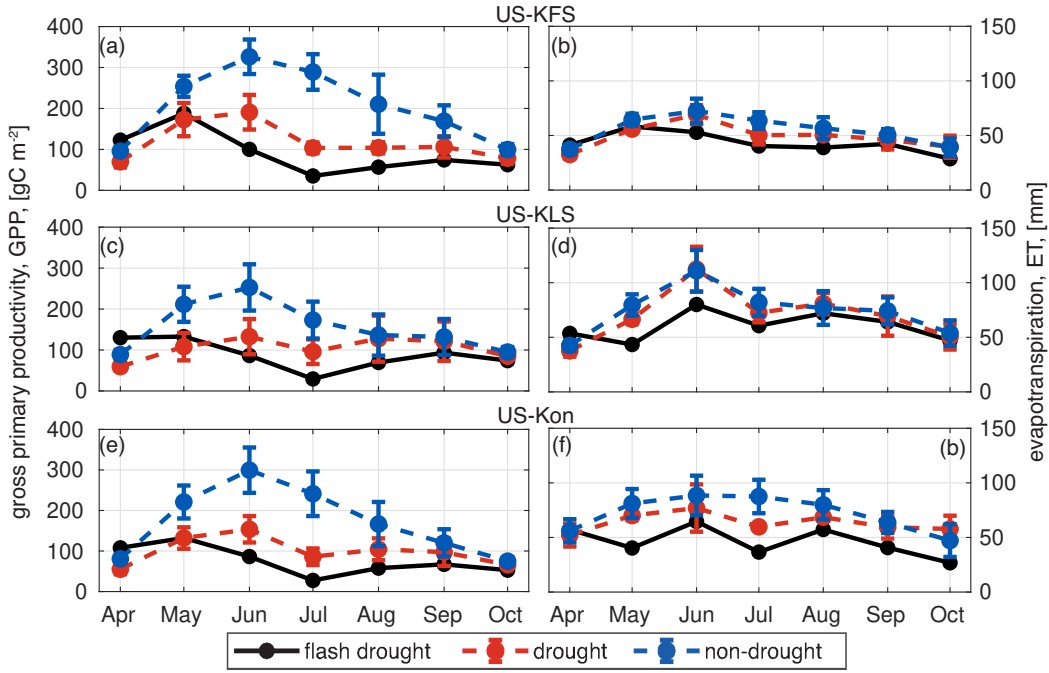

**Figure 10.** DCHM-PV 3YR monthly totals of GPP (a,c,e) and ET (b,d,f) for drought (red) and non-drought (blue) years compared to flash drought (black) for US-KFS, US-KLS, and US-Kon AmeriFlux sites. Monthly totals are computed from the ensemble means of the 2000 Monte Carlo simulations then averaged across drought or non-drought years. Error bars represent one standard deviation across drought and non-drought years, respectively.

### 3.3.2 Gross Primary Productivity

Monthly averages of GPP accumulations from DCHM-PV ensemble means throughout the water year (April - October) indicate that carbon uptake falls below drought averages from May to June during the flash drought year of 2012 (Figure 10 a,c,e). Flash drought carbon assimilation amounts remain below drought levels before converging to average drought/non-drought levels by the end of October. GPP amounts are up to 50% lower in drought years compared to non-drought years. During the flash drought, GPP monthly totals in June through August 2012 are at least one standard deviation lower than drought years averaged over the 2006-2019 simulation period. June 2012 GPP accumulations are half that of drought years and less than 30% of non-drought years. An even greater discrepancy is apparent in July with carbon assimilation amounts less than 30% of drought levels and 15% of non-drought levels. Despite increased GPP from July to August in 2012, accumulations are still one standard deviation below drought levels.

Seasonal variations of GPP at US-KFS (Figure 11) for simulations from the DCHM-V and -PV (3YR) with observations from MODIS and AmeriFlux for the flash drought year (2012), a drought year (2018), and a non-drought year (2019) can also be explored at the daily scale. Daily GPP is lower in drought versus non-drought years between April and October. During the flash drought year, there is a decline in GPP from 10 gC m$^2$ d$^{-1}$ in early May, above what was observed in 2018 and 2019, to

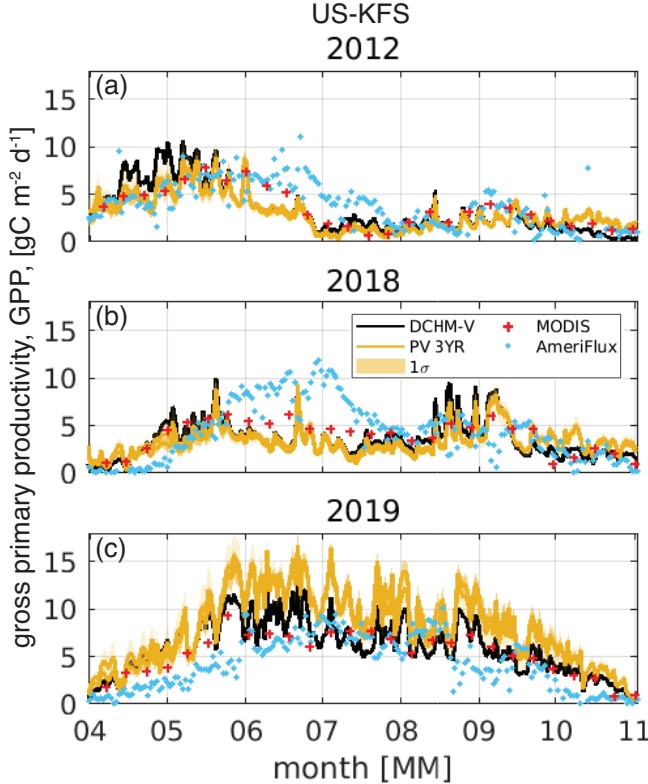

**Figure 11.** Daily gross primary productivity, GPP, at US-KFS for (a) 2012 flash drought, (b) 2018 drought and (c) 2019 a non-drought year. One standard deviation is shown as a shaded region for the DCHM-PV simulations. MODIS GPP are shown as red crosses and AmeriFlux GPP as blue dots.

near zero by July in 2012 (Figure 11,S15,S16). During the drought year (2018), daily GPP remains low throughout the growing season, but never decreases to below 1.2 gC m$^2$ d$^{-1}$ at US-KFS. From June to July in 2012, carbon uptake decreased from more than 5 to less than 1 gC m$^{-2}$ d$^{-1}$. This type of decline is not observed in a drought year (e.g., 2018). The rapid decline in GPP from May to July is what distinguishes the 2012 flash drought as a period of time where land-atmosphere interactions switch from resembling conditions of a wetter than an average wet year to a drier than an average dry year. The DCHM-PV GPP results are similar to MODIS GPP in most cases, except that it tends to underestimate GPP compared to MODIS in a drought year, which aligns with the higher MODIS estimates of FPAR and LAI during the same periods (Figure 6). Simulated GPP tends to underestimate flux tower GPP during June and July in 2012 and 2018, but overestimate in 2019.

### 3.3.3 Evapotranspiration

We consider monthly accumulations of ET for the flash drought year and averaged across non-flash drought years for the three study sites (Figure 10 b,d,f). ET accumulations are lower in the flash drought year starting in May, particularly at US-KLS

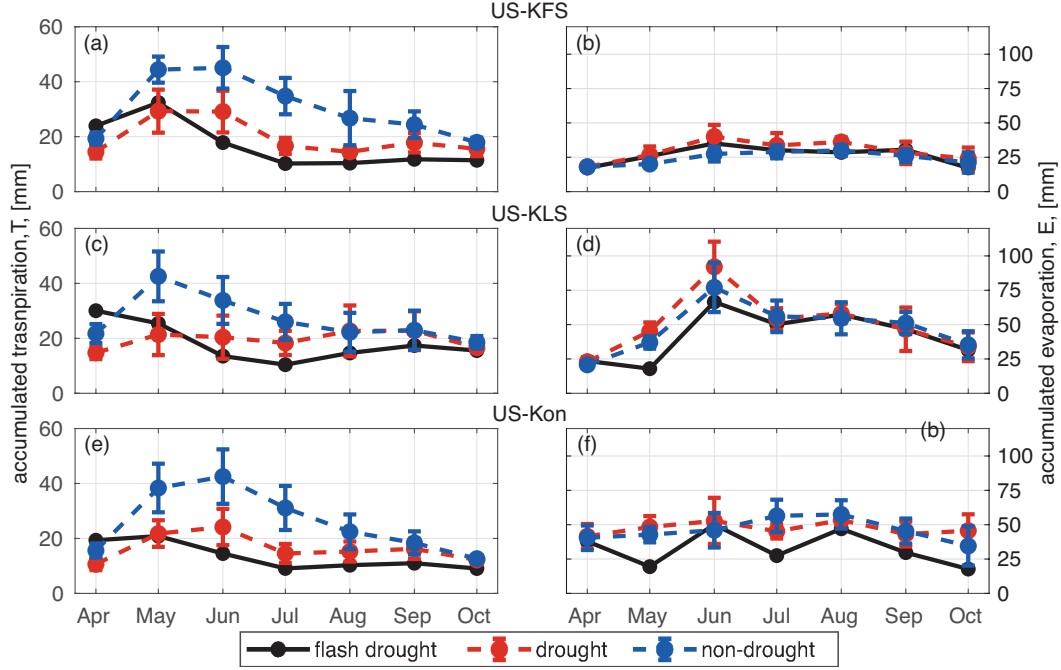

**Figure 12.** DCHM-PV 3YR monthly totals of transpiration, T, (a,c,e) and evaporation, E, (b,d,f) for drought (red) and non-drought (blue) years compared to flash drought (black) for US-KFS, US-KLS, and US-Kon AmeriFlux sites. Monthly totals are computed from the ensemble means of the 2000 Monte Carlo simulations then averaged across drought or non-drought years. Error bars represent one standard deviation across drought and non-drought years,respectively.

and US-Kon. Monthly ET during drought periods are slightly lower, but generally similar to non-drought at US-KFS and US-KLS, indicating that ET may not be a strong indicator of drought. However, parsing ET into its components of evaporation
and transpiration offers a different perspective. Simulated monthly transpiration accumulations follow trajectories similar to GPP during flash drought (Figure 12 a,c,e). Transpiration amounts during flash drought exceed non-drought years in April, match what is observed during drought years in May, and decline to levels below drought years through the rest of the growing season. Transpiration in July 2012 falls below one standard deviation of the drought years. At all sites, evaporation rates for drought and non-drought years are similar. At US-KFS, monthly evaporation is comparable to both drought and non-drought
years throughout the entire growing season (Figure 12b). At US-KLS, May and June evaporation totals are lower during the flash drought than drought and non-drought years. At US-Kon, May and July evaporation falls below drought and non-drought years.

During the flash drought, transpiration gradually declined from May to July (Figures 12, S19a). The fluctuations in total ET starting in June 2012 are the result of evaporation in response to small precipitation events. This suggests that following
precipitation events during flash drought onset, ET is dominated by evaporation. Reduced infiltration limits water available for root water uptake (Figures 7, S6). As transpiration is computed from root water uptake across the three soil layers, the

observation that transpiration decreases but maintains a small consistent rate through the flash drought indicates that vegetation is extracting water from deeper soil layers. ET never completely shuts down in 2012 because of the low rate of transpiration. However, evaporation completely halts during early July 2012, which is the peak of the flash drought period. Similar to flash drought, during drought in 2018, ET is dominated by evaporation (Figure S19b). But in the non-drought year 2019, transpiration makes up more than 50% of ET throughout the entire growing season except for short periods in July and August (Figure S19c).

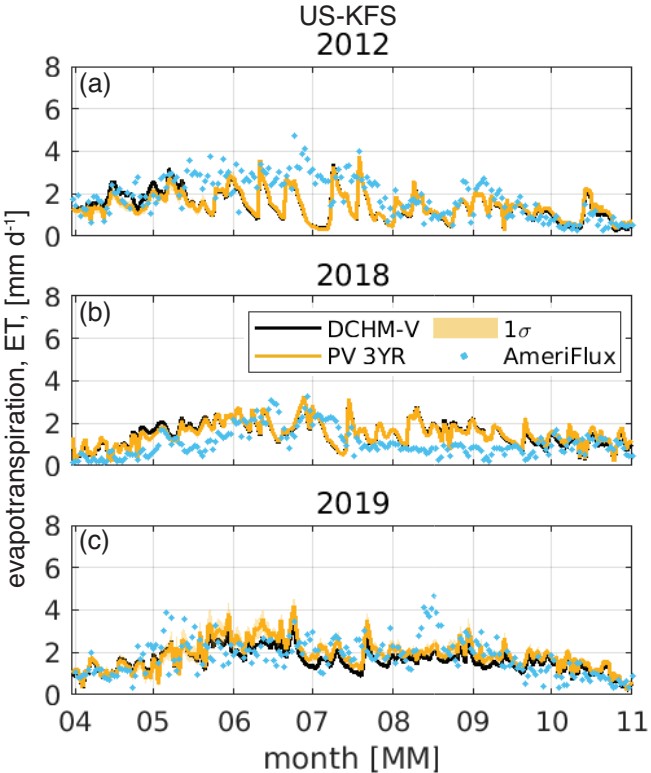

**Figure 13.** Daily evapotranspiration, ET, [mm d$^{-1}$], at US-KFS for (a) 2012 flash drought, (b) 2018 drought and (c) 2019 a non-drought year. Two standard deviations are shown for the DCHM-PV simulations. AmeriFlux ET is derived from latent heat measurements and shown as blue dots.

Daily ET estimated by the DCHM-PV matches well against AmeriFlux estimates at US-KFS during the flash-drought, and non-flash drought years (Figure 13). In 2012, DCHM-PV ET agrees with AmeriFlux through mid-May. From late May through July the model results tend to fall below AmeriFlux until August when they once again agree. In the drought (2018) and non-drought (2019) years, DCHM-PV ET appears to align with AmeriFlux throughout most of the season (Figure 13b,c). While model estimates of ET are higher than flux tower measurements in 2019 at US-KLS, they compare favorably in 2012 and 2018 (Figure S17). In contrast to model and flux tower comparisons at US-KFS and US-KLS, at US-Kon modeled ET (Figure S18) agrees with AmeriFlux in 2019 (non-drought), but underestimates during the summer months in 2012 (flash drought) and 2018 (drought). One explanation for the differences between model and tower ET data could be that water-use by vegetation during

flash drought is highly variable across sites, and the model is not able to represent all possible responses. Additionally, it is difficult for the DCHM and other Earth system models to account for plant access to deep water stores (Giardina et al., 2023).

## 4   Discussion

### 4.1   Mechanisms Controlling Plant Responses to Drought

#### 4.1.1   Stomatal and Non-stomatal Regulation of Gross Primary Productivity

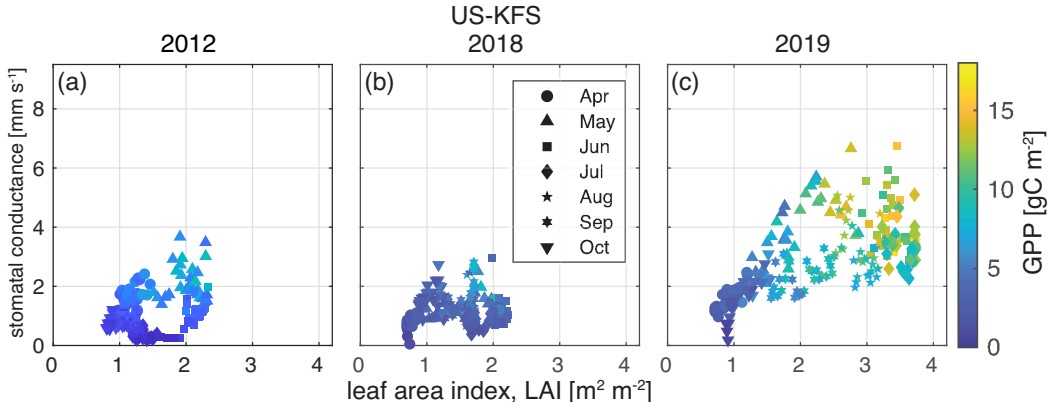

**Figure 14.** Stomatal conductance [mm s$^{-1}$] vs leaf area index, LAI [m$^2$ m$^{-2}$] for US-KFS for a flash drought year (2012), a drought year (2018), and a non-drought year (2019). Marker shapes indicate individual days between April 1 - October 31. Each month is given a unique shape whose color reflects daily accumulations of gross primary productivity [gC m$^{-2}$].

An objective of this work is to evaluate whether changes in phenology versus changes in stomatal conductance have a stronger control on carbon uptake during flash drought (H2, H3). We consider how GPP covaries during flash drought, drought, and non-drought years with sub-seasonal changes in LAI and stomatal conductance at US-KFS (Figure 14). During a non-drought year (2019), there exists a wider range of values of stomatal conductance, LAI, and GPP throughout the growing season (Figure 14c). There is also a clear seasonal cycle in the clockwise movement through the stomatal conductance-LAI
parameter space. Stomatal conductance increases faster than LAI in the early season before reaching maximum values around June. After LAI peaks, there is first a reduction in stomatal conductance and GPP at higher LAI before LAI decreases through August and September.

In contrast, during flash drought (2012) and drought (2018), peak stomatal conductance, LAI, and GPP values at US-KFS are approximately half of 2019 values. Both stomatal conductance and LAI remain low throughout the growing season and GPP is
below 10 gC m$^{-2}$ at all sites in 2012 (Figure S7). Stomatal conductance and LAI are highest in May 2012 as opposed to June and July 2019. While both 2012 and 2018 have low values of stomatal conductance, LAI, and GPP, an important difference is

the near-zero stomatal conductance during June and July 2012 for a range of LAI values (1-2 $m^2$ $m^{-2}$, Figure 14) that is not observed in 2018 and other drought years (Figure S11).

The relationship between stomatal conductance, LAI, and GPP is similar across all three sites when considering flash drought (Figure S7), drought (Figure S11), or non-drought periods (Figures S8, S9, S10). The observable clockwise movement through parameter space is not as clear in flash drought and drought as compared to non-drought. In drought years, stomatal conductance from April-October averages 1.4 mm $s^{-1}$ across all sites (Figure S11) compared to 2.3 mm $s^{-1}$ in non drought years (Figures S8, S9, S10) and 1.1 mm $s^{-1}$ in flash drought (Figure S7). Peak LAI is approximately 1-2 $m^2$ $m^{-2}$ higher in non-drought years compared to flash drought and other drought years. Similarly, non-drought GPP levels are approximately 6-8 gC $m^{-2}$ higher than flash drought and non-drought periods.

Prior work linked phenological responses to drought to changes in vegetation-atmosphere interactions (Lowman and Barros, 2018; Cui et al., 2017). Dynamically estimated FPAR and LAI tend to exert strong controls on the resulting GPP (Lowman and Barros, 2018). By updating phenological states using the phenology model rather than forcing phenology with remotely sensed values, we were able to capture the plant growth response to water availability. When more water is available, DCHM-PV simulation predicts higher values of FPAR, LAI, and thus higher values of GPP. At the onset of flash drought, DCHM-V and -PV respond faster to changes in LAI and FPAR than MODIS whose effects were also seen in differences in modeled and remotely sensed GPP (Figure 11). Moreover, regardless of the simulation, the rapidness of the change in LAI and FPAR is indicative of flash drought (Figures 5 and 6) and in agreement with Zhang et al. (2020). Decreases in phenological state due to the lack of soil water available to plants affected carbon and water exchanges, suggesting support for the third hypothesis (H3), however, decreases in stomatal conductance driven by increased VPD may compound the detrimental phenological effects.

### 4.1.2 Vapor Pressure Deficit Dependence

While phenology is an important component to consider when computing changes to transpiration and carbon uptake (Lowman and Barros, 2018; Flack-Prain et al., 2019), our results indicate that stomatal conductance is also critical for accurately representing these fluxes. Plants adaptively regulate their stomata during periods of water stress (Guo et al., 2020), and some have been demonstrated to maintain open stomata or even increase stomatal conductance under high VPD conditions (Urban et al., 2017). Stomatal conductance shuts down under high VPD in the DCHM (Figure 9), which does not account for the possibility of an adaptive stomatal regulation strategy. Since GPP is directly dependent on stomatal conductance (Farquhar and Sharkey, 1982), DCHM estimates of sub-daily GPP decrease in response to elevated VPD (Figure S22). Moreover, changes in phenological growth state (i.e. LAI) occur across longer (i.e. seasonal) time scales (Katul et al., 2001) than stomatal regulation, which controls carbon and water exchange at sub-daily timescales (Guo et al., 2020).

The differences between modeled and observed GPP and ET (Figures 11, 13) suggest that there are mechanisms controlling plant responses to drought stress not accounted for within the DCHM. For example, the DCHM could be too strict in representing the sensitivity of stomatal closure to elevated VPD for the Kansas study sites. There could be plant or climate specific VPD dependence (Grossiord et al., 2020), plants could have access to stores of water not accounted for (Giardina et al., 2023), or both. Guo et al. (2020) showed that isohydricity (i.e. stomatal regulation) exists on a spectrum and that some plants are able to

move along that spectrum at sub-daily time-scales with varying environmental conditions, such as higher VPD. Given the high VPD in 2012 at our study sites (Figures S5, S28, S29, S30), we expect the DCHM to estimate low stomatal conductance, and thus low GPP relative to AmeriFlux observations when under atmospheric water stress. Additionally, VPD estimated by the DCHM using the NLDAS-2 Forcing File A atmospheric variables is higher during 2012 and 2018 and lower in 2019 than the AmeriFlux observations (Figure S28), explaining in part the discrepancies between model and AmeriFlux GPP. As stomatal response to increasing VPD and resulting impacts on land-atmosphere water fluxes is more complex than how it is represented in LSMs (Vargas Zeppetello et al., 2023), future modeling studies should focus on how rising VPD drives stomatal closure across different vegetation types Grossiord et al. (2020).

## 4.2 Surface and Sub-surface Water Movement

### 4.2.1 Infiltration and Evaporation

At the onset of flash drought there is an increase in evaporative demand for water which leads to a temporary increase in surface evaporation (Lowman et al., 2023; Otkin et al., 2018) until the soil and canopy reservoirs no longer contain enough water to evaporate. Then evaporation shuts down. Despite evaporation tapering to zero during June and July of 2012 (Figure S19), pulses of rainfall lead to temporary rapid increases in rates of evaporation. Increased surface evaporation may reduce water infiltrating the soils. In May of 2012 at US-KFS there was 70 mm of water infiltrating the soils (Figure 7) with 35 mm of evaporation (Figure 12b). But in June and July total infiltration was 61 mm with 65 mm of evaporation over the two months. Similar comparisons can be found at US-KLS and US-Kon (Figures 7, 10). In contrast, at US-KFS, during non-drought years, June averages of infiltration are in excess of 100 mm with 41 mm of evaporation. Average drought years have 66 mm of infiltration with 47 mm of evaporation (Figure 7a). Since infiltration usually exceeds evaporation in the growing season, infiltration accumulations of similar magnitude to evaporation totals may indicate flash drought.

### 4.2.2 The Partitioning of Evapotranspiration

In this study, from 2006-2019, excluding 2012, growing season transpiration rates averaged more than 50% of total ET at US-KFS. This finding aligns with prior results from Hosseini et al. (2022) who used the Noah-MP LSM that also computes transpiration from root water uptake (Li et al., 2021). However, during the flash drought year, transpiration rates fell below 35% of overall ET at US-KFS (Figure 15a). Transpiration decreases approximately 40% from May to June at US-KLS (Figure 15b), and 20% at US-Kon (Figure 15c). The rapid decline in transpiration rates can be attributed to the slowing of root water uptake due to the lack of available water and decreased stomatal conductance (Figures S6, S7). In contrast, ET decreases at US-KFS during July 2019 while experiencing a brief period of low rainfall (Figure S19b), yet plants are able to maintain rates of GPP during this period due to the amount of available water in soils from the excessive precipitation during May and June (Figures S1c, S2c, S3c).

Accumulated monthly averages of transpiration as a fraction of evapotranspiration (T/ET) show a transition from at or above non-drought levels to at or below drought levels (Figure 15). At US-KFS drought years have a lower fraction of transpiration

throughout the growing season whereas drought and non-drought values are similar from July-October at US-KLS and US-Kon. US-Kon experiences larger fluctuations in the fraction of transpiration through the early and middle parts of the growing season (April - July). It is possible that the fluctuating T/ET at US-Kon, modeled as a grassland, is indicative of an adaptation to the water stresses.

## 4.3 Linking Carbon and Water Fluxes

Despite major reductions in infiltration and fluctuations in top layer soil moisture during flash drought onset, modeled root water uptake indicates that plants were still pulling small amounts of water through their roots, preventing them from completely shutting down. With the ability to tap into water stores from deeper layers (Giardina et al., 2023) and small rates of transpiration still occurring, modeled carbon uptake is still maintained (Figure 11a, S15a, S16a). Although GPP drastically slows, it does not stop. During the flash drought of 2012 (mid May - early July), we estimated steady declines in rates of GPP despite bursts in ET in response to rain recharge events (Figure S23a). We found evaporation to be the main contributor to total ET and the decreases in GPP followed changes in transpiration during flash drought (Figure S19a). The decreases in simulated GPP due to flash drought during June and July 2012 are consistent in terms of magnitude with decreases found in recent studies (Yao et al., 2022; Poonia et al., 2022; Zhang et al., 2020).

Plants are more efficient during non-drought periods, and are less efficient during flash drought onset (Figure 15). Ratios of T/ET also indicate plants that transpire more are more efficient in their water-use. WUE is similar at US-KFS in August-October in drought and non-drought years which might be attributed to the site being modeled as a cropland. WUE at all sites started off in 2012 with above average non-drought levels and an increase from April to May. However, from May-July WUE at all sites fell from above non-drought years to more than one standard deviation below drought years. With GPP differences being more substantial than ET between flash drought and non-flash drought periods (Figure 10), subseasonal reductions in WUE can be attributed to the losses in GPP. Reductions in WUE from above non-drought conditions to below drought conditions (e.g., the 60%-70% reduction from May to July in 2012, Figure 15 d,e,f), appear to be a feature of flash drought onset.

## 4.4 Uncertainty in Vegetation Responses

We implemented three different assimilation strategies to prepare ensemble parameters to be used in the predictive phenology routine in the DCHM-PV. The 2003-2005 period represented "average" conditions as it spanned periods of below and above average precipitation. Compared to the single year assimilation periods (WET and DRY), the uncertainty ranges in model parameters were smaller in the 3YR assimilation period. The results are consistent with Lowman and Barros (2018) in that uncertainty in phenology shrinks during dry periods. Daily standard deviations in LAI across simulations are approximately 0.5 m$^2$ m$^{-2}$ during the growing season of a wet year but shrink to values of 0.2 at the onset of flash drought and less than 0.1 during peak flash drought. The lower ensemble spread during the flash drought period corresponds with winter phenological variability when plants are dormant. Similarly, decreases in uncertainty in estimates of GPP and ET during the flash drought period fall to winter levels implying variability in plant life stage and functionality are similar in drought periods and dormant months.

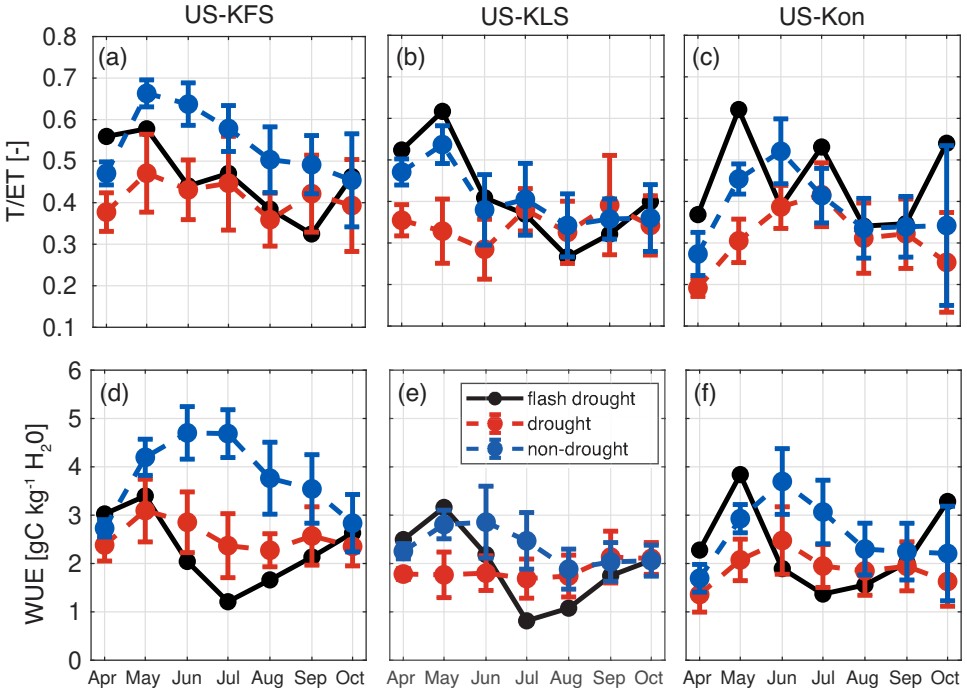

**Figure 15.** Ratio of transpiration to evapotranspiration, T/ET, and water-use efficiency, WUE, for for drought (red) and non-drought (blue) years compared to flash drought (black) for US-KFS, US-KLS, and US-Kon.

The $\gamma$ parameter values, which drive plant growth in the DCHM-PV, were all smaller in the 3YR assimilation period for all three test sites when compared to simulations from drought and wet years. Vegetation leaf out occurs later in the simulations from the 3YR assimilations (Figures 5 and 6). However, the more notable effects of the smaller growth parameters, with regards to flash drought, can be seen through the delayed phenology responses in the 3YR assimilations compared to the WET/DRY

assimilations. Across the three test sites, the FPAR and LAI decreases were slower in the simulations that used the 3YR assimilation period. The vegetation that was trained using average conditions was slower to change when faced with the abrupt decrease in water availability. Although it was an above average year for precipitation, there was little rainfall in early July 2019 at our Kansas sites. At US-KLS and US-Kon, there was a rapid decrease in LAI during this time (Figure 6d,f) with some recovery in August. Moreover, the slower changes in seasonal LAI and FPAR dynamics from the 3YR simulations show the

resiliency to abrupt phenological changes.

Future studies should use an assimilation period encompassing multiple precipitation regimes (i.e. multi year inference period) to best represent the variability of climatological conditions because it leads to reduced uncertainty in model outputs. However, if the intent of a future study is to investigate vegetation responses to extreme events in a changing climate (Kirono et al., 2020; Pearson et al., 2013, e.g.,), it may be appropriate to use inference periods encompassing extreme wet or dry

conditions. For example, one could fit parameters to a dry regime to investigate how plants accustomed to today's average conditions will function in a future climate where drier conditions are expected.

## 4.5 Land Cover Influences

The seasonal dynamics of FPAR, LAI, and GPP from the simulations match well against the remotely sensed observations from MODIS regardless of vegetation type. Effects of water stress on infiltration, stomatal conductances, and WUE are similar across all three sites and help to distinguish flash drought conditions from non-flash droughts. However, there are subtle differences in vegetation responses to water stress across all three sites. As seen in Figures 5 and 6, the phenological responses to flash drought at US-KLS (cropland) and US-Kon (grassland) follow a similar trajectory throughout the growing season. The savanna at US-KFS (note - AmeriFlux classifies US-KFS as a grassland) suggests more resilience to flash drought at first when compared to croplands and grasslands in that values of FPAR and LAI are maintained for a longer period before tapering in late June. This can also be seen in the slow reductions in GPP during May and June 2012 before reaching a minimum near the beginning of July marking stomatal closure and a shift toward more isohydric behavior (Meinzer, 2002).

When analyzing DCHM outputs against remotely sensed and eddy covariance measurements, we are comparing data across temporal and spatial scales. The flux towers exist within a 4 km by 4 km region defined by the Stage-IV spatial grid cell used in the DCHM. Flux tower spatial extents range from a couple hundred meters to a few kilometers (Baldocchi, 2003; Schmid, 1994) making the 4 km grid cell near the maximum range. Sub-grid scale heterogeneity can lead to considerable discrepancies between parameterized and actual fluxes (Schmid, 1994). One explanation for why flux tower data differs from model output is that the flux tower estimates incorporate a variety of vegetation types within the fetch contributing to the vertical fluxes, rather than the single vegetation type used within the model. Additionally, the size and orientation of the contributing fetch varies in time depending on measurement height and turbulent fluxes (Chu et al., 2021).

Differences in land cover classification could be another reason for discrepancies between modeled and observed FPAR and LAI. Though we use MODIS to determine the land cover type, we first interpolated the 500 m data to determine the value of the 4 km grid cell used in the DCHM. After upscale, the pixel at US-KFS is labeled as a savanna, but the 500 m MODIS grid cell containing US-KFS is classified as grassland. Regardless of the classification differences, the spectral reflectance method used by MODIS is inherently different from the predictive phenology routine used in the DCHM-PV, specifically in that it cannot account for how soil water availability influences vegetation growth (Lowman and Barros, 2018).

## 4.6 Model Performance and Limitations

### 4.6.1 Model vs Observations

This study allows us to investigate how vegetation responses can be used to study the effects of flash droughts on the total carbon and water budgets. Our modeling approach permits direct comparisons of remotely sensed observations to physically derived estimates. Generally, MODIS overestimates GPP compared to EC flux tower data (Heinsch et al., 2006; Running et al., 2004). Daily GPP from the DCHM tends to match the magnitude of MODIS and AmeriFlux GPP at US-KFS throughout much of the growing season but underestimates June and July observations in 2012 (flash drought) and 2018 (drought). The DCHM-PV tends to overestimate during 2019 (non-drought) while the DCHM-V more closely aligns with observations. Large discrepancies are also apparent in hourly estimates of GPP at US-KFS (Figure S22). The DCHM halts midday GPP in July

2018, but AmeriFlux values remain high. The differences are smaller in 2012, where AmeriFlux observed carbon assimilation rates of 1 gC m$^{-2}$ s$^{-1}$ throughout the daytime and the DCHM shut down carbon assimilation due to elevated VPD (Figure S22).

The DCHM-PV compares favorably against MODIS LAI during flash drought and non-drought at US-KFS and US-KLS (Figure 6a,c,d,f) and underestimate those sites during drought (Figure 6 b,e). At US-Kon, MODIS LAI during May, June, and July tends to be above DCHM-PV estimates. The higher DCHM-PV model estimates of FPAR and LAI during summer 2019 could be due to the model accounting for excess water availability and other meteorological conditions favorable for growth (temperature, VPD, etc.). MODIS estimates of FPAR and LAI are based on radiative transfer models using bidirectional reflectance of incoming radiation from the red and near infrared bands (Myneni et al., 2015; Yan et al., 2016). MODIS GPP is directly dependent on observations of FPAR (Running et al., 2015). This difference is apparent in DCHM-PV estimates of GPP exceeding estimates from the DCHM-V and MODIS GPP during the same period where the DCHM-PV predicts larger values of FPAR and LAI during 2019 (Figure 11). Our model performance against MODIS is similar to that found in Hosseini et al. (2022), who used a predictive phenology model coupled with Noah-MP. Across all 11 years in that study, their dynamic vegetation models tended to underestimate June and July LAI at US-Kon and slightly overestimate at US-KFS.

AmeriFlux estimates of GPP during June and early July of 2012 and 2018 are also above estimates from MODIS. This suggests that during drought and flash drought, plants are able to maintain higher levels of GPP than what can be recreated in land surface models and satellite remote sensing. Differences in DCHM-PV and AmeriFlux GPP cannot be fully attributed to carbon reallocation since the Noah-MP model accounts for carbon reallocation and similarly underestimated GPP compared to flux tower data (Hosseini et al., 2022). Even while accounting for carbon movement, they found that during June, July, and August they underestimated tower data by 100 gC m$^{-2}$ at US-Kon while overestimating by the same amount at US-KFS in April, May, and June (averaged across an 11-year study period encompassing wet and dry periods). The DCHM-PV, which does not account for carbon reallocation, responds to drought and flash drought differently than what is observed at flux tower sites. It matches better with AmeriFlux data during 2012, the flash drought year, at US-KFS and US-KLS compared to 2018, a drought year (Figure 11, S15).

Another difference between modeled and flux tower data could be that models may not be able to fully represent how vegetation can maintain ET by accessing groundwater or deep soil moisture, ultimately biasing models towards more severe effects of drought on vegetation (Giardina et al., 2023). DCHM has similar soil moisture profiles to NLDAS-2, derived from Noah-LSM, and Hosseini et al. (2022) who used Noah-MP configurations, for both the 2012 flash drought and the 2018 drought. The DCHM also follows trends similar to AmeriFlux in 2012, but AmeriFlux top layer soil moisture values are much smaller from May to October of 2018, often under 0.1 m$^3$ m$^{-3}$ during that time (Figure S1). Despite extremely low top layer soil moisture in 2018, AmeriFlux GPP reaches levels above 10 gC m$^{-2}$ d$^{-1}$ coinciding with a brief recharge in soil moisture at the end of June. The DCHM estimates of GPP are often less than 50 % of AmeriFlux GPP in 2012 and 2018. The model results from the Noah-MP similarly underestimate GPP and overestimate soil moisture during these drought periods (Hosseini et al., 2022) suggesting that access to deep water reserves are responsible for these differences (Giardina et al., 2023).

### 4.6.2 Implications for Land-surface Models

Capturing phenological responses and subsequent changes to carbon and water fluxes within a physically based model is not without its limitations. As we update phenological states during the DCHM-PV simulations, forced atmospheric conditions from NLDAS-2 and Stage-IV variables are the same as in the DCHM-V simulations. We continue to use these conditions to force the model, so it is possible that the meteorological observations are already accounting for some vegetation-atmosphere interactions. But, by explicitly considering plant tendencies, we can dynamically account for current meteorological conditions

and thus use physical principles to capture vegetation-atmosphere interactions.

    Vegetation responses to water stress are apparent through fluctuations in GPP (Zhang and Yuan, 2020; Jin et al., 2019) and ET (Chen et al., 2019). Decreases in GPP occur when plants close their stomata. With the stomata closed, plants will limit gas exchange affecting both photosynthesis and transpiration rates. Transpiration is only one part of ET, so we must be careful not to directly link fluctuations in GPP with fluctuations in ET. Evaporation can still be high when there is little to no transpiration,

but GPP tends to follow the same trajectories as transpiration (Figures 10, 12,  Beer et al., 2009). In some cases, vegetation can reallocate already processed carbon to their roots when under drought stress mitigating GPP losses (Ingrisch et al., 2020). However, modeled GPP losses are likely a result of modeled stomatal behavior, as the model does not account for reallocation of carbon stores within the plants. Sub-daily scale stomatal conductance reduces to zero in response to increased VPD (Figure 9) leading to similar reductions in modeled GPP (Figure S22). This limitation of the DCHM could explain why AmeriFlux

GPP tends to be higher than the modeled GPP.

    Vegetation activity is directly linked to the coupling of the water and carbon cycling through photosynthesis (Farquhar et al., 1980) and assimilating plant phenology into land-surface models (e.g., DCHM-V or Noah-MP) can improve estimates of GPP and ET (Hosseini et al., 2022; Xu et al., 2021; Mocko et al., 2021; Kumar et al., 2019). However, our findings also indicate that improved phenology cannot alone account for vegetation adaptations to water stress and ability to access water in ways

that current LSMs cannot account for (Giardina et al., 2023). Future studies should focus on improving our understanding how plants are able to tap into different stores of water to continue exchanging water and carbon despite lower precipitation or increased VPD. Additionally, as stomata control the movement of water and carbon, affecting GPP and water-use efficiency (Lawson and Vialet-Chabrand, 2019), accounting for plant adaptations that adaptively regulate stomatal sensitivity to drought stress, especially VPD, may improve model accuracy.

Moving forward, improvements made to phenological states of the entire plants (i.e. root systems included) rather than just the leaf phenology might better capture water movement through plants under water stress conditions. Future studies would benefit from improved estimates of root water uptake since it is directly linked to the amount of available water for transpiration. Vegetation types have distinct root characteristics leading to differences in hydraulic tendencies under variable water regimes and atmospheric conditions which distinguish vegetation that is more likely to survive or recover from drought

(McDowell et al., 2008; Martínez-Vilalta et al., 2002). Species specific hydraulic strategies may differ in a single location (Liu et al., 2020) so generalization of water-use by PFT in hydrologic models would represent the average tendency of vegetation to regulate water. It is also possible that the changing phenological state of root systems plays an important role in root water

uptake (McCormack et al., 2014). Moreover, models that can account for different vegetation behavior such as the reallocation of carbon storage and below ground respiration during drought may provide a better understanding of mechanisms driving drought resiliency and changes to carbon uptake during drought (Ingrisch et al., 2020; Sanaullah et al., 2012). These types of mechanisms could explain how a warm and wet spring mitigated the effects of the 2012 flash drought on GPP losses (Wolf et al., 2016).

## 5 Conclusions

To address how water stresses affect carbon and water cycling, we implemented a one-dimensional version of the DCHM-V coupled to a predictive phenology model and analyzed vegetation-atmosphere water and carbon exchanges during flash drought, drought, and non-drought periods. The modeling procedure first required running the DCHM-V with phenology updates from remotely sensed observations of FPAR and LAI. Coupling the predictive phenology model to the DCHM-V, we generated ensembles of model parameters and ran Monte Carlo simulations of the DCHM-PV with concurrent meteorological conditions. We ran three simulations using three distinct assimilation periods for three different sites in Kansas. Uncertainty in model parameters and outputs is reduced when a three year assimilation period (covering net-average conditions) is used.

Our findings indicate that both phenology and stomatal conductance play an important role controlling vegetation responses to extreme drought (H2, H3). Decreased infiltration due to increased days between precipitation during flash drought resulted in less soil water available for plant use (H1,H2). High vapor pressure led to stomatal closure within the model. With stomata closed, root uptake, transpiration, and carbon assimilation reduced to dormant levels which led to reductions in WUE during flash drought to levels more than one standard deviation below other drought periods (H2). FPAR and LAI also reduced during flash drought but did not exert as strong of a control on reductions in GPP, as observed with changes to stomatal conductance that resulted from increased VPD.

The seasonal timing of the flash drought was particularly detrimental as the rapid dry down occurred during the peak growing season. The amount of water available during the growing season has a major influence on vegetation activity. In this region of the United States, droughts can reduce monthly carbon assimilation by half as compared to non-drought periods while flash droughts are even more detrimental to the overall carbon budget. This has major implications for annual crop yields as well as the carbon uptake capacity for the grasslands and savannas that cover much of the Midwestern US. Future modeling studies should investigate how different vegetation types alter their water-use strategies in response to different water stresses by including (1) adaptive stomatal regulation under elevated VPD, (2) access to deep stores of water in soils and (3) wider ranges of plant functional types and climatological regimes.

*Data availability.* Results from model simulations are available for download via CUAHSI Hydroshare at
https://doi.org/10.4211/hs.331a4e26a36a48928817881a8f3e5db4

*Author contributions.* LL, TF, and JO originally conceived the idea for this paper. LL designed the methods and supervised implementation by NC. NC carried out all data processing and modeling work. NC and LL analyzed results. NC and LL wrote the manuscript and NC prepared the submissions. All authors contributed to reviewing and editing the manuscript drafts.

*Competing interests.* The authors declare that no competing interests exist.

*Acknowledgements.* The forcing data used in this study are available from a variety of sources. The NCEP/EMC Stage IV data were acquired from UCAR/NCAR - Earth Observing Laboratory and are avaiable at https://data.eol.ucar.edu/. NLDAS Phase 2, Noah-MP, and SMERGE data used in this study were acquired as part of the mission of NASA's Earth Science Division and archived and distributed by the Goddard Earth Sciences (GES) Data and Information Services Center (DISC). The MODIS MOD15A2H data were retrieved online from Appeears and distibuted by NASA Land Processes Distributed Active Archive Center (LP DAAC), http://appeears.earthdatacloud.nasa.gov/. Computations were performed using the Wake Forest University (WFU) High Performance Computing Facility, a centrally managed computational resource available to WFU researchers including faculty, staff, students, and collaborators (Information Systems and Wake Forest University, 2021). Data from this material is stored by CUAHSI via Hydroshare (http://www.hydroshare.org/resource/331a4e26a36a48928817881a8f3e5db4) with support from the National Science Foundation (NSF) Cooperative Agreement No. EAR-1849458. We acknowledge the following AmeriFlux sites for their data records: US-KFS, US-KLS, US-Kon. In addition, funding for AmeriFlux data resources was provided by the US Department of Energy's Office of Science. Maps were generated using data from the US Drought Monitor which is jointly produced by the National Drought Mitigation Center at the University of Nebraska-Lincoln, the United States Department of Agriculture, and the National Oceanic and Atmospheric Administration. This material is based upon work supported by the National Science Foundation under Award Number 2228047.

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
