# Peer review of "Unraveling phenological and stomatal responses to flash drought and implications for water and carbon budgets"

_Hydrology and Earth System Sciences, 2023_

## Author Comment (AC1)

Responses to Anonymous Reviewer 1 for paper: *Unraveling phenological to extreme drought and implications for water and carbon budgets*

The authors would like to thank the reviewer for taking the time to make thoughtful comments that will improve the manuscript. The questions, insights, and suggestions will help us to clarify, shape, and focus the readers on the main point of the article: that land-atmosphere interactions undergo rapid changes during flash drought not observed during drought or non-drought periods.

We respond to each of the Reviewer's comments below, which are in **bold and italics**. Author responses are in blue with proposed manuscript changes in **bold**. New figures that we propose to include have labels N# throughout this document. Revised figures have the same figure number as in the original manuscript or point to the figure from the manuscript they will replace. Figures used to support claims in this document that are not going into the revised manuscript do not have Figure numbers but we have included figure captions.

Given the updated analysis and comparison of flash drought to non-flash drought periods, the authors use the following language throughout this document and will update accordingly in the revised manuscript. We use "flash drought" when referring to 2012 and "drought" or "drought years" for all other droughts. We use "non-drought" for all years that are not drought or flash drought.

The authors noticed the following major themes from the Reviewer comments.

1. A need to clarify differences between flash drought and drought
   a. Hypotheses need to better address what distinguishes flash drought from drought (i.e., timing, magnitude of effects, etc.)
   b. Strengthen analyses by using all available years from model outputs when comparing results from the flash drought year against 'non-flash' drought years
2. Some claims could be better supported with figures using model outputs that were not previously used, specifically:
   a. Including results from infiltration
   b. Exploring the relationship between phenology, GPP, stomatal conductance, and transpiration
3. Differences between model predictions, flux tower measurements, and MODIS observations need more discussion.
   a. Model values of GPP and ET were much lower than AmeriFlux and MODIS GPP during flash drought
   b. Discussion of observed differences should explore plant processes (e.g., reallocating carbon, tapping into deep groundwater, stomatal regulation, etc.) that the DCHM may be missing.

Response to Major Comment 1a:

We updated our hypotheses to distinguish flash drought from other droughts. They include language that allows us to answer more specifically how land-atmosphere interactions differ in a flash drought from other times. The hypotheses now read:

**H1 During flash drought, there is an increase in days between precipitation events leading to larger reductions in total precipitation and infiltration as compared to non-flash drought events.**

**H2 Lower total infiltration observed during flash drought reduces soil water available for root water uptake, subsequently leading to reduced rates of transpiration, carbon uptake, and water use efficiency as compared to non-flash drought within a subseasonal time frame.**

**H3 In response to decreased water availability during flash drought, vegetation phenological states will be diminished as compared to non-flash drought years exacerbating the reduction of transpiration and carbon uptake.**

Response to Major Comment 1b:

We now present our results using all years of available model output and framing the analyses in terms of flash drought vs "non-flash" drought conditions. We used the United States Drought Monitor (USDM, Svboda et al., 2002) to determine "drought" and "non-drought" years in Kansas. The Central and East Central Kansas climate regions contain the three study sites (see figure from USDM below). From the USDM time series of the climate regions, drought years were determined if an entire year was in drought or if parts of the region reached D2 (Severe Drought) or higher. The years 2006, 2011, 2013, 2014, and 2018 are labeled as drought years for analysis. The years 2007-2010, 2015-2017, and 2019 are labeled as non-drought years. The flash drought year 2012 is kept separate from other drought years in the analyses. This change will be updated in the Methods section as well (see response to specific comment below).

[Figure]

Figure. Percent land area in the U.S. Drought Monitor Categories for two Kansas climate regions that contain our study sites US-KFS, US-KLS, US-Kon.

Response to Major Comment 2a:

Throughout the paper, including the hypotheses, we made claims about infiltration and its relation to evaporation and root water uptake without showing any infiltration results. Hypothesis 1 has been updated (see above) to incorporate how timing of rainfall events impact total infiltration. To investigate this hypothesis we include plots of monthly accumulations of infiltration, evaporation, and root uptake, and figures relating infiltration to the amount of time between precipitation events to build a better understanding of how infiltration affects available water for plant use. We also address this in the discussion and conclusion. See below for new analyses related to specific comments.

Response to Major Comment 2b:

We take a closer look at sub-daily GPP and stomatal conductance during selected weeks throughout the growing season for a flash drought, drought, and non-drought year in order to evaluate how stomatal conductance reduces with increased vapor pressure deficit (VPD) and leads to subsequent declines in transpiration and GPP. We also investigate whether reductions in stomatal conductance or leaf area index (LAI) have a larger impact on GPP in order to evaluate if there exists a phenological dependence during flash drought. See below for new analyses related to specific comments.

In light of new analyses during the process of responding to the Reviewer comments, we propose a new title: ***Unraveling phenological and stomatal responses to flash drought and implications for water and carbon budgets***

Response to Major Comment 3a:

One reason for the discrepancies between modeled output and flux tower data was that plots of daily average rates of GPP and ET had to do with how we were calculating daily averages for the figure. While it made sense to average these variables over the entire 24-hour period for the flux tower data, the model shuts off GPP and evaporation when there is no incoming solar radiation leading to zeroes during half of the day. Thus, we only average GPP and ET over the active period within the model to avoid including the unrealistic zeros. This is how the DCHM model results were previously presented in Lowman and Barros (2016, 2018) and Lowman et al. (2018). Presenting the results differently was a mistake that has now been fixed. Additionally, we were able to find available gap-filled time series for GPP for US-KFS and US-KLS from the AmeriFlux FLUXNET database (Pastorello et al., 2020), allowing us to make comparisons where previous data was missing in the analysis.

Response to Major Comment 3b:

We propose to add the following text to the Discussion section of the manuscript to address this major comment:

**In a recent paper, Giardina et al. (2023) argue that observed plant responses to water stress indicate the ability of plants to access deep groundwater and other stores of water that land surface models (LSMs) are not accounting for. The DCHM has similar soil moisture profiles to Noah-LSM and Hosseini et al., 2022, who used Noah-LM configurations, for both the 2012 flash drought and the 2018 drought. The DCHM also follows trends similar to AmeriFlux in 2012, but AmeriFlux top layer soil moisture values are much smaller from May to October of 2018, often under 0.1 $m^3\ m^{-3}$ during that time (Figure A1). Despite extremely low top layer soil moisture in 2018, AmeriFlux GPP reaches levels above 10 $gC\ m^{-2}\ d^{-1}$ coinciding with a brief recharge in soil moisture at the end of June. The DCHM estimates of GPP are often less than 50% of AmeriFlux GPP in 2012 and 2018. The model results from the Noah-LSM similarly underestimate GPP and overestimate soil moisture during these drought periods (Hosseini et al. 2022), suggesting that access to deep water reserves are responsible for these differences (Giardina et al. 2023).**

[Figure]

**Figure A1 (2018 to be added to manuscript.) Top layer soil moisture at US-KFS from the DCHM compared to AmeriFlux, Noah-LSM (from NLDAS-2). Daily precipitation totals are indicated on the top axis.**

**Using predictive phenology with NOAH-LM, which can account for carbon reallocation to leaves, stems, roots, and soils, Hosseini et al. (2022), compared predicted estimates to flux tower measurements of GPP. Even while accounting for carbon movement, they found that during June, July, and August they underestimated tower data by 100 gC m^{-2} at US-Kon while overestimating by the same amount at US-KFS in April, May, and June (averaged across an 11-year study period encompassing wet and dry periods). The DCHM-PV, which does not account for carbon reallocation, performs similarity to the Hosseini et al. (2022) results, suggesting the accounting for carbon allocation cannot explain underestimating GPP in 2012 and 2013. The DCHM-PV compares more favorably against AmeriFlux data during 2012, the flash drought year, at US-KFS and US-KLS (Figure 8, A11) compared to 2018, a drought year. This suggests that there are missing processes in both the DCHM and the NOAH-LM that cannot capture plant water use during drought, and cannot be attributed to carbon allocation.**

**During drought and flash drought, DCHM-PV values also compare favorably with MODIS and tend to be slightly higher than MODIS during a non-drought year like 2019. During drought and flash drought, the DCHM-V and DCHM-PV tend to follow similar trajectories**

but in response to little water stress, the predictive phenology model predicts increased carbon uptake compared to the DCHM-V results which align more with MODIS in 2019. AmeriFlux estimates of GPP during June and early July of 2012 and 2018 are also above estimates from MODIS. This suggests that during drought and flash drought, plants are able to maintain higher levels of GPP than what can be recreated in land surface models and satellite remote sensing. Differences in DCHM-PV and AmeriFlux GPP cannot be attributed to carbon reallocation since the NOAH-LM model accounts for carbon reallocation and similarly underestimated GPP compared to flux tower data (Hosseini et al. 2022). The working hypothesis is that plants have access to deeper water stores than can be accounted for in land surface models, as suggested by Giardina et al. (2023).

[Figure]

Figure 8 (updated). Time series of gross primary productivity, GPP, at US-KFS for (a) 2012, flash drought, (b) 2018 drought and (c) 2019 a non-drought year. One standard deviation is shown for the DCHM-PV simulations. MODIS GPP are shown as red crosses and AmeriFlux GPP as small dots.

[Figure]

**Figure A11 (updated). Time series of gross primary productivity, GPP, at US-KLS for (a) 2012, flash drought, (b) 2018 drought and (c) 2019 a non-drought year. One standard deviation is shown for the DCHM-PV simulations. MODIS GPP are shown as red crosses and AmeriFlux GPP as small dots.**

*This papers offers a detailed analysis of drought responses for vegetation in the Midwestern US using an ecohydrologic model and assimilation of MODIS FPAR and LAI data along with flux tower data. The paper addresses some important ecohydrologic questions about how rapidly transpiration and carbon assimilation decline during drought and how changing phenology, specifically above-ground photosynthetic capacity, accelerates GPP losses. The paper utilizes advanced modeling techniques and builds on previous work that has established useful ways to assimilated remote sensing data into the DCHM model.*

*While that paper has significant potential, it did not really make strategic use of the model and observations to address some of the questions posed in the introduction.*

*For example, the hypothesis posed do not really address the issues of 'flash' drought.*

*In H2 The idea that drought causes both carbon uptake and transpiration to decline is something that is quite well understood - and there is ample evidence that this occurs- we know plant shut down when they run out of water. There are elements of the timing of this that are perhaps less well understood - and questions about how water use efficiency changes during a drought- and indeed the authors get at this to some extent in the paper - The hypothesis should reflect this. There would also be ways to frame the study (and hypothesis) to look at the relative impact of "flash" drought versus other*

*types of drought that could be interesting. Some additional thinking about how to use the model to test more nuanced (and informative) hypothesis would be strengthen this paper*

The Reviewer's comments about H2 and other hypotheses are well taken. We have edited the Hypotheses in the paper to clearly articulate changes we expect to see in land-atmosphere interactions during flash drought that differ from drought (see response to Major Comment 1a above). The following changes were made to H1, H2, and H3.

H1 now addresses differences in timing of precipitation events during flash drought, drought, and non-drought years. H1 now reads as follows.

**H1 During flash drought, there is an increase in days between precipitation events leading to larger reductions in total precipitation and infiltration as compared to non-flash drought events.**

H2 now has language that broadens the types of vegetation responses we see due to decreased infiltration. We also have language to compare vegetation responses across flash, regular, and non-drought periods.

**H2 Lower total infiltration observed during flash drought reduces soil water available for root water uptake, subsequently leading to reduced rates of transpiration, carbon uptake, and water use efficiency as compared to non-flash drought within a subseasonal time frame.**

H3 was updated to draw comparisons of flash drought to non-flash drought years and to include more specific language other than "plant-atmospheric interactions".

**H3 In response to decreased water availability during flash drought, vegetation phenological states will be diminished as compared to non-flash drought years exacerbating the reduction of transpiration and carbon uptake.**

We also updated Figure 1 to incorporate the edited hypotheses.

[Figure]

**Figure 1 (updated). Schematic of water, carbon, and energy fluxes with hypotheses about ecological response to flash drought indicated with orange arrows. Decreased frequency of precipitation events leads to decreased infiltration and less water available for plant use during flash drought as compared to non-flash drought periods. The cascading effects of the decreased water availability, exacerbated by the decreased phenological states, include rapid reductions in transpiration and atmospheric carbon uptake to levels below drought.**

*Throughout the paper, there are statements made that are not well supported by graphs or analysis - for example - that evaporation exceeds infiltration (this is indirectly shown but it would be much convincing to show this directly - and model results could do this). In another perhaps more salient example for the paper, the authors state phenology declines reduce carbon and water exchanges (H3) - one could argue that because plants have already shut down stomates at that point in the season, phenological declines do not further reduce transpiration - I'm not suggesting this is true but graphs presented do not clearly rule this out in the testing of H3.*

The Reviewer's comments are well taken. We have combed through the paper to make sure that any claims are fully substantiated and demonstrated with figures or data from our results. We have taken a critical look at unsubstantiated claims throughout the paper as referenced by this review. In doing so, we also noticed that we could better support the updated hypotheses by incorporating additional results and analyses. For changes to hypotheses, see response to Major Comment 1a above.

Regarding the first comment on evaporation exceeding infiltration, we acknowledge that we previously did not show any results for infiltration. In addition to updating the wording of our hypotheses, we will add a subsection to our results section of the manuscript that discusses infiltration.

We propose to add a section in the results **Section 3.X Plant Available Water** with subsections for infiltration and root water uptake.

**Section 3.X.1 - Infiltration**

**During non-drought years, infiltration is over 100 mm per month, on average, from April to July with the highest rates in May (Figure N1). Conversely, during drought years, infiltration in April-July is less than non-drought years, and mean infiltration is often one standard deviation below infiltration in non-drought years. Furthermore, monthly accumulated infiltration is less for the flash drought year compared to both drought and non-drought years, indicating there is less available water for plant use. This is the case at US-KFS from April-October of 2012, with September totals similar to drought years. A drastic decline in May infiltration at US-KLS and US-Kon indicated infiltration rates that are 1-2 standard deviations below average drought conditions. All sites had infiltration rates below 100 mm for all months during 2012 with the exception of August at US-KLS.**

Low infiltration totals during the flash drought year can be attributed to lower precipitation accumulations coupled with an increase in the number of days between precipitation events (Figure N2) and an increase in atmospheric demand for water (Figure N4). During the drought, and non-drought years, the average number of days between rainfall events within a month ranges from 1 to 7 days, while the lower end for the flash drought year is at 2.5 days. Additionally, during drought and non-drought years, monthly infiltration exceeds 150 mm, but remains at or below 75 mm for all sites aside from August 2012 at KLS where monthly infiltration is ~110 mm. In 2012, all three sites averaged over four days between rainfall events during May, June, and July with US-KFS averaging over six days between rainfall events during both May and June and more than five days in July (Figure N2a). Across all three sites from April-October 2012, there were more than four days between precipitation events 80% percent of the time compared to just 20% of the time in non-flash drought years.

[Figure]

Figure N1. DCHM-PV 3 YR ensemble means of monthly infiltration accumulations for drought (red dashed line) and non-drought (blue dashed line) years compared to 2012 (black solid line) for all three study sites. Monthly sums are computed from the ensemble means of the 2000 Monte Carlo simulations then averaged across drought or non-drought years. Error bars represent one standard deviation across drought and non-drought years, respectively. Drought years are 2006, 2011, 2013, 2014, 2018 and non-drought years are 2007-2010, 2015-2017, 2019.

[Figure]

**Figure N2. Monthly infiltration accumulation plotted against the monthly average of days between precipitation events for (a) US-KFS, (b) US-KLS, and (c) US-KON. Each shape indicates one month over which the averaging occurred and colors distinguish flash drought (black) from drought (red) and non-drought (blue) years.**

[Figure]

**Figure N3 (appendix addition). Monthly accumulation of infiltration versus precipitation. Each shape indicates one month over which the averaging occurred and colors distinguish flash drought (black) from drought (red) and non-drought years (blue).**

[Figure]

**Figure N4 (appendix addition). Monthly average vapor pressure deficit [kPa] for the three AmeriFlux sites from April - October for the flash drought year 2012 (black), drought years (red), and non-drought years (blue). The error bar represents one standard deviation across drought and non-drought years.**

In addition to a new section on infiltration, we will also incorporate root water uptake to this new section as a subsection.

**Section 3.X.2 Root Water Uptake**

**Root water uptake is lower in flash drought than non-flash drought years for the period June to September (Figure A7). The middle soil layer is responsible for up to four times more root water uptake than the other layers so a major decline in root water uptake through the middle layer demonstrates how root water uptake is altered during drought (Figure A7). While root water uptake starts out in 2012 at levels above average non-drought years, it falls to more than one standard deviation below drought averages by July. This drastic shift is likely due to lower infiltration (Figure N1) and drives down rates of transpiration within the model over the same period.**

[Figure]

**Figure A7 (Replacing A7 and A8). DCHM-PV 3YR monthly root water uptake totals for drought (red) and non-drought (blue) years compared to 2012 (black) across three soil layers for our three study sites. Monthly sums are computed from the ensemble means of the 2000 Monte Carlo simulations then averaged across drought or non-drought years. Error bars represent one standard deviation across drought and non-drought years, respectively. Drought years are 2006, 2011, 2013, 2014, 2018 and non-drought years are 2007-2010, 2015-2017, 2019.**

Response to second part of comment:

The authors appreciate the reviewer bringing awareness about a potential counterpoint to our claim that *"phenology declines reduce carbon and water exchanges (H3) - one could argue that because plants have already shut down stomates at that point in the season, phenological declines do not further reduce transpiration - I'm not suggesting this is true but graphs presented do not clearly rule this out in the testing of H3."*

We acknowledge that there may be more at play with phenology, stomatal conductance and plant-atmosphere interactions which has led to new ways of interpreting the model output. Below, is a new proposed section that evaluates the relative contributions of changes in stomatal conductance vs LAI during the flash drought. We also address this with the proposed change in title.

**Section 3.XX Stomatal Conductance**

**An objective of this work is to evaluate whether changes in phenology vs changes in stomatal regulation have a stronger control on carbon uptake and transpiration during flash drought. We consider how GPP co-varies during flash drought, drought, and**

non-drought years with sub-seasonal changes in LAI and stomatal conductance. During flash drought, both stomatal conductance and LAI remain low throughout the growing season and GPP falls below 10 gC m^{-2} (Figure N7).

As expected during non-drought years, there exists a wider range of values of stomatal conductance, LAI, and GPP throughout the peak growing season (Figure N6). In non-drought years, there is a clear seasonal cycle in the clockwise movement through the stomatal conductance-LAI parameter space. Stomatal conductances increase faster than LAI in the early season before reaching maximums around June-July. After LAI peaks, there is first a reduction in stomatal conductance and GPP at higher LAI before LAI decreases through August-September.

Once stomatal conductance values surpass 4 mm s^{-1}, we observe increases in GPP. LAI seems to have some effect on GPP, but not the same magnitude as stomatal conductance. At peak season values of LAI above 3 m^2 m^{-2}, we observe smaller fluctuations in GPP than observed with stomatal conductance. (Figures N6 and Appendix). When LAI decreases in the latter part of the growing season, stomatal conductances can still remain at relatively constant levels. It appears that stomatal conductance exerts a stronger control on daily GPP than LAI.

In drought years, stomatal conductance from April-October averages 1.4 mm s^{-1} across all sites (Figure N5) compared to an 2.3 mm s^{-1} in non drought years (Figure N6) and 1.1 mm s^{-1} in flash drought (Figure N7) .  While both 2012 and 2018 have low values of stomatal conductance, LAI, and GPP, an important difference is the near-zero stomatal conductance during June and July for a range of LAI values (1-2 m^2 m^{-2}, Figure N7) that is not observed in 2018 and other drought years aside from US-KLS in 2011 (Figure N5l).

In 2012, stomatal conductances in the first week of May were as high or higher than in 2019, a non-drought year (Figure N9) at US-KFS. By July, due to increased VPD, stomatal conductances diminished (Figure N9b). Compared to a drought year, e.g. 2018 where stomatal conductance rates are similar in all three months, 2012 undergoes a major shift. This is likely due to plants regulating their stomata under dry atmospheric conditions and less likely attributed to the phenological changes as FPAR and LAI were similar in 2012 and 2018. In 2019, a non-drought year, some August stomatal conductance and GPP rates are slightly higher than in July at US-KFS despite decreased phenology (Figure N8c) suggesting that drought-induced changes in phenological state does not have as strong as a control on carbon and water exchanges as we have hypothesized.

[Figure]

**Figure (N5 - appendix addition). Daily stomatal conductance [mm s^{-1}] vs leaf area index, LAI [m^2m^{-2}] for all three sites and selected drought years. Marker shapes indicate an individual day from April 1 - October 31 from the selected drought year. Each month is given a unique shape and daily accumulations of gross primary productivity [gC m^{-2}] are indicated by color.**

[Figure]

**Figure (N6 will be added to the appendix. Similar figures for US-KLS and US-Kon will also be included). Daily stomatal conductance [mm s^{-1}] vs leaf area index, LAI [m^2m^{-2}] for US-KFS for selected non-drought years. Marker shapes indicate individual days from**

**April 1 - October 31 from the selected drought year. Each month is given a unique shape and daily totals of gross primary productivity [gC m^{-2}] are indicated by color.**

[Figure]

**Figure (N7 appendix addition). Daily stomatal conductance [mm s^{-1}] vs leaf area index, LAI [m^2m^{-2}] for all three sites during the flash drought of 2012. Marker shapes indicate individual days from April 1 - October 31 from the selected year. Each month is given a unique shape and daily totals of gross primary productivity [gC m^{-2}] are indicated by color.**

[Figure]

**Figure (N8). Stomatal conductance [mm s^{-1}] vs leaf area index, LAI [m^2m^{-2}] for US-KFS for 2012, 2018, and 2019. Marker shapes indicate individual days between April 1 - October 31 from the year. Each month is given a unique shape and daily accumulations of gross primary productivity [gC m^{-2}] are indicated by color.**

[Figure]

**Figure (N9). Hourly stomatal conductances [mm s^{-1}] for one week in May, July, and August of 2012, 2018, and 2019 compared with vapor pressure deficit (VPD, kPa) for US-KFS.**

*The author also make statements about flash droughts but do not really distinguish flash drought from other types of drought in their analysis - They have multiple years that they do not really make use of.*

The authors thank the review for this comment. This is one of the major comments we address in Major Comment 1 above and help drive most of the new analysis that better distinguishes flash drought from other drought and non-drought conditions.

As shown above, we are now making direct links to a drought year (2018) as well as averaging across drought and non-drought years. We determined the drought and non-drought years from the USDM (Svoboda et al., 2002). We will update the methods Section 2.7 beginning at line 243 with the change in bold.

Line 243: "…We highlight results from the three AmeriFluxes sites for 2012 (flash drought), **2018 (drought),** and 2019 (**non-drought)** to draw conclusions about plant response during flash drought **and how they differ from drought and non-drought years.** We also evaluate **model outputs** from 2006-2019 to assess the **differences** between the DCHM-V and DCHM-PV model configurations **during drought and non-drought years compared to a flash drought year. During this time period, we identified drought years as 2006, 2011, 2013, 2014, 2018 and non-drought years as 2007-2010, 2015-2017, 2019 using the USDM for the Central and East Central Kansas climate regions (Svoboda et al., 2002). Drought years were determined by whether parts of the region reached the D2 "Severe Drought" classification or higher. When computing drought and non-drought averages, we use the years listed here. Transpiration is calculated from total root water uptake through the three soil layers and total evaporation is computed from summing evaporation from ground and canopy surfaces allowing us to partition ET into evaporation and transpiration."**

Also to address this comment, we enhance our analysis to highlight that flash drought should be considered as the time period leading up to drought and the intensification rate (Otkin et al.,

2018). As such, we will put more emphasis on the change leading from non-drought to drought (May to July of 2012) in results and discussion.

*Finally there are also important differences between modeled and flux tower data that may be critical in the understanding flash drought responses. These differences need to be more rigorously explored (see detailed comments below)*

The reviewer's comment is appreciated. As noted above in response to Major Comment 3, the reason for the discrepancies was an error made when creating the figures that has now been fixed. Please see the updates in Figures 8 (above in response to Major Comment 3) and Figure 9 below.

[Figure]

**Figure 9 (updated). Time series of evapotranspiration, ET, at US-KFS for (a) 2012, flash drought, (b) 2018 drought and (c) 2019 a non-drought year. Two standard deviations are shown for the DCHM-PV simulations. AmeriFlux ET is derived from latent heat measurements and shown as small dots.**

One possible explanation for why flux tower data differs from model output is that the flux tower estimates incorporate a variety of vegetation types within the fetch contributing to the vertical fluxes, rather than the single vegetation type used within the model. Additionally, the size and orientation of the contributing fetch varies in time depending on measurement height and turbulent fluxes (Chu et al., 2021). Another difference could be that models may not be able to fully represent how vegetation can maintain ET by accessing groundwater or deep soil moisture, ultimately biasing models towards more severe effects of drought on vegetation (Giardina et al., 2023).

Further, the flux towers exist within a 4 km by 4 km region defined by the StageIV spatial grid cell used in the DCHM. Flux tower spatial extents range from a couple hundred meters to a few kilometers (Baldocci, 2003, Schmid, 1994) making the 4 km grid cell near the maximum range. Subgrid scale heterogeneity can lead to considerable discrepancies between parameterized and actual fluxes (Schmid, 1994). Since the DCHM treats the entire grid cell as a single vegetation type, our results hold some uncertainty as we cannot account for the heterogeneous mix of vegetation and land-use present on the ground (see Figure below). Inside of this grid is deciduous forest that could influence tower readings, that the DCHM does not account for.

[Figure]

Figure (non included in manuscript). US-KFS AmeriFlux tower site at the center of a 4 km by 4 km grid representing vegetation heterogeneity of the surrounding region.

We propose to update Section 4.5 Limitations:

Line 465: Capturing phenological responses **and subsequent changes to carbon and water fluxes** within a physically based model is not without its limitations.

We propose to remove lines 470-480, beginning with "For example…"

Line 470: …temporal and spatial scales. **The flux towers exist within a 4 km by 4 km region defined by the StageIV spatial grid cell used in the DCHM. Flux tower spatial extents range from a couple hundred meters to a few kilometers (Baldocci, 2003, Schmid, 1994) making the 4 km grid cell near the maximum range. Subgrid scale heterogeneity can lead to considerable discrepancies between parameterized and actual fluxes (Schmid, 1994). One explanation for why flux tower data differs from model output is that the flux tower estimates incorporate a variety of vegetation types within the fetch contributing to the vertical fluxes, rather than the single vegetation type used within the model. Additionally,**

the size and orientation of the contributing fetch varies in time depending on measurement height and turbulent fluxes (Chu et al., 2021).

Another difference between modeled and flux tower data could be that models may not be able to fully represent how vegetation can maintain ET by accessing groundwater or deep soil moisture, ultimately biasing models towards more severe effects of drought on vegetation (Giardina et al., 2023). Using predictive phenology with NOAM-LM, which can account for carbon reallocation to leaves, stems, roots, and soils, Hoessini et al. (2022), compared predicted estimates to flux tower measurements of GPP. Even while accounting for carbon movement, they found that during June, July, and August they underestimated tower data by 100 gC m^{-2} at US-Kon while overestimating by the same amount at US-KFS in April, May, and June (averaged across an 11-year study period encompassing wet and dry periods). The DCHM-PV, which does not account for carbon reallocation, responds to drought and flash drought differently than what is observed at flux tower sites. It matches better with AmeriFlux data during 2012, the flash drought year, at US-KFS and US-KLS (Figure 8, A11) compared to 2018, a drought year.

During drought and flash drought, DCHM-PV values also agree favorably with MODIS and tend to be slightly larger than MODIS during a non-drought year like 2019. During drought and flash drought, the DCHM-V and DCHM-PV tend to follow similar trajectories but in response to little water stress, the predictive phenology model predicts increased carbon uptake compared to the DCHM-V results which align more with MODIS in 2019. Drought levels of AmeriFlux observed GPP during June are above observed non-drought levels. Even during flash drought, GPP tended to be slightly higher than non-drought June levels. This suggests that during drought and flash drought, plants are able to maintain higher levels of GPP. Differences in DCHM-PV and AmeriFlux GPP less likely to be attributed to carbon reallocation since another predictive model that accounts for carbon reallocation underestimated AmeriFlux values, and more likely due the suggestion by Giardina et al., (2023) that vegetation is likely accessing water stores in ways that are different from how land surface models are currently representing it.

*Some detailed comments.*

*H1 is actually two hypothesis - it would be useful to separate them*

See response to Major Comment 1 above regarding our updated hypothesis H1.

*The data sets and modeling proposed here tend to focus on relatively shallow surface soil - including citations of expected rooting depths for the PPTs would be helpful support for the implementation (especially given that flux tower observations of ET tend to be higher than the model in dry years)*

The review comments are well taken. We propose two changes to the paper beginning at line 189.

1. For clarity, we remove parenthetical depths in line 189-190 since they are stated more clearly in the following sentence.
2. We add a reference to expected rooting depths using a combination of soil and PFT.

Line 189: **"We maintain the use of 8 cm for the top soil layer for model stability, but use 35 in for root zone depth and 72 in for the depth to the impermeable layer (Soil Survey Staff). This yields the three soil layers: top (0-8 cm), middle (8-89 cm) and bottom (89-183 cm) consistent with Kansas soil surveys. Rooting depths in the DCHM are determined using exponential root distribution functions whose input parameters are determined by PFT ( Zeng, 2001 and Jackson et al., 1996). Soil layer and rooting depths align with the different combinations of soil textures and PFTs found in Thornthwaite and Mather (1957)."**

*line 225 - Some additional (just one or two sentences) information about of how ensembles of phenology parameters are established is needed here (e.g what is done for each of the 3 periods to select the 2000 parameter distributions shown in Figure 4) - There needs to be a bit more context so that reader understands Figure 4 and what controls the variation in parameter sets.*

We propose to address this comment by adding the following text on lines 215-221, where we put more emphasis on the data assimilation procedure.

Line 217: Using **soil water potential and VPD** outputs from the DCHM-V and updating…

Line 219: ...and mixed conditions). **We implement a dual state-parameter EnKF that simultaneously predicts FPAR and LAI and parameters that determine the growing season index. The parameter estimation procedure consists of initializing ensembles of parameters by sampling them from Gaussian distributions (Lowman and Barros, 2018), and predicting phenological state variables from the sampled parameters at every timestep (hourly). When MODIS LAI and FPAR are available (every 8 days), we update ensembles of the parameters in the DCBP before updating predictions of phenological state variables (this is the data assimilation step) to ensure that predictions of FPAR and LAI do not stray far from observations. The data assimilation period determines whether the generated parameters represent wet, dry, or mixed precipitation regimes.**

*line 235 - The simulation period is relatively short - and isohydric-anisohydric differences may or not be distinguishable within the 3 years - thus you cannot really state that the vegetation model parameters trained on dry conditions will represent isohydroic vegetation?. Especially given that parameter values seem to change depending on period (Figure 4) but vegetation PFT does not.*

The authors appreciate the reviewer's comments.

Lowman and Barros (2018) showed that assimilation period can determine the water stress adaptations for the modeled vegetation state. We propose the following edits beginning at line 235.

Line 235: …affects the development of flash drought. **It has been shown under varied climatological conditions plants can be highly adaptable, transitioning from isohydric to anisohydric in a single season (Guo et al., 2020). Lowman and Barros (2018) showed that assimilation period can determine the water stress adaptations for the modeled vegetation state.** Broadly speaking vegetation model parameters trained using **data from years with minimal rainfall represent plants that are accustomed to drier conditions and therefore exhibit more regulation in their water use tendencies (Lowman and Barros, 2018)**.

*line 246 - That transpiration is calculated from root water uptake makes sense but it doesn't follow that this allows you to "to partition ET"…you would need to have a separate calculation of total ET to do that. Clarify*

The reviewer's comment is well taken. The model computes total ET (Figure 9) from totaling surface evaporation from soil and canopy and from computing transpiration as root water uptake through the three soils layers. We address the partitioning in the addition to the Methods Section 2.7 above and copied here:

Line 243: "…We highlight results from the three AmeriFluxes sites for 2012 (flash drought), **2018 (drought),** and 2019 (**non-drought**) to draw conclusions about plant response during flash drought **and how they differ from drought and non-drought years.** We also evaluate **model outputs** from 2006-2019 to assess the **differences** between the DCHM-V and DCHM-PV model configurations **during drought and non-drought years compared to a flash drought year. During this time period, we identified drought years as 2006, 2011, 2013, 2014, 2018 and non-drought years as 2007-2010, 2015-2017, 2019 using the USDM for the Central and East Central Kansas climate regions (Svoboda et al., 2002). Drought years were determined by whether parts of the region reached the D2 "Severe Drought" classification or higher. When computing drought and non-drought averages, we use the years listed here. Transpiration is calculated from total root water uptake through the three soil layers and total evaporation is computed from summing evaporation from ground and canopy surfaces allowing us to partition ET into evaporation and transpiration."**

*Line 259 - For clarity it would be helpful to be consistent in the naming conventions- e.g gamma or growth parameter not both*

We now use the gamma once it is defined throughout the remainder of the manuscript.

*Line 258- in what way is this in agreement with Lowman and Barros (e.g the choice of longer period for reducing uncertainty) - in a way that's not so surprising - more information usually reduces uncertainty?*

The authors appreciate the need for clarification. Line 258 now reads: **"... in agreement with Lowman and Barros (2018) who found that using assimilation periods with both wet and dry conditions has the effect of capturing adaptive plant water use strategies."**

*Line 259 That gamma values vary by site could be do to differences in climate (note that game values vary across wet and dry years) - so it is not a given that it varies by plant functional type - rather this is an assumption (e.g I think that you are assigning plant functional type parameters based on this analysis)- The wording of this paragraph could make that point more clearly*

The reviewer's comments are well taken. The gamma parameter may vary based on differences in both climate and land cover type. We propose the following change:

Line 259: The values of $\gamma$ vary by site due to **a combination of local climate and plant functional type (PFT)**. **US-KFS, modeled as a savanna, has the lowest value of gamma and standard deviation.** The smaller magnitudes…

*Line 265 - "slower" relative to what?*

We appreciate this comment and have incorporated language making more clear comparisons like the one needed here throughout other parts of the analysis. In particular, we make sure to note comparisons between "flash drought versus drought or non-drought years". This language also helps to distinguish between the model results from drought and non-drought periods from WET and DRY assimilation periods.

Line 264: …slower senescence and reduced variance when using the 3YR assimilation parameters as **compared to the WET and DRY parameters** during…

*The rationale for the continued focus (beyond Figure 4) of differences due to parameters sets based on wet, dry or both years is unclear - Given that using both wet and dry years clearly reduces uncertainty, I'm not sure why there is a need to compare estimates of FPAR, LAI, The authors may have a reason for this but if so it needs to be emphasized in the text. Removing this would allow the focus to be on DCHM-PV performance and the "actual" phenological mediated vegetation responses.*

The reviewer's comments are well taken.

Figure 4 helps to establish the 3YR assimilation period as the inference period with minimal uncertainty. However, we still have the goal to capture differences in model outputs across the three different assimilation periods. This is done in Figures 5,6,7, and 9 which show FPAR, LAI, yearly sums of GPP and ET, and daily ET, respectively. Figures 5 and 6 support the conclusion that the 3YR inference period does indeed show slower change in phenological state compared to WET and DRY. We agree that further comparison with the WET and DRY results is no longer needed in the body of the manuscript. We can include WET and DRY results in the appendix. We replaced Figure 7 with monthly accumulations of GPP and ET from the DCHM-PV 3YR. We removed the DCHM-PV WET and DRY from Figure 9.

The WET inference period predicts parameters using model outputs of soil water potential and VPD during a year that received ample rain, and therefore plenty of water for plant use, and less atmospheric demand for water. We are assuming that this inference period represents

vegetation that is not accustomed to water stress and therefore is less conservative in water use strategies (Lowman and Barros, 2018). This means that when plants run low on available water (e.g., in 2012), they will consume water normally and exacerbate dry down before abruptly shutting down functions dictating water and carbon cycling. This is in contrast to parameters produced during the DRY inference period. The assumption there is that vegetation will be more conservative in water use strategies, shutting down at the first sign of stress. In both cases, mean $\gamma$ values are higher than the 3YR inference period meaning that vegetation during the 3YR inference period is more likely to make steady changes and adaptations to water stress and less likely to make abrupt changes as seen with both the WET and DRY simulations. However, the difference between 3YR, WET, and DRY is generally minimal in terms of the magnitude of water and carbon exchanges and detailed discussion of these results is no longer included in the analysis.

*Section 3.12 and Section 3.1.3 - If I understand the methods correctly - Figure 6 and 7 show results using parameters conditions on prior information (e.g MODIS assimilation during the calibration period) and MODIS results for 2012 and 2019 - Given that, additional discussion about fit with MODIS would be helpful - How well does DCHM-PV do. Overall it captures patterns fairly well but there are some notable exceptions (e.g loss of FPAR in July and August at US-KFS in MOIDS that is not tracked by model) - It would be useful to have some presentation of model performance here*

We appreciate the reviewer's comments. We think these comments refer to Figures 5 and 6 regarding FPAR and LAI. We updated Figures 5 and 6 to include 2018 (see below). We propose to provide additional text comparing MODIS to the DCHM-PV FPAR and LAI estimates.

Before section 3.2 Vegetation responses:

Line 309: **Generally, the predictive phenology model compares favorably with the seasonal changes observed in MODIS FPAR and LAI (Figures 5 and 6). In the summer, at US-KFS and US-KLS during 2019, the model tends to predict FPAR and LAI values higher than MODIS. In 2019, at US-KFS, MODIS observed a steady decline in FPAR from 0.8 to 0.6 throughout July with an increase back to 0.8 over an 8-day period at the beginning of August (Figure 5c). The DCHM-PV results do not show the same decline. Similarly for LAI at US-KFS (Figure 6c), MODIS observes a drop and then abrupt increase in LAI with the model estimates higher than MODIS. Yet, in June 2019 at US-Kon, the model estimates are lower than MODIS LAI.**

[Figure]

**Figure 6. Time series of leaf area index (LAI) predicted from DCHM-PV for the flash drought year (2012), a drought year (2018), and a non-drought year (2019). Colors indicate the different data assimilation periods (yellow - 3YR (2003-2005), blue - WET (2005), red - DRY (2003)), with corresponding shaded regions representing one standard deviation of model outputs from the 2000 ensemble simulations. The 8-day MODIS MOD15A2H LAI is shown in black markers. The gray shaded region highlights the June to July decrease in FPAR during the 2012 flash drought.**

To be added to discussion:

**The higher model estimates of FPAR and LAI during summer 2019 could be due to the model accounting for excess water availability and other meteorological conditions (temperature, VPD, etc.) resulting in an increase in FPAR and LAI. MODIS estimates FPAR and LAI are based on radiative transfer models using bidirectional reflectance of incoming radiation from the red and near infrared bands (Myneni et al., 2015; Yan et al., 2016). MODIS GPP is directly dependent on observations of FPAR (Running & Zhao, 2015). This difference is apparent in DCHM-PV estimates of GPP exceeding estimates from the DCHM-V and MODIS GPP during the same period where the DCHM-PV predicts larger values of FPAR and LAI during 2019 (Figure 8). Our model performance against MODIS is similar to that found in the Hosseni et al. (2022), who also used a predictive phenology model coupled with Noah-LM. Across all 11 years in that study, their dynamic vegetation models tended to underestimate June and July LAI at US-Kon and slightly overestimate at US-KFS.**

**The DCHM-PV tends to compare favorably against MODIS during flash drought and non-drought at US-KFS and US-KLS (Figure 6 a,c,d,f) and underestimate those sites during drought (Figure 6 b,e). At US-Kon, MODIS LAI during May, June, and July tends to**

be above DCHM-PV estimates. Differences between our model at 4 km and MODIS at 500 m spatial scales could depend on vegetation heterogeneity. Differences in land cover classification could be another reason for discrepancies between modeled and observed FPAR and LAI. Though we use MODIS to determine the land cover type, we take the most frequent land cover type at 500 m to determine the value of the 4 km grid cell. After upscale, the pixel at US-KFS is labeled as a Savanna but the 500 m MODIS grid cell containing US-KFS is classified as grassland. Regardless of the classification differences, the spectral reflectance method used by MODIS is inherently different from the predictive phenology routine used in the DCHM-PV, specifically in that it cannot account for how soil water availability influences vegetation growth (Lowman and Barros, 2018).

*Line 315 - which are water stress years (e.g 2012). Also can you note which method (or averaged across all methods) does the 1kgCm2 reduction come from*

The authors appreciate this comment from the reviewer. We are replacing Figure 7 and this paragraph has been rewritten so lines 310-319 will be removed. Instead, we compare DCHM-PV 3YR results across drought and non-drought years as listed above instead of showing yearly sums of GPP and ET.

**3.2.1 GPP**

**Monthly averages of GPP accumulations from DCHM-PV ensemble means throughout the water year (April - October) indicate that carbon uptake falls below drought averages from May to June during the flash drought year of 2012 (Figure 7 a,c,e). Flash drought carbon assimilation amounts remain below drought levels before converging to average drought/non-drought levels by the end of October. GPP levels are consistently up to 50% lower in drought years compared to non-drought years. During the flash drought, GPP monthly totals in June to August 2012 are at least one standard deviation lower than drought years over the 2006-2019 simulation period. June GPP in 2012 is 40-50% of drought and less than 30% of non-drought years. An even greater discrepancy comes in July with carbon assimilation amounts being less than 30% of drought levels and 15% of non-drought levels.**

Line 321: …for the flash drought year (2012), **a drought year (2018), and a non-drought year (2019). GPP rates are lower in drought than non-drought, but the decline in rates of GPP from at or above average in May to near zero by July in 2012 (Figure 8, A11, A12). During drought, rates of GPP are low but steady throughout the growing season. The rapid decline in GPP over the two month period is what distinguishes the 2012 flash drought as a period of time when rates of transpiration and carbon uptake are rapidly changing. Our results match closely with MODIS GPP, but the DCHM-PV overestimates MODIS in a drought year, which corresponds with the overestimation of MODIS phenology at the same time (Figure 6). Simulated GPP tends to underestimate flux tower GPP during June and July. From June to July in 2012, carbon uptake decreased from about more than 5.0**

**(depending on site) to less than 1.0 gC m^{-2}day^{-1}. These types of declines do not happen in a drought year (e.g. 2018).**

[Figure]

**Figure 7 (replacement). DCHM-PV 3YR monthly totals of GPP (a,c,e) and ET (b,d,f) for drought (red) and non-drought (blue) years compared to 2012 (black) for US-KFS, US-KLS, and US-Kon AmeriFlux sites. Monthly sums are computed from the ensemble means of the 2000 Monte Carlo simulations then averaged across drought or non-drought years. Error bars represent one standard deviation across drought and non-drought years, respectively.**

We replace yearly totals in Figure 7 with monthly averages of accumulated GPP and ET for drought and non-drought years. We also include monthly averages of ET separated as evaporation and transpiration in a new figure.

Line 328: **Monthly accumulations of ET for the flash drought years are shown against accumulations averaged across regular and non-drought years for the three study sites (Figure 7 b,d,f). Accumulations are from DCHM-PV 3YR simulations and averaged across ensemble members. ET accumulations are lower in the flash drought year, particularly at US-KLS and US-Kon. ET amounts during drought periods are slightly lower but generally similar to non-drought at US-KFS and US-KLS, indicating that total ET may not be a strong indicator of drought. However, looking at the same accumulations with ET parsed into evaporation and transpiration offer a different perspective.**

**Simulated transpiration accumulations follow similar trajectories as GPP during flash drought (Figure N10 a,c,e). Indeed, transpiration rates are higher during flash drought than non-drought years in April but quickly level off before declining from May to July. July transpiration totals in 2012 fall below one standard deviation of drought years. At US-KFS, evaporation rates are not too different from normal drought and only slightly higher than non-drought years throughout the entire growing season (Figure N10b).**

**Evaporation accumulations follow different trajectories at all three study sites but have transpiration patterns that link transpiration and GPP.**

[Figure]

**Figure N10. Simulated evapotranspiration partitioned into evaporation and transpiration from the DCHM-PV 3YR. Transpiration accumulations are computed from root water uptake through the three soil layers. Monthly averages are computed from the ensemble means of the 2000 Monte Carlo simulations then averaged across drought (red) or non-drought (blue) years. The flash drought year 2012 is shown in black. Error bars represent one standard deviation across drought and non-drought years. Drought years are 2006, 2011, 2013, 2014, 2018 and non-drought years are 2007-2010, 2015-2017, 2019.**

*lLine 380 - The arguments in the first 3 sentences of this paragraph need a bit more detail. Just because there are fluctuations in evaporation this doesn't not necessarily mean that "all" water evaporated before it had a chance to infiltrate.*

We believe this quote comes from line 340. The authors appreciate this comment and we agree. The new infiltration section described in Major Comment 2a supports the claim that reduced infiltration drives down plant water availability.

The timing of infiltration and evaporation are different. While infiltration and evaporation occur in response to precipitation events, the timescales associated with each process are different. Water infiltrates the soils following precipitation but may not be available for plant use in instances when increased VPD (Figure N4) leads to enhanced evaporation of soil water. It was therefore prudent to account for accumulations over monthly timescales to better compare infiltration and evaporation totals.

Infiltration does occur, but it is drastically reduced during flash drought. Water availability is reduced due to both a lack of precipitation and increased days between precipitation events (Figures N1, N3) leading to reductions in root uptake (i.e. transpiration, Figures A7, N10). The large fluctuations in ET are therefore more associated with increased evaporative demand, reduced infiltration limiting root water uptake,and overall lack of water availability.

 Line 339: …are a result of evaporation in response to precipitation (Figure 10a). This suggests **that following precipitation events during flash drought onset, ET is dominated by evaporation. Reduced available water infiltrating the soils limits water available for root water uptake.** Since…

[Figure]

**Figure 10 (updated). Time series of evapotranspiration, ET, at US-KFS partitioned into evaporation and transpiration for (a) 2012, flash drought, (b) 2018 drought and (c) 2019 a non-drought year. The curves represent ensemble means from the DCHM-PV 3YR.**

*Note the substantial underestimation of ET by the models relative to Ameriflux in 2012 should be noted here as well along with some discussion of why*

This comment and suggestion from the review is well taken. Please see response to Major Comment 3a above.

As noted above, modeled ET should only consider the daytime values when computing daily averages because the model shuts down evaporation at nighttime when there is no incoming solar radiation. However, in the original manuscript we computed daily averages over a 24 hour period. With the update to Figure 9, we see that the modeled results match well against AmeriFlux. Differences still occur in 2012, with modeled ET agreeing with AmeriFlux starting in April through mid-May. Once the flash drought onsets (late May through July), modeled results tend to be lower than AmeriFlux. Once the flash drought ends in August, modeled ET once

again agrees with AmeriFlux. In a drought and non-drought year, modeled ET appears to match better throughout most of the season (Figure 9). One explanation could be that water use by vegetation during flash drought is considerably different, and the model is not able to recreate this change in survival strategy.

In the results section which will be updated to be section 3.2.2 ET, we intend to add some of the above explanation to the first paragraph. We will also remove some of the comparisons of results that involve the DCHM-PV WET and DRY since we have agreed that analysis should focus on 3YR. We intend to cut the current line 332-337 and replace it with the following.

Line 332: …beginning of the growing season (April). **Modeled ET results match well against AmeriFlux at US-KFS during flash-drought, and non-flash drought periods (Figure 9). Differences still occur in 2012 at US-KFS, with modeled ET agreeing with AmeriFlux starting in April through mid-May. Once the flash drought onsets (late May through July), modeled results tend to be lower than AmeriFlux. Once the flash drought ends in August, modeled ET once again agrees with AmeriFlux. In a drought and non-drought year, modeled ET appears to match better throughout most of the season (Figure 9). While model estimates of ET are higher than flux tower measurements in non-drought at US-KLS, they match favorably in drought and flash drought (Figure A13). In contrast to model and flux tower comparisons at US-KFS and US-KLS, at US-Kon modeled ET (Figure A14) performs best in 2019 (non-drought) compared to 2012 (flash drought) and 2018 (drought). One explanation for the varied differences between model and tower ET data could be that water use by vegetation during flash drought is considerably different, and the model is not able to recreate this change in survival strategy. Depending on plant function type and soil characteristics, it may be difficult for the DCHM, or other models, to account for plant access to deep water stores (Giardina et al, 2023).**

[Figure]

**Figure 9 (updated). Time series of evapotranspiration, ET, at US-KFS for (a) 2012, flash drought, (b) 2018 drought and (c) 2019 a non-drought year. Two standard deviations are shown for the DCHM-PV simulations. AmeriFlux ET is derived from latent heat measurements and shown as small dots.**

[Figure]

**Figure A13 (updated). Time series of evapotranspiration, ET, at US-KLS for (a) 2012, flash drought, (b) 2018 drought and (c) 2019 a non-drought year. Two standard deviations are shown for the DCHM-PV simulations. AmeriFlux ET is derived from latent heat measurements and shown as small dots.**

[Figure]

**Figure A14 (updated). Time series of evapotranspiration, ET, at US-Kon for (a) 2012, flash drought, (b) 2018 drought and (c) 2019 a non-drought year. Two standard deviations are shown for the DCHM-PV simulations. AmeriFlux ET is derived from latent heat measurements and shown as small dots.**

*line 360 - Its worth noting that the drought response can be more complicated than simply shutting stomata - importantly grasses can shift their allocation of carbon - and this will be reflected in above ground biomass (the GPP measured by MODIS) but also in below ground stores and fluxes - For example see Ingrisch, Johannes, Stefan Karlowsky, Roland Hasibeder, Gerd Gleixner, and Michael Bahn. "Drought and recovery effects on belowground respiration dynamics and the partitioning of recent carbon in managed and abandoned grassland." Global Change Biology 26, no. 8 (2020): 4366-4378.*

This helpful comment from the reviewer is well taken. Stomata closure implies less gas exchange. Ultimately, this drives the decreases in modeled losses to ET and GPP, but could explain the higher GPP observed in the AmeriFlux data. Vegetation, at least grasses, can reallocate already processed carbon to their roots when under drought stress mitigating GPP losses (Ingrisch, et al., 2020). This means that MODIS may see a reduction in phenological states (LAI and FPAR) but maybe GPP is less affected.

Line 360: …ET (Chen et al., 2019). **In some cases, vegetation can reallocate already processed carbon to their roots when under drought stress mitigating GPP losses (Ingrisch, et al., 2020). However, modeled GPP losses are likely a result of modeled stomatal behavior, as the model does not account for reallocation of carbon stores within the plants. This limitation of the model could explain why AmeriFlux GPP tends to be higher than the modeled GPP. Sub-daily scale stomatal conductance reduces to zero in response to increased VPD (Figure N9) leading to similar reductions in modeled GPP (Figure N11).**

[Figure]

**Figure N11. Hourly gross primary productivity [g C m^{-2} s^{-1}] from the DCHM-V and DCHM-PV shown against AmeriFlux 30-minute estimates for one week in May, July, and August of 2012, 2018, and 2019 atUS-KFS.**

*Similarly on line 380 - there is ample evidence of changing root allocation (and root respiration) for grasslands that would be worth citing here.*

The reviewer's comment is well-taken. We now discuss changing root allocation and respiration with relevant citations. See edited text below.

Line 380: …leading to differences in hydraulic tendencies under variable water regimes **and atmospheric conditions which distinguish vegetation that is more likely to survive or recover from drought (McDowell et al., 2008; Martinez-Vilalta et al., 2002).**

Line 384 …role in root water uptake. **Moreover, models that can account for different vegetation behavior such as the reallocation of carbon storage and below ground respiration during drought may provide a better understanding of mechanisms driving drought resiliency and changes to carbon uptake during drought (Ingrisch et al., 2020; Sanaullah et al., 2012). These types of mechanisms could explain why results of studies like Wolf et al., (2016) who found that a warm and wet spring mitigated the effects of the 2012 flash drought on GPP losses.**

*line 369 - Note that authors don't really show that evaporation after rain effects uses all available water - so it doesn't infiltrate (as stated in the hypothesis) - but they could do this at least with the model since both daily evaporation and precipitation is available.*

We appreciate this comment and it is part of a larger theme addressed in the response to Major Comment 2. Note from Figure N3, precipitation exceeds infiltration. Excess amounts of water not contributing to infiltration can be attributed to runoff.

Line 365: layer…**Monthly infiltration (Figure N1) and evaporation accumulations (Figure 7 b,d,f) for June and July during the flash drought at all three sites are similar. However, in drought and non-drought periods, infiltration far exceeds evaporation. This implies that even as water is infiltrating the soils, comparable amounts are evaporating contributing to the soil drying down, leaving less new water available for plant use. In the time series of GPP during June and July,** at the onset of flash drought…

Line 367 …evaporation shuts down. **Despite evaporation tapering to zero during June and July of 2012 (Figure 10), pulses of rainfall lead to temporary rapid increases in rates of evaporation before falling back down. In May of 2012 at US-KFS there was 70 mm of water infiltrating the soils (Figure N1) with 35 mm of evaporation (Figure N10b). But in June and July total infiltration was 61 mm with 65 mm of evaporation over the two months. Similar comparisons can be found at US-KLS and US-Kon (Figures N1 and N10). In contrast, at US-KFS, during non-drought years, June averages of infiltration are in excess of 100 mm with 41 mm of evaporation. Average drought years have 66 mm of infiltration with 47 mm of evaporation (Figure N1a). Infiltration usually exceeds evaporation in the growing season, but infiltration accumulations comparable to evaporation totals appears to be an indication of flash drought.**

*line 375 - "since GPP is decreasing" as the authors themselves note - declines in GPP do not always reflect what's happening with transpiration so this statement needs some caveats*

The point is well taken, especially with grasses as noted above, reallocating their stores of carbon which can be another reason why modeled GPP isn't solely tied to changes in transpiration. However, the DCHM strongly links transpiration to GPP and does not account for reallocating carbon storages in other ways. We will use a new paragraph at Line 367-370 that leads into the current paragraph at line 371.

Line 367: **Despite major reductions in infiltration and fluctuations in top layer soil moisture during the peak of flash drought onset, modeled root water uptake indicated that plants were still pulling small amounts of water in through their roots, preventing plants from completely shutting down. With the ability to tap into water stores from deeper layers (Giardina et al., 2023) and small rates of transpiration still occurring, modeled carbon uptake is still maintained (Figure 8a, and A11a, A12a). Although it drastically slows, it does not stop.**

*In the discussion, one of the challenges here is that observations and models suggest differences in plant ability to pull water from deeper layers - The first paragraph blurs these distinctions - so for example the line "we did find that even during the peak..plants were still" - is this based on flux towers, or model?. I also note that model and observed in Figure 11 suggest different stories about water use efficiency during the drought. The observed data suggests plants maintain higher water use efficiency, longer, during the drought than the model - this is very interesting - its suggests plants are doing something that the model misses which is informative - but needs to be much more of a focus in the paper.*

The reviewer's comments are well taken.

We now include language in the paragraph above (previous response, Line 367) to indicate that we were referring to model outputs regarding root water uptake. In the larger context, we acknowledge that there may be discrepancies between models and flux tower measurements due to below ground processes that models are missing (see response to Reviewer Comment 3b above).

Figure 11 has been updated to average over daytime hours for water use efficiency. Corresponding figures for US-KLS and US-Kon are included in the appendix. Updates to figure 11 include the results from averaging WUE over the daytime and the addition of 2018. We also updated the legend and y-axis label as well as an extension of the viewing window to include all of October. We now align with tower values better in 2012. However, in 2018 the DCHM underestimation of tower WUE can be attributed to the differences in the 2018 modeled and flux tower measurements of GPP since modeled ET at US-KFS matches well against tower measurements. See the updated Figures 8 and 9 above for reference to GPP and ET, respectively.

[Figure]

**Figure 11. Computed growing season water use efficiency (WUE=GPP/ET) from the DCHM-V, DCHM-PV 3YR, and AmeriFlux for (a) 2012 (b) 2018, drought (c) 2019, non-drought at US-KFS. AmeriFlux WUE is computed by converting latent heat into eT by dividing by the coefficient of vaporization.**

*Figure 12 - its not so easy to see from this figure that when plant are transpiring more they are more efficient in their water use - Simply graphing transpiration vs WUE would show this much more clearly.*

The reviewer's comments are well taken and appreciated. In fact, our results agree with the point that plants transpiring more are not necessarily more efficient in their water use.

Figure 12 has been updated below. The main point of the new Figure 12 is to show that transpiration as a percentage of overall ET and WUE follow the same transition from normal or above average levels to drought levels from May to July. This suggests that both follow the same pattern and can be used to mark the rapid transition to drought state.

Given the new analysis and figures of transpiration accumulation and average water use efficiency in drought and non-drought years compared with non-drought years we can better observe the trends associated with WUE and transpiration. The new Figure 12 suggests that plants are more efficient in non-drought years. There is an association with decreased T/ET and decreases WUE. However, that does not substantiate the claim that "plants that are transpiring more are more efficient." We no longer like to make this claim.

[Figure]

**Figure 12 (replacement). Modeled growing season monthly averages of transpiration as a fraction of ET (a-c) and water use efficiency (WUE, d-f) for drought (red) and non-drought (blue) years compared with the flash drought year of 2012 (black) for US-KFS, US-KLS, and US-Kon AmeriFlux sites. Monthly averages are computed from the ensemble means of the 2000 Monte Carlo simulations then averaged across drought or non-drought years. Error bars represent one standard deviation across drought and non-drought years. Drought years are 2006, 2011, 2013, 2014, 2018 and non-drought years are 2007-2010, 2015-2017, 2019.**

Figures 11 and 12 were updated. Figure 11 now includes a daily average time series at US-KFS for 2018 in addition to 2012 and 2019. Corresponding figures for US-KLS and US-Kon will be included in the appendix figures replacing A17 and A18. Figure 12 was updated to compare flash drought T/ET ration and WUE to aggregate monthly averages from drought and non-drought years. We also added October to the analysis period.

New comments for Figure 12.

A more apt comment using Figure 12 in the discussion is
Line 378: **Plants are more efficient during non-drought periods, and are especially inefficient during flash drought onset (Figure 12). Accumulated monthly averages of transpiration as a fraction of evapotranspiration show a transition from at or above non-drought levels to at or below drought levels. At US-KFS drought years show a lower fraction of transpiration throughout the growing season whereas drought and non-drought values are similar from July-October at US-KLS and US-Kon. US-Kon experiences larger fluctuations in the fraction of transpiration through the early and middle parts of the growing season (April - July). It is possible that US-Kon, modeled as a grassland, is showing an adaptation to the water stresses. Another possible explanation**

for fluctuations in T/ET is that at the same time E/ET is fluctuation in response to precipitation and increased atmospheric demand for water.

Water use efficiency is generally higher in non-drought years compared to drought, though WUE is similar across all non-flash drought years from August-October at US-KLS. WUE at all sites started off in 2012 with above average non-drought levels and an increase from April to May. However, from May-July WUE at all sites falls from above average to more than one standard deviation below drought years. With GPP differences being more substantial than ET between flash drought and non-flash drought periods (Figure 7), subseasonal reductions in WUE (WUE=GPP/ET) can be attributed to the losses in GPP. These reductions in WUE from above average drought conditions to below drought conditions (e.g., the 60%-70% reduction from May to July in 2012, Figure 12 d,e,f), could be used as an indicator of flash drought onset.

From Figure N12, we can see that changes in transpiration explain a small component, less than 10%, of the variability observed in WUE. Generally when plants have more available water (e.g., 2019), they transpire more, but higher values of WUE can be seen in flash drought and drought years (e.g. 2012, 2018) despite having lower rates of transpiration.

[Figure]

**Figure N12 (appendix). Daily averages of water use efficiency versus transpiration for 2012, 2018, and 2019.**

*Also in figure 12 - some strategic use of color to differentiate wet versus dry years would be helpful here*

We appreciate this comment.

Upon initial submission of the manuscript, the processing editor suggested making some changes and updates to make plots and figures more readable. Changing to figures include updated color schemes, font sizes, line weights, line styles, and markers. Color schemes were generated using a tool Coloring for Colorblindness https://davidmathlogic.com/colorblind/#%23D81B60-%231E88E5-%23FFC107-%23004D40 and figures were tested on Coblis - Color Blind Simulator https://www.color-blindness.com/coblis-color-blindness-simulator/.

*Line 403 - The idea of flash drought responses is intriguing but I think the paper could do much more to support these ideas (e.g that the rapidness of the change is indicative of a flash drought) - Some more strategic comparison of the declines across different type of drought - flash versus "non-flash" (Of course this would require clearly distinguish what a flash drought is from other types of drought - but that seems to be part of the paper's motivation)*

The authors thank the reviewer for this thoughtful comment. A major goal of the authors' responses to the reviewer comments throughout this document has been to refocus analysis to compare results (e.g., GPP, ET, WUE, etc.) from a flash drought year to drought and non-drought years (see response to Major Comment 1). It is our intent that the rapid changes seen from May to July in 2012 model results of vegetation-atmosphere interactions along with the decreased precipitation and increased atmospheric demand for water highlight the transitional period of flash drought intensification. Furthermore, we hope this emphasizes flash drought as being the rapid development of drought that can be observed through land-atmosphere interactions.
* * *
Concluding Remarks

The authors would like to thank the reviewer again for their comments, questions, and suggestions that help improve this manuscript. The authors acknowledge that the updated analysis necessitates updated language throughout the manuscript reflecting the major changes presented here.
* * *
**References**

1. Baldocchi, D. D. (2003). Assessing the eddy covariance technique for evaluating carbon dioxide exchange rates of ecosystems: past, present and future. *Global change biology*, *9*(4), 479-492.
2. Chen, L. G., Gottschalck, J., Hartman, A., Miskus, D., Tinker, R., & Artusa, A. (2019). Flash drought characteristics based on US drought monitor. *Atmosphere*, *10*(9), 498.

3. Chu, H., Luo, X., Ouyang, Z., Chan, W. S., Dengel, S., Biraud, S. C., ... & Zona, D. (2021). Representativeness of Eddy-Covariance flux footprints for areas surrounding AmeriFlux sites. *Agricultural and Forest Meteorology*, *301*, 108350.

4. Giardina, F., Gentine, P., Konings, A. G., Seneviratne, S. I., & Stocker, B. D. (2023). Diagnosing evapotranspiration responses to water deficit across biomes using deep learning. *New Phytologist*.

5. Guo, J. S., Hultine, K. R., Koch, G. W., Kropp, H., & Ogle, K. (2020). Temporal shifts in iso/anisohydry revealed from daily observations of plant water potential in a dominant desert shrub. *New Phytologist*, *225*(2), 713-726.

6. Hosseini, A., Mocko, D. M., Brunsell, N. A., Kumar, S. V., Mahanama, S., Arsenault, K., & Roundy, J. K. (2022). Understanding the impact of vegetation dynamics on the water cycle in the Noah-MP model. *Frontiers in Water*, *4*, 925852.

7. Ingrisch, J., Karlowsky, S., Hasibeder, R., Gleixner, G., & Bahn, M. (2020). Drought and recovery effects on belowground respiration dynamics and the partitioning of recent carbon in managed and abandoned grassland. *Global Change Biology*, *26*(8), 4366-4378.

8. Jackson, R. B., Canadell, J., Ehleringer, J. R., Mooney, H. A., Sala, O. E., & Schulze, E. D. (1996). A global analysis of root distributions for terrestrial biomes. *Oecologia*, *108*, 389-411.

9. Lowman, L. E., & Barros, A. P. (2016). Interplay of drought and tropical cyclone activity in SE US gross primary productivity. *Journal of Geophysical Research: Biogeosciences*, *121*(6), 1540-1567.

10. Lowman, L. E., & Barros, A. P. (2018). Predicting canopy biophysical properties and sensitivity of plant carbon uptake to water limitations with a coupled eco-hydrological framework. *Ecological Modelling*, *372*, 33-52.

11. Lowman, L. E., Wei, T. M., & Barros, A. P. (2018). Rainfall variability, wetland persistence, and water–carbon cycle coupling in the Upper Zambezi river basin in Southern Africa. *Remote Sensing*, *10*(5), 692.

12. Martínez-Vilalta, J., Piñol, J., & Beven, K. (2002). A hydraulic model to predict drought-induced mortality in woody plants: an application to climate change in the Mediterranean. *Ecological Modelling*, *155*(2-3), 127-147

13. McDowell, N., Pockman, W. T., Allen, C. D., Breshears, D. D., Cobb, N., Kolb, T., ... & Yepez, E. A. (2008). Mechanisms of plant survival and mortality during drought: why do some plants survive while others succumb to drought?. *New phytologist*, *178*(4), 719-739.

14. Myneni, R., Knyazikhin, Y., Park, T. (2015). MOD15A2H MODIS Leaf Area Index/FPAR 8-Day L4 Global 500m SIN Grid V006. NASA EOSDIS Land Processes DAAC.

15. Otkin, J. A., Svoboda, M., Hunt, E. D., Ford, T. W., Anderson, M. C., Hain, C., & Basara, J. B. (2018). Flash droughts: A review and assessment of the challenges imposed by rapid-onset droughts in the United States. *Bulletin of the American Meteorological Society*, *99*(5), 911-919.

16. Pastorello, G., Trotta, C., Canfora, E., Chu, H., Christianson, D., Cheah, Y. W., ... & Law, B. (2020). The FLUXNET2015 dataset and the ONEFlux processing pipeline for eddy covariance data. *Scientific data*, *7*(1), 1-27.

17. Running, S. W., & Zhao, M. (2015). Daily GPP and annual NPP (MOD17A2/A3) products NASA Earth Observing System MODIS land algorithm. *MOD17 User's Guide*, *2015*, 1-28.

18. Sanaullah, M., Chabbi, A., Rumpel, C., & Kuzyakov, Y. (2012). Carbon allocation in grassland communities under drought stress followed by 14C pulse labeling. *Soil Biology and Biochemistry*, *55*, 132-139.

19. Schmid, H. P. (1994). Source areas for scalars and scalar fluxes. *Boundary-Layer Meteorology*, *67*(3), 293-318.

20. Svoboda, M., LeComte, D., Hayes, M., Heim, R., Gleason, K., Angel, J., ... & Stephens, S. (2002). The drought monitor. *Bulletin of the American Meteorological Society*, *83*(8), 1181-1190.

21. Thornthwaite, C. W., Mather J.R. (1957). Instructions and tables for computing potential evapotranspiration and the water balance. *Publications on Climatology*, *10*, 185-310.

22. Wolf, S., Keenan, T. F., Fisher, J. B., Baldocchi, D. D., Desai, A. R., Richardson, A. D., ... & Van Der Laan-Luijkx, I. T. (2016). Warm spring reduced carbon cycle impact of the 2012 US summer drought. *Proceedings of the National Academy of Sciences*, *113*(21), 5880-5885.

23. Yan, K., Park, T., Yan, G., Chen, C., Yang, B., Liu, Z., ... & Myneni, R. B. (2016). Evaluation of MODIS LAI/FPAR product collection 6. Part 1: Consistency and improvements. *Remote Sensing*, *8*(5), 359.

24. Zeng, X. (2001). Global vegetation root distribution for land modeling. *Journal of Hydrometeorology*, *2*(5), 525-530

---

## Author Comment (AC2)

Responses to Anonymous Reviewer 2 for paper: *Unraveling phenological to extreme drought and implications for water and carbon budgets*

The authors greatly appreciate the reviewer taking time to provide feedback on our manuscript. We think that the comments are helpful in shaping the new analysis and clarifying how important mechanisms driving plant controls of water and carbon movements may be affected during flash drought.

In light of new analysis described in our responses below, we propose a new title: ***Unraveling phenological and stomatal responses to flash drought and implications for water and carbon budgets***

We respond to each of the Reviewer's comments below, which are in **bold and italics**. Author responses are in blue with proposed manuscript changes in **bold**. Responses in *italics* are used to reference exact quotes from the author's responses to Reviewer Comment 1 (RC1). New figures that we propose to include have labels N# throughout this document and the number corresponds to how the new figures were introduced starting with RC1. Revised figures have the same figure number as in the original manuscript. Figures used to support claims in this document that are not going into the revised manuscript do not have Figure numbers but we have included figure captions.

*This study investigates vegetation phenological processes during drought and non-drought periods using a set of DCHM-V and DCHM-PV model simulations. The focus is on three sites in Kansas USA, as they experienced extreme drought and pluvial conditions in recent decades, and also have ground-based and satellite-based observations available to compare with model simulations and study observed processes. The modeling experiments are neatly designed, and the investigation is systematic to study sources of uncertainty in vegetation phenology. I however have a number of comments that need to be addressed – please see below.*

*Main comments*

*1)   This study uses 2012 and 2019 to exemplify the contrast of vegetation phenology between flash drought vs non-drought years. It would help to add a non-flash/conventional drought year to the study, as the vegetation phenology could differ considerably between flash drought and conventional drought. It would be interesting to see how the evolution of plant phenology, water use, and productivity may differ between the two drought cases.*

The authors appreciate this comment and agree with the reviewer that we should highlight the differences between flash drought and "non-flash/conventional drought". To this end, we propose to add 2018 (conventional drought) as a specific case study to compare against 2012 (flash drought) and 2019 (non-drought). Additionally, we focus components of the analysis to consider average conditions across conventional "drought" and "non-drought" over the course of

the period 2006-2019. This major change is described in our response to Major Comment #1 from Reviewer #1.

From response to Review Comment 1 (RC1): *We now present our results using all years of available model output and framing the analyses in terms of flash drought vs "non-flash" drought conditions. We used the United States Drought Monitor (USDM, Svboda et al., 2002) to determine "drought" and "non-drought" years in Kansas. The Central and East Central Kansas climate regions contain the three study sites (see figure from USDM below). From the USDM time series of the climate regions, drought years were determined if an entire year was in drought or if parts of the region reached D2 (Severe Drought) or higher. The years 2006, 2011, 2013, 2014, and 2018 are labeled as drought years for analysis. The years 2007-2010, 2015-2017, and 2019 are labeled as non-drought years. The flash drought year 2012 is kept separate from other drought years in the analyses.*

[Figure]

Figure Caption. Percent land area in U.S. Drought Monitor Categories for two Kansas climate regions that contain our study sites US-KFS, US-KLS, US-Kon.

We will update the methods Section 2.7 beginning at line 243 with the change in bold.

Line 243: "…We highlight results from the three AmeriFluxes sites for 2012 (flash drought), **2018 (drought),** and 2019 (**non-drought)** to draw conclusions about plant response during flash drought **and how they differ from drought and non-drought years.** We also evaluate **model outputs** from 2006-2019 to assess the **differences** between the DCHM-V and DCHM-PV model configurations **during drought and non-drought years compared to a flash drought year. During this time period, we identified drought years as 2006, 2011, 2013, 2014, 2018 and non-drought years as 2007-2010, 2015-2017, 2019 using the USDM for the Central and**

**East Central Kansas climate regions (Svoboda et al., 2002). Drought years were determined by whether parts of the region reached the D2 "Severe Drought" classification or higher. When computing drought and non-drought averages, we use the years listed here. Transpiration is calculated from total root water uptake through the three soil layers and total evaporation is computed from summing evaporation from ground and canopy surfaces allowing us to partition ET into evaporation and transpiration."**

An example of how we compare flash drought to non-flash drought can be seen with updated Figure 7. We replace yearly totals in Figure 7 with monthly averages of accumulated GPP and ET for drought and non-drought years. We also include monthly averages of ET separated as evaporation and transpiration in a new figure.

[Figure]

**Figure 7 (replacement). DCHM-PV 3YR monthly totals of GPP (a,c,e) and ET (b,d,f) for drought and non-drought years compared to 2012 for US-KFS, US-KLS, and US-Kon AmeriFlux sites. Monthly sums are computed from the ensemble means of the 2000 Monte Carlo simulations then averaged across drought or non-drought years. Error bars represent one standard deviation across drought and non-drought years, respectively.**

An example of how we integrate 2018, a non-flash drought, to compare against 2012 and 2019 can be seen in the updated Figure 11.

[Figure]

**Figure 11. Computed growing season water use efficiency (WUE=GPP/ET) from the DCHM-V, DCHM-PV 3YR, and AmeriFlux for (a) 2012 (b) 2018, drought (c) 2019, non-drought at US-KFS. AmeriFlux WUE is computed by converting latent heat into eT by dividing by the coefficient of vaporization.**

*2)    Much of the findings are based on the DCHM-V and DCHM-PV simulations, and are thus subject to the performance of the DCHM and its predictive phenology in simulating observed land surface and vegetation processes. The comparison between the model results and independent observations (e.g., MODIS, AmeriFlux) however shows considerable differences: some of the models vs. observations differences are so substantial that they are much larger than the differences between different model experiments (e.g., Figs.5-9). While these differences could be in part due to the data comparison across different spatial and temporal scales (Section 4.5), they also make one wonder about the performance of the DCHM and its predictive phenology. I suggest the study provides more information/results on the fidelity of the DCHM and its predictive component is simulating basic land surface variables (e.g., soil moisture, evapotranspiration) and observed vegetation phonology related fields (e.g., LAI, FPAR), e.g., in terms of climatology, seasonal cycle and year-to-year variations, to make the model-based findings more convincing. Also see some of my detailed comments below.*

The author's appreciate this reviewer's comment and we agree that more discussion is needed involving the comparison of the results from the DCHM and measurements and observations from AmeriFlux and MODIS.

Major Comment 3 from our responses to RC1, who shared similar feedback, addressed this point.

From our responses to RC1:

*One reason for the discrepancies between modeled output and flux tower data was that plots of daily average rates of GPP and ET had to do with how we were calculating daily averages for the figure. While it made sense to average these variables over the entire 24-hour period for the flux tower data, the model shuts off GPP and evaporation when there is no incoming solar radiation leading to zeroes during half of the day. Thus, we only average GPP and ET over the active period within the model to avoid including the unrealistic zeros. This is how the DCHM model results were previously presented in Lowman and Barros (2016, 2018) and Lowman et al. (2018). Presenting the results differently was a mistake that has now been fixed. Additionally, we were able to find available gap-filled time series for GPP for US-KFS and US-KLS from the AmeriFlux FLUXNET database (Pastorello et al., 2020), allowing us to make comparisons where previous data was missing in the analysis.*

We proposed the following text to be added to the Discussion section of the manuscript in RC1 to discuss the discrepancies.

***In a recent paper, Giardina et al. (2023) argue that observed plant responses to water stress indicate the ability of plants to access deep groundwater and other stores of water that land surface models (LSMs) are not accounting for. The DCHM has similar soil moisture profiles to the NLDAS-2 and Hosseini et al., 2022, who used Noah-MP configurations, for both the 2012 flash drought and the 2018 drought. The DCHM also follows trends similar to AmeriFlux in 2012, but AmeriFlux top layer soil moisture values are much smaller from May to October of 2018, often under 0.1 m^3 m^{-3}, which is below wilting point, during that time (Figure A1). Despite extremely low top layer soil moisture in 2018, AmeriFlux GPP reaches levels above 10 gC m^{-2} d{-1} coinciding with a brief recharge in soil moisture at the end of June. The DCHM estimates of GPP are often less than 50% of AmeriFlux GPP in 2012 and 2018. The model results from the Noah-LSM similarly underestimate GPP and overestimate soil moisture during these drought periods (Hosseini et al. 2022), suggesting that access to deep water reserves that LSMs cannot reproduce could explain these differences (Giardina et al. 2023).***

***Hosseini et al. (2022) compared predicted estimates of GPP to flux tower measurements at US-KFS and US-KON using predictive phenology with Noah-MP, which accounts for carbon reallocation to leaves, stems, roots, and soils. Even while accounting for carbon movement, they found that during June, July, and August they underestimated tower carbon uptake by 100 gC m^{-2} at US-Kon while overestimating by the same amount at US-KFS in April, May, and June (averaged across an 11-year study period encompassing wet and dry periods). The DCHM-PV, which does not account for carbon reallocation, performs similarity to the model employed by Hosseini et al. (2022), suggesting the accounting for carbon allocation cannot explain underestimating GPP in 2012 and 2018. The DCHM-PV compares more favorably against AmeriFlux data during 2012, the flash drought year, at US-KFS and US-KLS as opposed to 2018, a drought year (Figure 8, A11). This suggests that there are missing processes in both the DCHM and the Noah-MP that***

*cannot capture plant water use during drought, and cannot be attributed to carbon allocation.*

*During drought and flash drought, the DCHM-V and DCHM-PV tend to follow similar trajectories. However, when not under water stress, the predictive phenology model predicts higher carbon uptake than the DCHM-V, which aligns more with MODIS in 2019. AmeriFlux estimates of GPP during June and early July of 2012 and 2018 are also above estimates from MODIS. GPP estimates from the flux tower are higher than the DCHM and MODIS, suggesting that plants are able to maintain higher levels of GPP than what can be recreated in land surface models and satellite remote sensing during drought and flash drought. Differences in DCHM-PV and AmeriFlux GPP cannot be attributed to carbon reallocation since the Noah-MP model accounts for carbon reallocation and similarly underestimated GPP compared to flux tower data (Hosseini et al. 2022). A likely hypothesis is that plants have access to deeper water stores than can be accounted for in land surface models, as suggested by Giardina et al. (2023).*

Updated Figure 8 includes averaging DCHM only over daytime and the inclusion of 2018.

[Figure]

**Figure 8 (updated). Time series of gross primary productivity, GPP, at US-KFS for (a) 2012, flash drought, (b) 2018 drought and (c) 2019 a non-drought year. One standard deviation is shown for the DCHM-PV simulations. MODIS GPP are shown as red crosses and AmeriFlux GPP as small dots.**

[Figure]

**Figure A11 (updated). Time series of gross primary productivity, GPP, at US-KLS for (a) 2012, flash drought, (b) 2018 drought and (c) 2019 a non-drought year. One standard deviation is shown for the DCHM-PV simulations. MODIS GPP are shown as red crosses and AmeriFlux GPP as small dots.**

*As further explanation, the flux towers exist within a 4 km by 4 km region defined by the StageIV spatial grid cell used in the DCHM. Flux tower footprints cover areas with length dimensions ranging from a couple hundred meters to a few kilometers (Baldocchi, 2003, Schmid, 1994) making the 4 km grid cell near the maximum range. Subgrid scale heterogeneity can lead to considerable discrepancies between parameterized and actual fluxes (Schmid, 1994). Since the DCHM treats the entire grid cell as a single vegetation type, our results hold some uncertainty as we cannot account for the heterogeneous mix of vegetation and land-use present on the ground (see Figure below). In addition to savanna, there are deciduous forests within this gridcell that could influence tower readings, and that the DCHM does not account for.*

[Figure]

Figure (non included in manuscript). US-KFS AmeriFlux tower site at the center of a 4 km by 4 km grid representing vegetation heterogeneity of the surrounding region.

We will also add to the discussion section 4.5 Limitations

*Line 465: Capturing phenological responses **and subsequent changes to carbon and water fluxes** within a physically based model is not without its limitations.*

*We propose to remove lines 470-480, beginning with "For example…"*

*Line 470: …temporal and spatial scales. **The flux towers exist within a 4 km by 4 km region defined by the StageIV spatial grid cell used in the DCHM. Flux tower spatial extents range from a couple hundred meters to a few kilometers (Baldocci, 2003, Schmid, 1994) making the 4 km grid cell near the maximum range. Subgrid scale heterogeneity can lead to considerable discrepancies between parameterized and actual fluxes (Schmid, 1994). One explanation for why flux tower data differs from model output is that the flux tower estimates incorporate a variety of vegetation types within the fetch contributing to the vertical fluxes, rather than the single vegetation type used within the model. Additionally, the size and orientation of the contributing fetch varies in time depending on measurement height and turbulent fluxes (Chu et al., 2021).***

***Another difference between modeled and flux tower data could be that models may not be able to fully represent how vegetation can maintain ET by accessing groundwater or deep soil moisture, ultimately biasing models towards more severe effects of drought on vegetation (Giardina et al., 2023). Using predictive phenology with NOAM-LM, which can account for carbon reallocation to leaves, stems, roots, and soils, Hoessini et al. (2022), compared predicted estimates to flux tower measurements of GPP. Even while***

*accounting for carbon movement, they found that during June, July, and August they underestimated tower data by 100 gC m^{-2} at US-Kon while overestimating by the same amount at US-KFS in April, May, and June (averaged across an 11-year study period encompassing wet and dry periods). The DCHM-PV, which does not account for carbon reallocation, responds to drought and flash drought differently than what is observed at flux tower sites. It matches better with AmeriFlux data during 2012, the flash drought year, at US-KFS and US-KLS (Figure 8, A11) compared to 2018, a drought year.*

*During drought and flash drought, DCHM-PV values also agree favorably with MODIS and tend to be slightly larger than MODIS during a non-drought year like 2019. During drought and flash drought, the DCHM-V and DCHM-PV tend to follow similar trajectories but in response to little water stress, the predictive phenology model predicts increased carbon uptake compared to the DCHM-V results which align more with MODIS in 2019. Drought levels of AmeriFlux observed GPP during June are above observed non-drought levels. Even during flash drought, GPP tended to be slightly higher than non-drought June levels. This suggests that during drought and flash drought, plants are able to maintain higher levels of GPP. Differences in DCHM-PV and AmeriFlux GPP are less likely to be attributed to carbon reallocation since the model used by Hosseini et al. (2022) accounted for carbon reallocation and still underestimated AmeriFlux.*

*Detailed comments*

*1)   It would help to briefly discuss the implications of the findings (e.g., based on WET vs DRY vs 3YR) to subseasonal prediction of vegetation.*

Our original intention was to simulate different plant isohydric and anisohydric tendencies following Lowman and Barros (2018) who showed that the data assimilation period can be used to generate phenology model parameters that represent different water use strategies. Following this logic led us to test parameters using WET, DRY, or mixed conditions (3YR) to simulate anisohydric vs isohydric tendencies among the different plants. Our results show that the data assimilation period may not be the only factor to consider when trying to simulate water use strategies. The DCHM predicts stomatal conductance depending on vapor pressure deficit (VPD), light exposure, and soil moisture. High temperatures and low relative humidity lead to increases in VPD. In the model, high VPD leads to very low (or zero) stomatal conductance. (Figure N9). With little water available and high VPD, the DCHM-V and the DCHM-PV follow very closely. The DCHM-PV predicts higher stomatal conductance than the DCHM-V when ample water is available and there are lower values of VPD (Figure N9). This translates to higher GPP predictions in non-drought years (Figure 8).

We suggest the following updates to the discussion section 4.1 Vegetation Responses to Flash Drought.

**While phenology is an important component to consider when computing changes to transpiration and carbon uptake (Lowman and Barros, 2018; Flack-Prain et al., 2019), our results indicate that stomatal conductance is also critical for accurately representing**

these fluxes. Plants adaptively regulate their stomata during periods of water stress (Guo et al., 2020), and some have been demonstrated to maintain open stomata or even increase stomatal conductance under high VPD conditions (Urban et al., 2017). Stomatal conductance shuts down under high VPD in the DCHM (Figure N9), which does not account for the possibility of an adaptive stomatal regulation strategy. Since GPP is directly dependent on stomatal conductance (Farquhar and Sharkey, 1982), DCHM estimates of sub-daily GPP decrease in response to elevated VPD (Figure N11). Moreover, changes in phenological growth state (i.e. LAI) occur across longer (i.e. seasonal) time scales (Katul et al., 2001) than stomatal regulation, which controls carbon and water exchange at sub-daily timescales (Guo et al., 2020). The differences between modeled and observed GPP and ET suggest that there are mechanisms controlling plant responses to drought stress not accounted for within the DCHM. For example, the DCHM could be too strict in representing the sensitivity of stomatal closure to elevated VPD for the Kansas study sites. There could be plant or climate specific VPD dependence (Grossiord et al., 2020), plants could have access to stores of water not accounted for (Giardina et al., 2023), or both.

Guo et al. (2020) showed that isohydricity (i.e. stomatal regulation) exists on a spectrum and that some plants are able to move along that spectrum at sub-daily time-scales with varying environmental conditions, such as higher VPD. Given the high VPD in 2012 at our test sites (Figures N4, N13), we expect the DCHM to estimate low stomatal conductance, and thus low GPP relative to AmeriFlux observations when under atmospheric water stress. We also highlight that the VPD estimated by the DCHM using the NLDAS-2 Forcing File A atmospheric variables is higher during 2012 and 2018 and lower in 2019 compared to AmeriFlux (Figure N13), explaining in part the discrepancies between model and AmeriFlux GPP. As stomatal response to increasing VPD is more complex than how it is represented in LSMs, we agree with Grossiord et al. (2020) who suggest that future modeling studies should focus on how rising VPD drives stomatal closure across different plant functional types.

Daily GPP from the DCHM tends to match the magnitude of AmeriFlux daily GPP at US-KFS in 2012 (flash drought) throughout much of the growing season while greatly underestimating June and July observations in 2018 (drought). The larger discrepancies are also apparent in hourly estimates of GPP (Figure N13). The DCHM halts midday GPP in July 2018, but AmeriFlux values remain high. The differences are smaller in 2012, where AmeriFlux observed carbon assimilation rates of 1 gC m$^{-2}$ s$^{-1}$ throughout the daytime and the DCHM shut down carbon assimilation due to elevated VPD. This again points to the ability for vegetation to access water in ways that current LSMs cannot account for (Giardina et al. 2023). If plants have access to deeper water or are able to tap into stores of water not currently accounted for, they may be able continue (at least temporarily) exchanging water and carbon despite lower precipitation or increased VPD. As stomata control the movement of water and carbon, affecting GPP and water use efficiency (Lawson and Vialet-Chabrand, 2019), accounting for plant adaptations that adaptively regulate stomatal sensitivity to drought stress may improve model accuracy.

[Figure]

**Figure N9 (Adapted). Hourly stomatal conductances [mm s^{-1}] for one week in May, and July of 2012, 2018, and 2019 compared with vapor pressure deficit (VPD, kPa) for US-KFS.**

[Figure]

**Figure N11 (Adapted). Hourly gross primary productivity [g C m^{-2} s^{-1}] from the DCHM-V and DCHM-PV shown against AmeriFlux 30-minute estimates for one week in May, July, and August of 2012, 2018, and 2019 atUS-KFS.**

[Figure]

**Figure N4 (appendix addition). Monthly average vapor pressure deficit [kPa] for the three AmeriFlux sites from April - October for the flash drought year 2012 (black), drought years (red), and non-drought years (blue). The error bar represents one standard deviation across drought and non-drought years.**

[Figure]

**Figure N13 (Appendix) Daily vapor pressure deficit at US-KFS for (a) 2012 - flash drought, (b) 2018 - drought and (c) 2019 - non-drought. The DCHM computes VPD using air temperature and vapor pressure from NLDAS-2 Forcing File A.**

*2)   Noah-LSM: Noah LSM has multiple versions. If the Noah-LSM used in this study refers to the Noah in NLDAS-2, please specify.*

The reviewer's point is well-taken. The Noah-LSM in this study does refer to the Noah model employed in NLDAS-2 (Xia et al., 2012). We will update the soil moisture figure captions and any references to NLDAS-2 soil moisture computed using Noah-LSM in the main body of the manuscript and in the Appendix. We also propose to combine figures from the Appendix so that Figure A1 and A2 become A1 a,b,c to represent the top layer soil moisture at US-KFS for 2012, 2018, and 2019.

[Figure]

**Figure A1** (new and combined with A2**). Top layer soil moisture at US-KFS for (a) 2012, flash drought, (b) 2018 drought and (c) 2019 a non-drought year using the DCHM-V (black dotted line), the DCHM-PV with two standard deviations (red), AmeriFlux (blue dashed line), NLDAS-2 derived from Noah-LSM (yellow) and Stage IV precipitation on the top and right axes (blue).**

*3)  line 259: of gamma => of the growth rate parameter*

This comment is well taken. Following a similar comment from Review 1, we now use gamma once it is defined throughout the remainder of the manuscript rather than going back and forth between gamma and the growth rate parameter.

*4)  Figure 12: May want to increase the thickness of curves for 2012 and 2019 to highlight the results for these two years*

This comment is well taken. We have updated many figures to use thicker lines, varied color, dashed lines, and new marker shapes to help distinguish between simulations/years. Figure 12 has been completely reformatted so the flash drought can be compared to other drought and non-drought periods, as opposed to solely 2019.

[Figure]

**Figure 12 (replacement). Modeled growing season monthly averages of transpiration as a fraction of ET (a-c) and water use efficiency (WUE, d-f) for drought and non-drought years compared with the flash drought year of 2012 for US-KFS, US-KLS, and US-Kon AmeriFlux sites. Monthly averages are computed from the ensemble means of the 2000 Monte Carlo simulations then averaged across drought or non-drought years. Error bars represent one standard deviation across drought and non-drought years. Drought years are 2006, 2011, 2013, 2014, 2018 and non-drought years are 2007-2010, 2015-2017, 2019.**

*5) Line 390: (Figure 10 => (Figure 10)*

This review comment is well taken, and we will implement this change.

*6) Figures A3, A5. Middle and deep layer soil moisture for the flash drought year 2012. How to explain the substantial differences between DCHM-V/DCHM-PV and Noah-LSM? Noah-LSM seems to make more sense as it shows a notable decline after June 2012. In contrast, the soil moisture in DCHM-V/DCHM-PV remains relatively steady throughout 2012 and does not seem to be responsive to the strong precipitation deficits during 2012, which looks odd; this is concerning as any issues in simulating soil moisture would adversely impact the simulation of vegetation and evapotranspiration processes etc. Please also see my second main comment.*

The reviewer's comment is well taken. First, see updates to Figure A3, which will now be A2 and combine 2012, 2018, and 2019 middle layer soil moisture for US-KFS. We respond below by (1) explaining why we see differences between the DCHM and NLDAS-2 soil moisture, and (2) by describing how these differences impact estimates of carbon uptake (GPP) and transpiration (T). We investigate soil moisture, GPP, and T by comparing our results to another modeling study who investigated US-KFS and US-Kon during 2012 and 2018 (Hosseini et al. 2022).

**NOTE: We cannot reproduce the figures referenced from Hosseini et al. (2022) here. Instead, we reference specific figures and panels from their paper for comparison. In reference to soil moisture, see the bottom four panels of Figure 6 in Hosseini et al. (2022). In reference to GPP, see bottom panels of Figure 3 in Hosseini et al. (2022). In reference to transpiration, see the third panel of Figure 5 in Hosseini et al. (2022). In reference to LAI, see the top panels of Figure 6 in Hosseini et al. (2022).

  (1) Why we see differences

**NOTE: In the following paragraphs we compare DCHM soil moisture from different layers to other products (SMERGE, NLDAS-2) and model outputs Noah-MP (Hosseini et al., 2022). Layer depths do not directly compare so for reference, we briefly state the various depths used.

The DCHM top layer soil moisture is an average over 0-8 cm, the middle layer is 8-89 cm, and the deep layer is 89-183). Depths were determined from the Kansas Soil Survey (Soil Survey Staff).  In Hosseini et al. (2022) the top layer in Noah-MP is 0-10 cm and the deep layer average soil moisture they present comes from three layers with thicknesses of 30, 60, and 100 cm. Effectively, this is an average over 10-200 cm vs the DCHM which ranges from 8-183 cm . We average the DCHM middle and deep layers for comparison (see Figure below) and convert Noah-MP estimates into volumetric soil water content for comparison. The NLDAS-2 soil moisture depths used for comparison are 0-10 cm, 0-100 cm, 100-200 cm (Xia et al. 2012) to compare against the DCHM top, middle, and deep layers, respectively. In figures of the middle layer soil moisture, we include comparisons to SMERGE 0-40 cm, computed from "merging" NLDAS and the European Space Agency satellite soil moisture (Tobin et al. 2019).

A first explanation for why we see differences between DCHM modeled soil moisture and NLDAS-2 is that NLDAS-2 soil moisture was estimated from the Noah-LSM without predictive phenology (Xia et al., 2012). However, Hosseini et al. (2022) used various Noah-MP configurations (including with and without predictive phenology) to compute soil moisture, and the DCHM results match well with their soil moisture at US-Kon in 2012 and 2018 (see Figure 6 in Hosseini et al. 2022). Converting units from mm to m^3m^{-3}, we see that Noah-MP predicts a drop in 2012 soil moisture at US-Kon from ~0.35 to 0.28 m^3 m^{-3} from January to September while the DCHM sees a drop of about from ~0.36 to 0.30 m^3 m^{-3}. The Noah-MP model configuration that uses dynamic LAI and vegetation fraction (V3-LD-FD) predicts soil moisture decay from June-September that shows the least steep decline in soil moisture from late June to late August (Hosseini et al. 2022, Figure 6 bottom panel), aligning with results from the DCHM-V and -PV (Figure N14, N15 and additional figure below).

[Figure]

Figure (averaging outputs from N14 and N15). DCHM-PV 3YR volumetric soil moisture averaged across middle and deep layers for US-Kon in 2012, 2018.

It is also important to note that Hosseini et al. (2022) estimates of the top 10 cm of soil moisture match well the magnitude of flux tower soil moisture, fluctuating between ~0.15-0.3 m^3 m^{-3}, between May and July. These findings agree favorably with DCHM top layer soil moisture in 2012 (Figure N14). However, like the DCHM, all model configurations of Noah-MP in Hosseini et al. (2022) estimate lower soil moisture compared to field measurements in the top layer from mid-February to early May 2012 and higher soil moisture from early May through the rest of the year except for some spikes preceding larger rainfall events. Similarly for top layer soil moisture results from 2018, all of the Hosseini models and the DCHM overestimate soil moisture compared to field observations starting in late April and throughout the end of the year (Figure N14a). Thus, the DCHM model results for soil moisture in 2012 and 2018 at KON are in line with what has previously been estimated from different configurations of the Noah-MP that use

predictive phenology and differ similarly from the NLDAS-2 dataset and field observations of soil moisture.

A second explanation of the DCHM estimating higher soil moisture than NLDAS-2 might have to do with cascading effects high VPD has on stomatal conductance. In response to high VPD in the DCHM, stomatal conductance shuts down (Figure N9). Therefore plants are not transpiring. Reduced transpiration is directly tied to reduced root water uptake, resulting in the soils retaining comparatively higher levels of moisture. Figure A7 shows that modeled middle and deep layer root water uptake decreases ~50% from May to July 2012 at US-KFS. Within the DCHM, reduced root water uptake (Figure A7) is likely why estimates of soil moisture in the middle and deep layers remain higher compared to SMERGE and NLDAS-2 (using Noah-LSM) soil moisture (Figures A2 and A3) at US-Kon. However, the DCHM and SMERGE agree favorably in 2012 and 2018 throughout most of the growing season at US-KFS. Note that the DCHM matches well middle and deep layer estimates of soil moisture from NLDAS-2 and SMERGE in 2019 when there is ample water available for plant use within the DCHM.

    (2) How these differences impact estimates of GPP and transpiration

GPP

The DCHM estimates low GPP and stomatal conductance rates during the flash drought period in 2012, while eddy covariance data recorded elevated rates of GPP (e.g., Figure 8, N9, N11). The low estimates of GPP and stomatal conductance from the DCHM are directly related to high atmospheric aridity (or VPD) indicating that the DCHM slows carbon and water exchanges under atmospheric water stress, despite sufficient soil moisture to undergo photosynthesis.

Hosseini et al. (2022) report 11-year (2008-2018) averages of GPP for US-Kon and US-KFS using different Noah-MP configurations, MODIS and AmeriFlux data (Figure 3 in Hosseini et al. 2022). In the figure below, we show the same 11-year averages computed from the DCHM-PV. Noah-MP using predictive LAI configurations estimates higher GPP in April (~150-200 gC m^{-2}) and May (~300 gC m^{-2}), than the DCHM by ~100 gC m^{-2} for similar soil moisture during this time (see Figure 6 in Hosseini et al. 2022 and Figure A2 and N14 below). Both Noah-MP and the DCHM GPP peak in June and the Noah-MP results fall within one standard deviation of the DCHM in June and July at both sites. However, the DCHM 11-year averages of GPP match well the Apr-Oct averages from flux towers for KFS. Noah-MP includes routines for reallocating carbon to different parts of plants (i.e. stems, roots, etc.) that may account for the higher estimates of GPP compared to the DCHM, which does not include such processes.

[Figure]

[Figure]

Figure. Monthly GPP averages across the same 11-year period (2008-2018) as Hosseini et al. (2022) using ensemble mean estimates from the DCHM-PV 3YR. Error bars represent one standard deviation from the 11-year average.

Transpiration

The maximum daily transpiration rate estimated from the DCHM, which computes transpiration from root water uptake, is 1.25 mm d$^{-1}$ in 2012 and 2018 (Figure below), but the Noah-MP modeled transpiration reach over 2mm d$^{-1}$ in May and June for both 2012 and 2018. July - September rates of transpiration for both the DCHM and Noah-MP (with dynamic LAI) fall to less than 0.5 mm d$^{-1}$. Peak transpiration in May and June of 2012 before a decrease to lower transpiration rates in July-October is observed in both the DCHM and Noah-MP (see the third panel of Figure 5 Hosseini et al., 2022) although there are differences in magnitude of transpiration, some of which can be attributed to the differences in computed LAI. Like Hosseini et al., (2022), the DCHM estimates two seasonal peaks of transpiration in June and September of 2018. The late season peak seems to align with large increases in late season precipitation.

Some of the discrepancies in transpiration may result from differences in estimated LAI from both models. The DCHM-PV estimates of LAI tend to agree favorably with the timing of green up and seasonal changes compared to MODIS (see RC1 for full Figure 6 showing LAI at all three sites from 2012, 2018, 2019).  At US-Kon, the DCHM-PV 3YR shows April LAI less than 1 m^2 m$^{-2}$ (see our Figure 6g below,), but Hosseini et al., (2022) estimates leaf out earlier and with April LAI at ~2.7 m^2 m$^{-2}$ (see top panels of their Figure 6). The uptick in transpiration seen by Hoesseini in September 2012 might also be due to the increase in LAI from 0.2 to 2.0 m^2 m$^{-2}$ that they found at the same time. Meanwhile, the uptick in LAI seen by the DCHM-PV was from 1.0 to 1.2 m^2 m$^{-2}$.

Overestimating LAI leads to overestimating latent heat fluxes, as transpiration is a component of latent heat. DCHM estimates of latent heat in May and June of 2012 are less than that of flux tower by ~100 W m$^{-2}$ and match tower measurements well when during wet periods, like 2019 (Figure below). In Noah-MP (Niu et al, 2011; Ma et al., 2017, Li et al., 2021) and in the DCHM, transpiration is directly related to root water uptake which depends on canopy (and stomatal) conductance and both models compute canopy conductance using LAI. Soil moisture across the two models was similar, but LAI varied by over 1 m^2 m$^{-2}$ during the growing

season. Thus, LAI and not differences in soil moisture are likely responsible for differences in modeled GPP and transpiration. .

[Figure]

Figure. Daily transpiration averaged over daytime.

[Figure]

**Cropped from Figure 6. Time series of leaf area index (LAI) predicted from DCHM-PV for the flash drought year (2012), a drought year (2018), and a non-drought year (2019). Colors indicate the different data assimilation periods (yellow - 3YR (2003-2005), blue - WET (2005), red - DRY (2003)), with corresponding shaded regions representing one standard deviation of model outputs from the 2000 ensemble simulations. The 8-day MODIS MOD15A2H LAI is shown in black markers. The gray shaded region highlights the June to July decrease in FPAR during the 2012 flash drought.**

[Figure]

Figure. DCHM estimates of latent heat at US-Kon for 2012, 2018, 2019 compared with AmeriFlux.

[Figure]

**Figure A2** (newly created to combine A3 and A4 and adding 2018). **Middle layer soil moisture at US-KFS for (a) 2012, flash drought, (b) 2018 drought and (c) 2019 a non-drought year using the DCHM-V (black dotted line), the DCHM-PV with two standard deviations (red), SMERGE (green dashed line), NLDAS-2 derived from Noah-LSM (yellow) and Stage IV precipitation on the top and right axes (blue).**

[Figure]

**Figure A7 (Replacing A7 and A8). DCHM-PV 3YR monthly root water uptake totals for drought (red) and non-drought (blue) years compared to 2012 (black) across three soil layers for our three study sites. Monthly sums are computed from the ensemble means of the 2000 Monte Carlo simulations then averaged across drought or non-drought years. Error bars represent one standard deviation across drought and non-drought years, respectively. Drought years are 2006, 2011, 2013, 2014, 2018 and non-drought years are 2007-2010, 2015-2017, 2019.**

[Figure]

**Figure (N14 - appendix). Top layer soil moisture at US-Kon for (a) 2012, flash drought, (b) 2018 drought and (c) 2019 a non-drought year using the DCHM-V (black dotted line), the DCHM-PV with two standard deviations (red), AmeriFlux (blue dashed line), NLDAS-2 derived from Noah-LSM (yellow) and Stage IV precipitation on the top and right axes (blue).**

[Figure]

**Figure N15** (appendix). **Middle layer soil moisture at US-Kon for (a) 2012, flash drought, (b) 2018 drought and (c) 2019 a non-drought year using the DCHM-V (black dotted line), the DCHM-PV with two standard deviations (red), SMERGE (green-dashed line), NLDAS-2 derived from Noah-LSM (yellow) and Stage IV precipitation on the top and right axes (blue).**

*7)    Figure A6 is identical to Figure A5 and appears to be incorrect. Please check if it plots the results for 2019.*

The author's appreciate the reviewer pointing this out. We have fixed this mistake and combined into one figure while adding 2018. This mistake also happened with A1 and A2 (see combination above). We can make similar combinations of soil moisture plots for other sites and layers to add to the appendix.

[Figure]

**Figure A3** (new and result of combining A5 and A6 with results from 2018**). Deep layer soil moisture at US-KFS for (a) 2012, flash drought, (b) 2018 drought and (c) 2019 a non-drought year using the DCHM-V (black dotted line), the DCHM-PV with two standard deviations (red), NLDAS-2 derived from Noah-LSM (yellow) and Stage IV precipitation on the top and right axes (blue).**

*8) Figure A10: "during 2012"=>"during 2019'?*

The authors thank the reviewer for pointing out this error. We will update the figure caption accordingly. We also propose to provide updated figures with the DCHM averaged over only the daytime hours as mentioned above in response to Major Comment 2. We update the color scheme to be monochromatic grayscale to be more vision friendly. It should be noted that the WET and DRY were identical to the 3YR. This was a bug in the plotting code that we fixed.

An example of one of the new figures is below. With the addition of a new figure, A10 might not be the label in the revised manuscript.

[Figure]

**Figure A10 (replacement). MODIS (MOD17A2H) vs DCHM-PV 3YR, WET, and DRY for all three sites during 2019.**

*9) Figure A11a: The difference between Ameriflux and model simulation is striking. The inclusion of Ameriflux appears to cause confusion rather than providing a truthful evaluation of the model results.*

The authors appreciate this comment from the reviewer. The data discrepancies were striking and were the result of an error made when plotting. See Response to Major Comment 3a from the responses to RC1 and Response to Major Comment 2 above.

With updates to how we compute daily averages from model GPP and the use of AmeriFlux FLUXNET, we see that model and AmeriFlux are in better alignment. There is still a striking difference in June and July of 2018 (newly added drought year) that suggests during drought there may be something plants are doing below ground to maintain higher rates of GPP that the DCHM is not capturing. We feel that the use of AmeriFlux FLUXNET in updated figures (including Figure A11 above) are now more useful in evaluating model performance.
* * *
Closing remarks

The authors would like to express our gratitude for the thoughtful comments and that our replies provide clearer and deeper analysis of evaluating the role of vegetation of the movement of water and carbon during flash drought. We understand that should this manuscript be accepted for publication, that there are several new passages and figures (both here and in our response to Reviewer Comment 1) that will need to be included (or removed) and that other changes to enhance cohesiveness of the manuscript in light of the new analysis will need to be incorporated.
* * *
References

1. Baldocchi, D. D. (2003). Assessing the eddy covariance technique for evaluating carbon dioxide exchange rates of ecosystems: past, present and future. *Global change biology*, *9*(4), 479-492.
2. Chu, H., Luo, X., Ouyang, Z., Chan, W. S., Dengel, S., Biraud, S. C., ... & Zona, D. (2021). Representativeness of Eddy-Covariance flux footprints for areas surrounding AmeriFlux sites. *Agricultural and Forest Meteorology*, *301*, 108350.
3. Farquhar, G. D., & Sharkey, T. D. (1982). Stomatal conductance and photosynthesis. *Annual review of plant physiology*, *33*(1), 317-345.
4. Flack-Prain, S., Meir, P., Malhi, Y., Smallman, T. L., & Williams, M. (2019). The importance of physiological, structural and trait responses to drought stress in driving spatial and temporal variation in GPP across Amazon forests. *Biogeosciences*, *16*(22), 4463-4484.
5. Garcia-Quijano, J. F., & Barros, A. P. (2005). Incorporating canopy physiology into a hydrological model: photosynthesis, dynamic respiration, and stomatal sensitivity. *Ecological Modelling*, *185*(1), 29-49.
6. Giardina, F., Gentine, P., Konings, A. G., Seneviratne, S. I., & Stocker, B. D. (2023). Diagnosing evapotranspiration responses to water deficit across biomes using deep learning. *New Phytologist*.
7. Grossiord, C., Buckley, T. N., Cernusak, L. A., Novick, K. A., Poulter, B., Siegwolf, R. T., ... & McDowell, N. G. (2020). Plant responses to rising vapor pressure deficit. *New Phytologist*, *226*(6), 1550-1566.

8.  Guo, J. S., Hultine, K. R., Koch, G. W., Kropp, H., & Ogle, K. (2020). Temporal shifts in iso/anisohydry revealed from daily observations of plant water potential in a dominant desert shrub. *New Phytologist*, *225*(2), 713-726.

9.  Hosseini, A., Mocko, D. M., Brunsell, N. A., Kumar, S. V., Mahanama, S., Arsenault, K., & Roundy, J. K. (2022). Understanding the impact of vegetation dynamics on the water cycle in the Noah-MP model. *Frontiers in Water*, *4*, 925852.

10. Katul, G., Lai, C. T., Schäfer, K., Vidakovic, B., Albertson, J., Ellsworth, D., & Oren, R. (2001). Multiscale analysis of vegetation surface fluxes: from seconds to years. *Advances in Water Resources*, *24*(9-10), 1119-1132.

11. Lawson, T., & Vialet-Chabrand, S. (2019). Speedy stomata, photosynthesis and plant water use efficiency. *New Phytologist*, *221*(1), 93-98.

12. Li, L., Yang, Z. L., Matheny, A. M., Zheng, H., Swenson, S. C., Lawrence, D. M., ... & Leung, L. R. (2021). Representation of plant hydraulics in the Noah-MP land surface model: Model development and multiscale evaluation. *Journal of Advances in Modeling Earth Systems*, *13*(4), e2020MS002214.

13. Lowman, L. E., & Barros, A. P. (2016). Interplay of drought and tropical cyclone activity in SE US gross primary productivity. *Journal of Geophysical Research: Biogeosciences*, *121*(6), 1540-1567.

14. Lowman, L. E., & Barros, A. P. (2018). Predicting canopy biophysical properties and sensitivity of plant carbon uptake to water limitations with a coupled eco-hydrological framework. *Ecological Modelling*, *372*, 33-52.

15. Lowman, L. E., Wei, T. M., & Barros, A. P. (2018). Rainfall variability, wetland persistence, and water–carbon cycle coupling in the Upper Zambezi river basin in Southern Africa. *Remote Sensing*, *10*(5), 692.

16. Ma, N., Niu, G. Y., Xia, Y., Cai, X., Zhang, Y., Ma, Y., & Fang, Y. (2017). A systematic evaluation of Noah-MP in simulating land-atmosphere energy, water, and carbon exchanges over the continental United States. *Journal of Geophysical Research: Atmospheres*, *122*(22), 12-245.

17. Niu, G. Y., Yang, Z. L., Mitchell, K. E., Chen, F., Ek, M. B., Barlage, M., ... & Xia, Y. (2011). The community Noah land surface model with multiparameterization options (Noah-MP): 1. Model description and evaluation with local-scale measurements. *Journal of Geophysical Research: Atmospheres*, *116*(D12).

18. Pastorello, G., Trotta, C., Canfora, E., Chu, H., Christianson, D., Cheah, Y. W., ... & Law, B. (2020). The FLUXNET2015 dataset and the ONEFlux processing pipeline for eddy covariance data. *Scientific data*, *7*(1), 1-27.

19. Schmid, H. P. (1994). Source areas for scalars and scalar fluxes. *Boundary-Layer Meteorology*, *67*(3), 293-318.

20. Soil Survey Staff, Natural Resources Conservation nService, USDA: Web Soil Survey. Available online. Accessed 07 December 2022, https://www.nrcs.usda.gove/resources/data-and-reports/web-soil-survey.

21. Svoboda, M., LeComte, D., Hayes, M., Heim, R., Gleason, K., Angel, J., ... & Stephens, S. (2002). The drought monitor. *Bulletin of the American Meteorological Society*, *83*(8), 1181-1190.

22. Tobin, K. J., Crow, W. T., Dong, J., & Bennett, M. E. (2019). Validation of a new root-zone soil moisture product: Soil MERGE. *IEEE Journal of Selected Topics in Applied Earth Observations and Remote Sensing*, *12*(9), 3351-3365.
23. Urban, J., Ingwers, M., McGuire, M. A., & Teskey, R. O. (2017). Stomatal conductance increases with rising temperature. *Plant signaling & behavior*, *12*(8), e1356534.
24. Xia, Y., Mitchell, K., Ek, M., Cosgrove, B., Sheffield, J., Luo, L., ... & Lohmann, D. (2012). Continental-scale water and energy flux analysis and validation for North American Land Data Assimilation System project phase 2 (NLDAS-2): 2. Validation of model-simulated streamflow. *Journal of Geophysical Research: Atmospheres*, *117*(D3).

---

## Author Response (AR1)

**Combined Reviewer 1 and 2 Comments and Corresponding Author Responses**

The authors would like to express our gratitude to Reviewer 1 and 2 for the thoughtful comments and hope that our replies provide clearer and deeper analysis of evaluating the role of vegetation in altering water and carbon fluxes during flash drought.

The following responses are updated versions of the initial author response to reviewer comments from HESS Discussions. These updated responses include implementation of language into the final manuscript reflecting the major revisions and reframing. We significantly edited the writing throughout the manuscript to improve the flow and readability. We respond to each of the Reviewer's comments below, which are in **bold and italics**. Author responses are in blue with manuscript changes in **bold**.

Major changes include:

1.  Updated Analysis of Model Results:  Model outputs of infiltration, vapor pressure deficit, and stomatal conductance are now included in the analysis of the manuscript. The new analysis resulted in a title change (see below) and the addition of figures in the main manuscript and supplemental material.
2.  New Title: *Unraveling phenological and stomatal responses to flash drought and implications for water and carbon budgets*
3.  More relevant hypotheses: We updated the hypotheses to reflect comments from both Reviewers (see responses below for more details)
4.  Updated Figures: Color schemes and formatting have been updated to reflect the journal standards and saved as high quality vectorized images. Figures numbers in this document reflect how they appear in the update manuscript or supplemental figures document. Any figures included herein that are used only for justification of responses to reviewer comments and do not appear in the manuscript or supplemental material are not labeled but are given a caption.
* * *
**Detailed author responses to reviewer comments.**

Responses to Anonymous Reviewer 1 for paper: *Unraveling phenological to extreme drought and implications for water and carbon budgets*

The authors would like to thank the reviewer for taking the time to make thoughtful comments that will improve the manuscript. The questions, insights, and suggestions will help us to clarify, shape, and focus the readers on the main point of the article: that land-atmosphere interactions undergo rapid changes during flash drought not observed during drought or non-drought periods.

Given the updated analysis and comparison of flash drought to non-flash drought periods, the authors use the following language throughout this document and will update accordingly in the revised manuscript. We use "flash drought" when referring to 2012 and "drought" or "drought

years" for all other droughts. We use "non-drought" for all years that are not drought or flash drought.

The authors noticed the following major themes from the Reviewer comments.

1. A need to clarify differences between flash drought and drought
   a. Hypotheses need to better address what distinguishes flash drought from drought (i.e., timing, magnitude of effects, etc.)
   b. Strengthen analyses by using all available years from model outputs when comparing results from the flash drought year against 'non-flash' drought years
2. Some claims could be better supported with figures using model outputs that were not previously used, specifically:
   a. Including results from infiltration
   b. Exploring the relationship between phenology, GPP, stomatal conductance, and transpiration
3. Differences between model predictions, flux tower measurements, and MODIS observations need more discussion.
   a. Model values of GPP and ET were much lower than AmeriFlux and MODIS GPP during flash drought
   b. Discussion of observed differences should explore plant processes (e.g., reallocating carbon, tapping into deep groundwater, stomatal regulation, etc.) that the DCHM may be missing.

Response to Major Comment 1a:

We updated our hypotheses to distinguish flash drought from other droughts. They include language that allows us to answer more specifically how land-atmosphere interactions differ in a flash drought from other times. The hypotheses now read:

**H1 During flash drought, there is an increase in days between precipitation events leading to larger reductions in total precipitation and infiltration as compared to non-flash drought events.**

**H2 Lower total infiltration and higher atmospheric demand for water observed during flash drought reduces soil water available for root water uptake. This decreases stomatal conductance, subsequently leading to reduced rates of transpiration, carbon uptake, and water-use efficiency as compared to non-flash drought within a subseasonal time frame.**

**H3 In response to decreased water availability during flash drought, vegetation phenological states will be diminished as compared to non-flash drought years exacerbating the reduction of transpiration and carbon uptake.**

Response to Major Comment 1b:

We now present our results using all years of available model output and framing the analyses in terms of flash drought vs "non-flash" drought conditions. We used the United States Drought

Monitor (USDM, Svboda et al., 2002) to determine "drought" and "non-drought" years in Kansas. The Central and East Central Kansas climate regions contain the three study sites (see figure from USDM below). From the USDM time series of the climate regions, drought years were determined if an entire year was in drought or if parts of the region reached D2 (Severe Drought) or higher. The years 2006, 2011, 2013, 2014, and 2018 are labeled as drought years for analysis. The years 2007-2010, 2015-2017, and 2019 are labeled as non-drought years. The flash drought year 2012 is kept separate from other drought years in the analyses. This change will be updated in the Methods section as well (see response to specific comment below).

[Figure]

Figure. Percent land area in the U.S. Drought Monitor Categories for two Kansas climate regions that contain our study sites US-KFS, US-KLS, US-Kon.

Response to Major Comment 2a:

Throughout the paper, including the hypotheses, we made claims about infiltration and its relation to evaporation and root water uptake without showing any infiltration results. Hypothesis 1 has been updated (see above) to incorporate how timing of rainfall events impact total infiltration. To investigate this hypothesis we include plots of monthly accumulations of infiltration, evaporation, and root uptake, and figures relating infiltration to the amount of time between precipitation events to build a better understanding of how infiltration affects available water for plant use. We also address this in the discussion and conclusion. See below for new analyses related to specific comments.

Response to Major Comment 2b:

We take a closer look at sub-daily GPP and stomatal conductance during selected weeks throughout the growing season for a flash drought, drought, and non-drought year in order to evaluate how stomatal conductance reduces with increased vapor pressure deficit (VPD) and leads to subsequent declines in transpiration and GPP. We also investigate whether reductions in stomatal conductance or leaf area index (LAI) have a larger impact on GPP in order to evaluate if there exists a phenological dependence during flash drought. See below for new analyses related to specific comments.

In light of new analyses during the process of responding to the Reviewer comments, we propose a new title: ***Unraveling phenological and stomatal responses to flash drought and implications for water and carbon budgets***

Response to Major Comment 3a:

One reason for the discrepancies between modeled output and flux tower data was that plots of daily average rates of GPP and ET had to do with how we were calculating daily averages for the figure. While it made sense to average these variables over the entire 24-hour period for the flux tower data, the model shuts off GPP and evaporation when there is no incoming solar radiation leading to zeroes during half of the day. Thus, we only average GPP and ET over the active period within the model to avoid including the unrealistic zeros. This is how the DCHM model results were previously presented in Lowman and Barros (2016, 2018) and Lowman et al. (2018). Presenting the results differently was a mistake that has now been fixed. Additionally, we were able to find available gap-filled time series for GPP for US-KFS and US-KLS from the AmeriFlux FLUXNET database (Pastorello et al., 2020), allowing us to make comparisons where previous data was missing in the analysis.

Response to Major Comment 3b:

We propose to add the following text to the Discussion section of the manuscript to address this major comment:

[revised manuscript text omitted]

*This papers offers a detailed analysis of drought responses for vegetation in the Midwestern US using an ecohydrologic model and assimilation of MODIS FPAR and LAI data along with flux tower data.  The paper addresses some important ecohydrologic questions about how rapidly transpiration and carbon assimilation decline during drought and how changing phenology, specifically above-ground photosynthetic capacity, accelerates GPP losses.  The paper utilizes advanced modeling techniques and builds on previous work that has established useful ways to assimilated remote sensing data into the DCHM model.*

*While that paper has significant potential, it did not really make strategic use of the model and observations to address some of the questions posed in the introduction.*

*For example, the hypothesis posed do not really address the issues of 'flash' drought.*

*In H2 The idea that drought causes both carbon uptake and transpiration to decline is something that is quite well understood - and there is ample evidence that this occurs- we know plant shut down when they run out of water.  There are elements of the timing of*

*this that are perhaps less well understood - and questions about how water use efficiency changes during a drought- and indeed the authors get at this to some extent in the paper - The hypothesis should reflect this. There would also be ways to frame the study (and hypothesis) to look at the relative impact of "flash" drought versus other types of drought that could be interesting. Some additional thinking about how to use the model to test more nuanced (and informative) hypothesis would be strengthen this paper*

The Reviewer's comments about H2 and other hypotheses are well taken. We have edited the Hypotheses in the paper to clearly articulate changes we expect to see in land-atmosphere interactions during flash drought that differ from drought (see response to Major Comment 1a above). The following changes were made to H1, H2, and H3.

H1 now addresses differences in timing of precipitation events during flash drought, drought, and non-drought years.

H2 now has language that broadens the types of vegetation responses we see due to decreased infiltration. We also have language to compare vegetation responses across flash, regular, and non-drought periods.

H3 was updated to draw comparisons of flash drought to non-flash drought years and to include more specific language other than "plant-atmospheric interactions".

See hypotheses written above.

We also updated Figure 1 to incorporate the edited hypotheses.

[Figure]

**Figure 1. Schematic of water, carbon, and energy fluxes with hypotheses about ecological response to flash drought indicated with orange arrows. Decreased frequency of precipitation events leads to decreased infiltration and less water available for plant use during flash drought as compared to non-flash drought periods. During flash drought, the cascading effects of decreased water availability, exacerbated by the**

**reduced phenological states and stomatal conductance, include rapid reductions in transpiration and atmospheric carbon uptake to levels below other drought periods.**

*Throughout the paper, there are statements made that are not well supported by graphs or analysis - for example - that evaporation exceeds infiltration (this is indirectly shown but it would be much convincing to show this directly - and model results could do this). In another perhaps more salient example for the paper, the authors state phenology declines reduce carbon and water exchanges (H3) - one could argue that because plants have already shut down stomates at that point in the season, phenological declines do not further reduce transpiration - I'm not suggesting this is true but graphs presented do not clearly rule this out in the testing of H3.*

The Reviewer's comments are well taken. We have combed through the paper to make sure that any claims are fully substantiated and demonstrated with figures or data from our results. We have taken a critical look at unsubstantiated claims throughout the paper as referenced by this review. In doing so, we also noticed that we could better support the updated hypotheses by incorporating additional results and analyses. For changes to hypotheses, see response to Major Comment 1a above.

Regarding the first comment on evaporation exceeding infiltration, we acknowledge that we previously did not show any results for infiltration. In addition to updating the wording of our hypotheses, we will add a subsection to our results section of the manuscript that discusses infiltration.

We propose to add a section in the results **Section 3.2 Sub-surface Water** with subsections for infiltration, soil moisture, and root water uptake.

**Section 3.2.1 - Infiltration**

[revised manuscript text omitted]

Response to second part of comment:

The authors appreciate the reviewer bringing awareness about a potential counterpoint to our claim that *"phenology declines reduce carbon and water exchanges (H3) - one could argue that because plants have already shut down stomates at that point in the season,*

*phenological declines do not further reduce transpiration  - I'm not suggesting this is true but graphs presented do not clearly rule this out in the testing of H3."*

We acknowledge that there may be more at play with phenology, stomatal conductance and plant-atmosphere interactions which has led to new ways of interpreting the model output. Below, is a new proposed section that evaluates the relative contributions of changes in stomatal conductance at the sub-daily scale and a discussion section detailing how stomatal conductance and LAI affect GPP. We also address this with the proposed change in title.

**Section 3.3.1 Sub-daily Stomatal Conductance**

**Figure 9. Stomatal conductance [mm s$^{-1}$] and vapor pressure deficit (VPD, kPa) for one week in May and July of 2012, 2018, and 2019 for US-KFS.**

**Sub-daily estimates of stomatal conductance highlight how VPD can drive stomatal activity within the DCHM. In 2012, stomatal conductance in the first week of May was as high or higher than in 2019, a non-drought year at US-KFS (Figure 9). But by July, major differences in 2012 and 2019 stomatal conductance coincided with changes to VPD. In July 2012, high VPD shuts down midday stomatal conductance whereas lower values of VPD allow for higher rates of stomatal conductance during the same time in 2019. The large reduction in stomatal conductance from the first week of May to the first week of July during the flash drought year of 2012 is unlike that seen in a drought year like 2018 where stomatal conductance rates are similar in May and July.**

**Discussion**

**Section 4.1 Mechanisms Controlling Plant Responses to Drought**

**4.1.1 Stomatal and Non-stomatal Regulation of GPP**

[Figure]

**Figure 14. Stomatal conductance [mm s$^{-1}$] vs leaf area index, LAI [m$^2$ m$^{-2}$] for US-KFS for a flash drought year (2012), a drought year (2018), and a non-drought year (2019). Marker shapes indicate individual days between April 1 - October 31. Each month is given a unique shape whose color reflects daily accumulations of gross primary productivity [gC m$^{-2}$].**

An objective of this work is to evaluate whether changes in phenology versus changes in stomatal conductance have a stronger control on carbon uptake during flash drought (H2, H3). We consider how GPP covaries during flash drought, drought, and non-drought years with sub-seasonal changes in LAI and stomatal conductance at US-KFS (Figure 14). During a non-drought year (2019), there exists a wider range of values of stomatal conductance, LAI, and GPP throughout the growing season (Figure 14c). There is also a clear seasonal cycle in the clockwise movement through the stomatal conductance-LAI parameter space. Stomatal conductance increases faster than LAI in the early season before reaching maximum values around June. After LAI peaks, there is first a reduction in stomatal conductance and GPP at higher LAI before LAI decreases through August and September.

In contrast, during flash drought (2012) and drought (2018), peak stomatal conductance, LAI, and GPP values at US-KFS are approximately half of 2019 values. Both stomatal conductance and LAI remain low throughout the growing season and GPP is below 10 gC m$^{-2}$ at all sites in 2012 (Figure S7). Stomatal conductance and LAI are highest in May 2012 as opposed to June and July 2019. While both 2012 and 2018 have low values of stomatal conductance, LAI, and GPP, an important difference is the near-zero stomatal conductance during June and July 2012 for a range of LAI values (1-2 m$^2$ m$^{-2}$, Figure 14) that is not observed in 2018 and other drought years (Figure S11).

The relationship between stomatal conductance, LAI, and GPP is similar across all three sites when considering flash drought (Figure S7), drought (Figure S11), or non-drought periods (Figures S8,S9,S10). The observable clockwise movement through parameter

space is not as clear in flash drought and drought as compared to non-drought. In drought years, stomatal conductance from April-October averages 1.4 mm s$^{-1}$ across all sites (Figure S11) compared to 2.3 mm s$^{-1}$ in non drought years (Figures S8,S9,S10) and 1.1 mm s$^{-1}$ in flash drought (Figure S7). Peak LAI is approximately 1-2 m$^2$ m$^{-2}$ higher in non-drought years compared to flash drought and other drought years. Similarly, non-drought GPP levels are approximately 6-8 gC m$^{-2}$ higher than flash drought and non-drought periods.

Prior work linked phenological responses to drought to changes in vegetation-atmosphere interactions (Lowman and Barros, 2018, Cui et al., 2017). Dynamically estimated FPAR and LAI tend to exert strong controls on the resulting GPP (Lowman and Barros, 2018). By updating phenological states using the phenology model rather than forcing phenology with remotely sensed values, we were able to capture the plant growth response to water availability. When more water is available, DCHM-PV simulation predicts higher values of FPAR, LAI, and thus higher values of GPP. At the onset of flash drought, DCHM-V and -PV respond faster to changes in LAI and FPAR than MODIS whose effects were also seen in differences in modeled and remotely sensed GPP (Figure11). Moreover, regardless of the simulation, the rapidness of the change in LAI and FPAR is indicative of flash drought (Figures 5 and 6) and in agreement with Zhang et al. (2020). Decreases in phenological state due to the lack of soil water available to plants affected carbon and water exchanges, suggesting support for the third hypothesis (H3), however, decreases in stomatal conductance driven by increased VPD may compound the detrimental phenological effects.

[Figure]

Figure S7. Daily stomatal conductance [mm s$^{-1}$] vs leaf area index, LAI [m$^2$ m$^{-2}$] for all three sites during the flash drought of 2012. Marker shapes indicate individual days from April 1 - October 31 from the selected year. Each month is given a unique shape and daily totals of gross primary productivity [gC m$^{-2}$] are indicated by color.

[Figure]

**Figure S8.** Daily stomatal conductance [mm s$^{-1}$] vs leaf area index, LAI [m$^2$ m$^{-2}$] for US-KFS for selected non-drought years. Marker shapes indicate individual days from April 1 - October 31 from the selected drought year. Each month is given a unique shape and daily totals of gross primary productivity [gC m$^{-2}$] are indicated by color.

**Figure S9.** Daily stomatal conductance [mm s$^{-1}$] vs leaf area index, LAI [m$^2$ m$^{-2}$] for US-KLS for selected non-drought years. Marker shapes indicate individual days from April 1 - October 31 from the selected drought year. Each month is given a unique shape and daily totals of gross primary productivity [gC m$^{-2}$] are indicated by color.

[Figure]

**Figure S10. Daily stomatal conductance [mm s$^{-1}$] vs leaf area index, LAI [m$^2$ m$^{-2}$] for US-Kon for selected non-drought years. Marker shapes indicate individual days from April 1 - October 31 from the selected drought year. Each month is given a unique shape and daily totals of gross primary productivity [gC m$^{-2}$] are indicated by color.**

[Figure]

**Figure S11. Daily stomatal conductance [mm s$^{-1}$] vs leaf area index, LAI [m$^2$ m$^{-2}$] for all three study sites for selected drought years. Marker shapes indicate individual days from April 1 - October 31 from the selected drought year. Each month is given a unique shape and daily totals of gross primary productivity [gC m$^{-2}$] are indicated by color.**

*The author also make statements about flash droughts but do not really distinguish flash drought from other types of drought in their analysis - They have multiple years that they do not really make use of.*

The authors thank the review for this comment. This is one of the major comments we address in Major Comment 1 above and help drive most of the new analysis that better distinguishes flash drought from other drought and non-drought conditions.

As shown above, we are now making direct links to a drought year (2018) as well as averaging across drought and non-drought years. We determined the drought and non-drought years from the USDM (Svoboda et al., 2002). We will update the methods Section 2.7 beginning at line 240 in the original manuscript, now circa line 264

**In this manuscript, we are interested in exploring whether land-surface, subsurface, and atmospheric interactions are distinct in flash drought compared to drought and non-drought periods. We focus on results from the three AmeriFlux sites for 2012 (flash drought), 2018 (drought), and 2019 (non-drought) to draw conclusions about plant response during flash drought and how they differ from drought and non-drought years.**

**We also evaluate model outputs from 2006-2019 to assess the differences between the DCHM-V and DCHM-PV model configurations during drought and non-drought years compared to a flash drought year. During this time period, we identified drought years as 2006, 2011, 2013, 2014, 2018 and non-drought years as 2007-2010, 2015-2017, 2019 using the USDM for the Central and East Central Kansas climate regions (Svoboda et al., 2002). Drought years were determined by whether parts of the region reached the D2 "Severe Drought" classification or higher. When computing drought and non-drought averages, we use the years listed here. In many time series results, we display the water year (April-October) rather than the entire year because plants are largely dormant outside of the water year in a temperate region (Dai et al., 2016; Wang et al., 2003; Towne and Owensby, 1984). Transpiration is calculated from total root water uptake through the three soil layers and total evaporation is computed from summing evaporation from ground and canopy surfaces allowing us to partition ET into evaporation and transpiration (Lowman and Barros, 2018; Lai and Katul, 2000). Water-use efficiency is represented as the ratio of GPP and ET (WUE = GPP/ET, Beer et al., 2009). We highlight differences between the DCHM-V and DCHM-PV model simulations and compare outputs to remotely sensed and in situ observations where available.**

Also to address this comment, we enhance our analysis to highlight that flash drought should be considered as the time period leading up to drought and the intensification rate (Otkin et al., 2018). As such, we will put more emphasis on the change leading from non-drought to drought (May to July of 2012) in results and discussion.

*Finally there are also important differences between modeled and flux tower data that may be critical in the understanding flash drought responses. These differences need to be more rigorously explored (see detailed comments below)*

The reviewer's comment is appreciated. As noted above in response to Major Comment 3, the reason for the discrepancies was an error made when creating the figures that has now been fixed. Please see the updates in Figures 8 (now Figure 11) above in response to Major Comment 3 and updates to Figure 9 (now Figure 13) below.

[Figure]

**Figure 13. Daily evapotranspiration, ET, [mm d$^{-1}$], at US-KFS for (a) 2012 flash drought, (b) 2018 drought and (c) 2019 a non-drought year. Two standard deviations are shown for the DCHM-PV simulations. AmeriFlux ET is derived from latent heat measurements and shown as blue dots.**

One possible explanation for why flux tower data differs from model output is that the flux tower estimates incorporate a variety of vegetation types within the fetch contributing to the vertical fluxes, rather than the single vegetation type used within the model. Additionally, the size and orientation of the contributing fetch varies in time depending on measurement height and turbulent fluxes (Chu et al., 2021). Another difference could be that models may not be able to fully represent how vegetation can maintain ET by accessing groundwater or deep soil moisture, ultimately biasing models towards more severe effects of drought on vegetation (Giardina et al., 2023).

Further, the flux towers exist within a 4 km by 4 km region defined by the StageIV spatial grid cell used in the DCHM. Flux tower spatial extents range from a couple hundred meters to a few kilometers (Baldocci, 2003, Schmid, 1994) making the 4 km grid cell near the maximum range. Subgrid scale heterogeneity can lead to considerable discrepancies between parameterized and actual fluxes (Schmid, 1994). Since the DCHM treats the entire grid cell as a single vegetation type, our results hold some uncertainty as we cannot account for the heterogeneous mix of vegetation and land-use present on the ground (see Figure below). Inside of this grid is deciduous forest that could influence tower readings, that the DCHM does not account for.

[Figure]

Figure (non included in manuscript). US-KFS AmeriFlux tower site at the center of a 4 km by 4 km grid representing vegetation heterogeneity of the surrounding region.

We proposed to split  Section 4.5 Limitations into multiple subsections. Extensive rewriting, and for ease of reading is included in full below.

**Section 4.6 Model Performance and Limitations**

**Section 4.6.1 Model vs Observations**

**This study allows us to investigate how vegetation responses can be used to study the effects of flash droughts on the total carbon and water budgets. Our modeling approach permits direct comparisons of remotely sensed observations to physically derived estimates. Generally, MODIS overestimates GPP compared to EC flux tower data (Heinsch et al., 2006; Running et al., 2004). Daily GPP from the DCHM tends to match the magnitude of MODIS and AmeriFlux GPP at US-KFS throughout much of the growing season but underestimates June and July observations in 2012 (flash drought) and 2018 (drought). The DCHM-PV tends to overestimate during 2019 (non-drought) while the DCHM-V more closely aligns with observations. Large discrepancies are also apparent in hourly estimates of GPP at US-KFS (Figure S22). The DCHM halts midday GPP in July 2018, but AmeriFlux values remain high. The differences are smaller in 2012, where AmeriFlux observed carbon assimilation rates of 1 gC m$^{-2}$ s$^{-1}$ throughout the daytime and the DCHM shut down carbon assimilation due to elevated VPD (Figure S22).**
…The rest of this section is included above in response to Major Comment 3b.

**Section 4.6.2 Implications for Land-surface Models**

Capturing phenological responses and subsequent changes to carbon and water fluxes within a physically based model is not without its limitations. As we update phenological states during the DCHM-PV simulations, forced atmospheric conditions from NLDAS-2 and Stage-IV variables are the same as in the DCHM-V simulations. We continue to use these conditions to force the model, so it is possible that the meteorological observations are already accounting for some vegetation-atmosphere interactions. But, by explicitly considering plant tendencies, we can dynamically account for current meteorological conditions and thus use physical principles to capture vegetation-atmosphere interactions.

Vegetation responses to water stress are apparent through fluctuations in GPP (Zhang and Yuan, 2020; Jin et al., 2019) and ET (Chen et al., 2019). Decreases in GPP occur when plants close their stomata. With the stomata closed, plants will limit gas exchange affecting both photosynthesis and transpiration rates. Transpiration is only one part of ET, so we must be careful not to directly link fluctuations in GPP with fluctuations in ET. Evaporation can still be high when there is little to no transpiration, but GPP tends to follow the same trajectories as transpiration (Figures 10, 12; Beer et al., 2009). In some cases, vegetation can reallocate already processed carbon to their roots when under drought stress mitigating GPP losses (Ingrisch et al., 2020). However, modeled GPP losses are likely a result of modeled stomatal behavior, as the model does not account for reallocation of carbon stores within the plants. Sub-daily scale stomatal conductance reduces to zero in response to increased VPD (Figure 9) leading to similar reductions in modeled GPP (Figure S22). This limitation of the DCHM could explain why AmeriFlux GPP tends to be higher than the modeled GPP.

Vegetation activity is directly linked to the coupling of the water and carbon cycling through photosynthesis (Farquhar et al., 1980) and assimilating plant phenology into land-surface models (e.g., DCHM-V or Noah-MP) can improve estimates of GPP and ET (Hosseini et al., 2022; Xu et al., 2021; Mocko et al., 2021; Kumar et al., 2019). However, our findings also indicate that improved phenology cannot alone account for vegetation adaptations to water stress and ability to access water in ways that current LSMs cannot account for (Giardina et al., 2023). Future studies should focus on improving our understanding how plants are able to tap into different stores of water to continue exchanging water and carbon despite lower precipitation or increased VPD. Additionally, as stomata control the movement of water and carbon, affecting GPP and water-use efficiency (Lawson and Vialet-Chabrand, 2019), accounting for plant adaptations that adaptively regulate stomatal sensitivity to drought stress, especially VPD, may improve model accuracy.

Moving forward, improvements made to phenological states of the entire plants (i.e. root systems included) rather than just the leaf phenology might better capture water movement through plants under water stress conditions. Future studies would benefit from improved estimates of root water uptake since it is directly linked to the amount of

**available water for transpiration. Vegetation types have distinct root characteristics leading to differences in hydraulic tendencies under variable water regimes and atmospheric conditions which distinguish vegetation that is more likely to survive or recover from drought (McDowell et al., 2008; Martinez-Vilalta et al., 2002) . Species specific hydraulic strategies may differ in a single location (Liu et al., 2020) so generalization of water-use by PFT in hydrologic models would represent the average tendency of vegetation to regulate water. It is also possible that the changing phenological state of root systems plays an important role in root water uptake (McCormack et al., 2014). Moreover, models that can account for different vegetation behavior such as the reallocation of carbon storage and below ground respiration during drought may provide a better understanding of mechanisms driving drought resiliency and changes to carbon uptake during drought (Ingirsch et al., 2020; Sanaullah et al., 2012). These types of mechanisms could explain how a warm and wet spring mitigated the effects of the 2012 flash drought on GPP losses (Wolf et al., 2016).**

*Some detailed comments.*

*H1 is actually two hypothesis - it would be useful to separate them*

See response to Major Comment 1 above regarding our updated hypothesis H1.

*The data sets and modeling proposed here tend to focus on relatively shallow surface soil - including citations of expected rooting depths for the PPTs would be helpful support for the implementation (especially given that flux tower observations of ET tend to be higher than the model in dry years)*

The review comments are well taken. We propose two changes to the paper beginning at line 189.

1. For clarity, we remove parenthetical depths in line 189-190 since they are stated more clearly in the following sentence. We also make units in mm instead of inches or cm.
2. We add a reference to expected rooting depths using a combination of soil and PFT.

Line 189, now line 205

**We use 80 mm for the top layer soil depth to ensure model stability, but middle and deep layers were selected to best match the USDA Kansas soil profile (Soil Survey Staff). The this yields three soil layers: top (0-80 mm), middle (80-890 mm) and bottom (890-1830 mm). Rooting depth and density, which are used to determine the total root water uptake in the DCHM, are calculated using empirical exponential root distribution functions that vary by PFT (Lowman and Barros, 2016; Zeng, 2001; Lai and Katul, 2000; Jackson et al., 1996; Clausnitzer and Hopmans, 1994) Soil layer and rooting depths align with the different combinations of soil textures and PFTs found in (Thornthwaite and Mather, 1957).**

*line 225 - Some additional (just one or two sentences) information about of how ensembles of phenology parameters are established is needed here (e.g what is done for each of the 3 periods to select the 2000 parameter distributions shown in Figure 4) - There needs to be a bit more context so that reader understands Figure 4 and what controls the variation in parameter sets.*

We propose to address this comment by rewriting Methods section 2.5 Model Description and breaking it into multiple subsections.

**2.5 Description of Modeling Work**

**2.5.1 Land-Surface Hydrology Modeling**

**2.5.2 Predictive Phenology**

**The DCBP is the predictive phenology model that determines future plant growth based on differences between current and potential phenological states. The growing season index (GSI) determines potential phenological state based on current climate conditions (Stockli et al., 2008; Jolly et al., 2005). Specifically it is a function of temperature, photoperiod, soil water potential, and VPD (Lowman et al., 2023; Lowman and Barros 2016, 2018) adapted the framework to incorporate soil water parameters that affect predictions of plant growth stage. The DCBP is implemented within the DCHM-PV to estimate phenologic state with the the land-surface hydrology model. However, in order to implement the predictive phenology model within the DCHM-PV, we first must estimate parameters that determine plant growth rates and sensitivity to meteorological and soil conditions.**

**A Bayesian hierarchical approach is used to estimate the parameters for the DCBP. Specifically, a dual state-parameter ensemble Kalman filter (EnKF) is used to jointly estimate the phenologic states of FPAR and LAI and the eleven other parameters within the DCBP (Lowman et al., 2023; Lowman and Barros 2018). This method was described by Moradkhani et al., (2005) as a way of simultaneously predicting states and parameters in hydrologic models, and later implemented by Stockli et al. (2008) to assimilate remotely sensed observations of LAI and FPAR into a predictive phenology model.**

**Section 2.6 Model Simulations**

*line 235 - The simulation period is relatively short - and isohydric-anisohydric differences may or not be distinguishable within the 3 years - thus you cannot really state that the vegetation model parameters trained on dry conditions will represent isohydroic vegetation?.  Especially given that parameter values seem to change depending on period (Figure 4) but vegetation PFT does not.*

The authors appreciate the reviewer's comments.

Lowman and Barros (2018) showed that assimilation period can determine the water stress adaptations for the modeled vegetation state. We propose the following edits beginning at line 235.

Line 235: …affects the development of flash drought. **It has been shown under varied climatological conditions plants can be highly adaptable, transitioning from isohydric to anisohydric in a single season (Guo et al., 2020). Lowman and Barros (2018) showed that assimilation period can determine the water stress adaptations for the modeled vegetation state.** Broadly speaking vegetation model parameters trained using **data from years with minimal rainfall represent plants that are accustomed to drier conditions and therefore exhibit more regulation in their water use tendencies (Lowman and Barros, 2018)**.

*line 246 - That transpiration is calculated from root water uptake makes sense but it doesn't follow that this allows you to "to partition ET"…you would need to have a separate calculation of total ET to do that. Clarify*

The reviewer's comment is well taken. The model computes total ET (now Figure 13) from totaling surface evaporation from soil and canopy and from computing transpiration as root water uptake through the three soils layers. We address the partitioning in the addition to the Methods Section 2..

*Line 259 - For clarity it would be helpful to be consistent in the naming conventions- e.g gamma or growth parameter not both*

We now use the gamma symbol once it is defined throughout the remainder of the manuscript.

*Line 258- in what way is this in agreement with Lowman and Barros (e.g the choice of longer period for reducing uncertainty) - in a way that's not so surprising - more information usually reduces uncertainty?*

The authors appreciate the need for clarification. Line 258 now reads: **"... in agreement with Lowman and Barros (2018) who found that using assimilation periods with both wet and dry conditions has the effect of capturing adaptive plant water use strategies."**

*Line 259 That gamma values vary by site could be do to differences in climate (note that game values vary across wet and dry years) - so it is not a given that it varies by plant functional type - rather this is an assumption (e.g I think that you are assigning plant functional type parameters based on this analysis)- The wording of this paragraph could make that point more clearly*

The reviewer's comments are well taken. The gamma parameter may vary based on differences in both climate and land cover type. We propose the following change:

Line 259: The values of $\gamma$ vary by site due to **a combination of local climate and plant functional type (PFT)**. **US-KFS, modeled as a savanna, has the lowest value of and standard deviation of gamma .** The smaller magnitudes…

*Line 265 - "slower" relative to what?*

We appreciate this comment and have incorporated language making more clear comparisons like the one needed here throughout other parts of the analysis. In particular, we make sure to note comparisons between "flash drought versus drought or non-drought years". This language also helps to distinguish between the model results from drought and non-drought periods from WET and DRY assimilation periods.

Line 264: ...slower senescence and reduced variance when using the 3YR assimilation parameters as **compared to the WET and DRY parameters** during…

*The rationale for the continued focus (beyond Figure 4) of differences due to parameters sets based on wet, dry or both years is unclear - Given that using both wet and dry years clearly reduces uncertainty, I'm not sure why there is a need to compare estimates of FPAR, LAI, The authors may have a reason for this but if so it needs to be emphasized in the text. Removing this would allow the focus to be on DCHM-PV performance and the "actual" phenological mediated vegetation responses.*

The reviewer's comments are well taken.

Figure 4 helps to establish the 3YR assimilation period as the inference period with minimal uncertainty. However, we still have the goal to capture differences in model outputs across the three different assimilation periods. This is done in Figures 5,6,10,11, and 13 which show FPAR, LAI, yearly sums of GPP and ET, daily GPP, and daily ET, respectively. Figures 5 and 6 support the conclusion that the 3YR inference period does indeed show slower change in phenological state compared to WET and DRY. We agree that further comparison with the WET and DRY results is no longer needed in the body of the manuscript. We can include WET and DRY results in the appendix. We replaced Figure 7 (now Figure 10) with monthly accumulations of GPP and ET from the DCHM-PV 3YR. We removed the DCHM-PV WET and DRY from Figure 9 and it is now Figure 13.

The WET inference period predicts parameters using model outputs of soil water potential and VPD during a year that received ample rain, and therefore plenty of water for plant use, and less atmospheric demand for water. We are assuming that this inference period represents vegetation that is not accustomed to water stress and therefore is less conservative in water use strategies (Lowman and Barros, 2018). This means that when plants run low on available water (e.g., in 2012), they will consume water normally and exacerbate dry down before abruptly shutting down functions dictating water and carbon cycling. This is in contrast to parameters produced during the DRY inference period. The assumption there is that vegetation will be more conservative in water use strategies, shutting down at the first sign of stress. In both cases, mean $\gamma$ values are higher than the 3YR inference period meaning that vegetation

during the 3YR inference period is more likely to make steady changes and adaptations to water stress and less likely to make abrupt changes as seen with both the WET and DRY simulations. However, the difference between 3YR, WET, and DRY is generally minimal in terms of the magnitude of water and carbon exchanges and detailed discussion of these results is no longer included in the analysis.

*Section 3.12 and Section 3.1.3 - If I understand the methods correctly - Figure 6 and 7 show results using parameters conditions on prior information (e.g MODIS assimilation during the calibration period) and MODIS results for 2012 and 2019 - Given that, additional discussion about fit with MODIS would be helpful - How well does DCHM-PV do. Overall it captures patterns fairly well but there are some notable exceptions (e.g loss of FPAR in July and August at US-KFS in MOIDS that is not tracked by model) - It would be useful to have some presentation of model performance here*

We appreciate the reviewer's comments. We think these comments refer to Figures 5 and 6 regarding FPAR and LAI. We updated Figures 5 and 6 to include 2018 (see below). We propose to provide additional text comparing MODIS to the DCHM-PV FPAR and LAI estimates.

An example comes from the new section 3.1.3 Leaf Area Index

Line 309 (New line 333): **Generally, the predictive phenology model compares favorably with the seasonal changes observed in MODIS FPAR and LAI (Figures 5 and 6). In the summer, at US-KFS and US-KLS during 2019, the model tends to predict FPAR and LAI values higher than MODIS. In 2019, at US-KFS, MODIS observed a steady decline in FPAR from 0.8 to 0.6 throughout July with an increase back to 0.8 over an 8-day period at the beginning of August (Figure 5c). The DCHM-PV results do not show the same decline. Similarly for LAI at US-KFS (Figure 6c), MODIS observes a drop and then abrupt increase in LAI with the model estimates higher than MODIS. Yet, in June 2019 at US-Kon, the model estimates are lower than MODIS LAI.**

[Figure]

**Figure 6. Leaf area index (LAI) predicted from DCHM-PV for the flash drought year (2012), a drought year (2018), and a non-drought year (2019). Colors indicate the different data assimilation periods (yellow - 3YR (2003-2005), blue - WET (2005), red - DRY (2003)), with corresponding shaded regions representing one standard deviation of model outputs from the 2000 ensemble simulations. The 8-day MODIS MOD15A2H LAI is shown in black markers. The gray shaded regions in the left most panels highlights the 2012 flash drought period..**

To be added to discussion under new section 4.6.1 Model vs Observations. An excerpt is provided above in response to Major Comment 3b

*Line 315 - which are water stress years (e.g 2012). Also can you note which method (or averaged across all methods) does the 1kgCm2 reduction come from*

The authors appreciate this comment from the reviewer. We are replacing Figure 7 and this paragraph has been rewritten so lines 310-319 will be removed. Instead, we compare DCHM-PV 3YR monthly totals  across drought and non-drought years as listed above instead of showing yearly sums of GPP and ET.

New section 3.3.2 Gross Primary Productivity with the following excerpt

[revised manuscript text omitted]

*lLine 380 - The arguments in the first 3 sentences of this paragraph need a bit more detail. Just because there are fluctuations in evaporation this doesn't not necessarily mean that "all" water evaporated before it had a chance to infiltrate.*

We believe this quote comes from line 340. The authors appreciate this comment and we agree. The new infiltration section described in Major Comment 2a supports the claim that reduced infiltration drives down plant water availability.

The timing of infiltration and evaporation are different. While infiltration and evaporation occur in response to precipitation events, the timescales associated with each process are different. Water infiltrates the soils following precipitation but may not be available for plant use in instances when increased VPD (Figure A5) leads to enhanced evaporation of soil water. It was therefore prudent to account for accumulations over monthly timescales to better compare infiltration and evaporation totals.

Infiltration does occur, but it is drastically reduced during flash drought. Water availability is reduced due to both a lack of precipitation and increased days between precipitation events (Figures 8, S4) leading to reductions in root uptake (i.e. transpiration, Figures 12, S6). The large fluctuations in ET are therefore more associated with increased evaporative demand, reduced infiltration limiting root water uptake,and overall lack of water availability.

 Line 339: …are a result of evaporation in response to precipitation (Figure 19a). This suggests **that following precipitation events during flash drought onset, ET is dominated by evaporation. Reduced available water infiltrating the soils limits water available for root water uptake.** Since…

[Figure]

**Figure S19. Daily evapotranspiration, ET, partitioned into evaporation, E, and transpiration, T, in mm d$^{-1}$ at US-KFS for (a) 2012 flash drought, (b) 2018 drought, and (c) a 2019 non-drought year. The curves represent ensemble means from the DCHM-PV 3YR. Daily precipitation accumulation is shown on the right axis..**

*Note the substantial underestimation of ET by the models relative to Ameriflux in 2012 should be noted here as well along with some discussion of why*

This comment and suggestion from the review is well taken. Please see response to Major Comment 3a above.

As noted above, modeled ET should only consider the daytime values when computing daily averages because the model shuts down evaporation at nighttime when there is no incoming solar radiation. However, in the original manuscript we computed daily averages over a 24 hour period. With the update to Figure 9 (now Figure 13), we see that the modeled results match well against AmeriFlux. Differences still occur in 2012, with modeled ET agreeing with AmeriFlux starting in April through mid-May. Once the flash drought onsets (late May through July), modeled results tend to be lower than AmeriFlux. Once the flash drought ends in August, modeled ET once again agrees with AmeriFlux. In a drought and non-drought year, modeled ET appears to match better throughout most of the season (Figure 9). One explanation could be that water use by vegetation during flash drought is considerably different, and the model is not able to recreate this change in survival strategy.

In the results section which will be updated to be section 3.3.3 Evapotranspiration. Some of Section 3.3.3 was presented above. The rest is here.

**During the flash drought, transpiration gradually declined from May to July (Figures 12, S19a). The fluctuations in total ET starting in June 2012 are the result of evaporation in response to small precipitation events. This suggests that following precipitation events during flash drought onset, ET is dominated by evaporation. Reduced infiltration limits water available for root water uptake (Figures 7 , S6). As transpiration is computed from root water uptake across the three soil layers, the observation that transpiration decreases but maintains a small consistent rate through the flash drought indicates that vegetation is extracting water from deeper soil layers. ET never completely shuts down in 2012 because of the low rate of transpiration. However, evaporation completely halts during early July 2012, which is the peak of the flash drought period. Similar to flash drought, during drought in 2018, ET is dominated by evaporation (Figure S19b). But in the non-drought year 2019, transpiration makes up more than 50$\%$ of ET throughout the entire growing season except for short periods in July and August (Figure S19c).**

**Daily ET estimated by the DCHM-PV matches well against AmeriFlux estimates at US-KFS during the flash-drought, and non-flash drought years (Figure 13). In 2012, DCHM-PV ET agrees with AmeriFlux through mid-May. From late May through July the model results tend to fall below AmeriFlux until August when they once again agree. In the drought (2018) and non-drought (2019) years, DCHM-PV ET appears to align with AmeriFlux throughout most of the season (Figure 13 b,c). While model estimates of ET are higher than flux tower measurements in 2019 at US-KLS, they compare favorably in 2012 and 2018 (Figure S17). In contrast to model and flux tower comparisons at US-KFS and US-KLS, at US-Kon modeled ET (Figure S18) agrees with AmeriFlux in 2019 (non-drought), but underestimates during the summer months in 2012 (flash drought)**

**and 2018 (drought). One explanation for the differences between model and tower ET data could be that water-use by vegetation during flash drought is highly variable across sites, and the model is not able to represent all possible responses. Additionally, it is difficult for the DCHM and other Earth system models to account for plant access to deep water stores (Giardina et al., 2023).**

[Figure]

**Figure 13. Daily evapotranspiration, ET, [mm d$^{-1}$], at US-KFS for (a) 2012 flash drought, (b) 2018 drought and (c) 2019 a non-drought year. Two standard deviations are shown for the DCHM-PV simulations. AmeriFlux ET is derived from latent heat measurements and shown as blue dots.**

Similarly Figures are included in Supplemental Material for US-KLS (S17) and US-Kon (S18).

*line 360 - Its worth noting that the drought response can be more complicated than simply shutting stomata - importantly grasses can shift their allocation of carbon - and this will be reflected in above ground biomass (the GPP measured by MODIS) but also in below ground stores and fluxes - For example see Ingrisch, Johannes, Stefan Karlowsky, Roland Hasibeder, Gerd Gleixner, and Michael Bahn. "Drought and recovery effects on belowground respiration dynamics and the partitioning of recent carbon in managed and abandoned grassland." Global Change Biology 26, no. 8 (2020): 4366-4378.*

This helpful comment from the reviewer is well taken. Stomata closure implies less gas exchange. Ultimately, this drives the decreases in modeled losses to ET and GPP, but could

explain the higher GPP observed in the AmeriFlux data. Vegetation, at least grasses, can reallocate already processed carbon to their roots when under drought stress mitigating GPP losses (Ingrisch, et al., 2020). This means that MODIS may see a reduction in phenological states (LAI and FPAR) but maybe GPP is less affected.

Line 360: …ET (Chen et al., 2019). **In some cases, vegetation can reallocate already processed carbon to their roots when under drought stress mitigating GPP losses (Ingrisch, et al., 2020). However, modeled GPP losses are likely a result of modeled stomatal behavior, as the model does not account for reallocation of carbon stores within the plants. This limitation of the model could explain why AmeriFlux GPP tends to be higher than the modeled GPP. Sub-daily scale stomatal conductance reduces to zero in response to increased VPD (Figure 9) leading to similar reductions in modeled GPP (Figure S22).**

[Figure]

**Figure S22. Simulated daily totals of GPP and ET from the DCHM-PV 3YR assimilation period for (a) 2012, flash drought year and (b) 2019, wet year.**

*Similarly on line 380 - there is ample evidence of changing root allocation (and root respiration) for grasslands that would be worth citing here.*

The reviewer's comment is well-taken. We now discuss changing root allocation and respiration with relevant citations. See edited text below.

The changes have been implemented in Section 4.6.2 Implications for Land-surface Models as written above.

*line 369 - Note that authors don't really show that evaporation after rain effects uses all available water - so it doesn't infiltrate (as stated in the hypothesis) - but they could do this at least with the model since both daily evaporation and precipitation is available.*

We appreciate this comment and it is part of a larger theme addressed in the response to Major Comment 2. Note from Figure S4, precipitation exceeds infiltration. Excess amounts of water not contributing to infiltration can be attributed to runoff.

Much of our response is addressed with the addition of Section 3.2.1 Infiltration above. We also include a section in the Discussion

**4.2.1 Infiltration and Evaporation**

**At the onset of flash drought there is an increase in evaporative demand for water which leads to a temporary increase in surface evaporation (Lowman et al., 2023; Otkin et al., 2018) until the soil and canopy reservoirs no longer contain enough water to evaporate. Then evaporation shuts down. Despite evaporation tapering to zero during June and July of 2012 (Figure S19), pulses of rainfall lead to temporary rapid increases in rates of evaporation. Increased surface evaporation may reduce water infiltrating the soils. In May of 2012 at US-KFS there was 70 mm of water infiltrating the soils (Figure 7) with 35 mm of evaporation (Figure 12b). But in June and July total infiltration was 61 mm with 65 mm of evaporation over the two months. Similar comparisons can be found at US-KLS and US-Kon (Figures 7, 10). In contrast, at US-KFS, during non-drought years, June averages of infiltration are in excess of 100 mm with 41 mm of evaporation. Average drought years have 66 mm of infiltration with 47 mm of evaporation (Figure 7a). Since infiltration usually exceeds evaporation in the growing season, infiltration accumulations of similar magnitude to evaporation totals may indicate flash drought.**

*line 375 - "since GPP is decreasing" as the authors themselves note - declines in GPP do not always reflect what's happening with transpiration so this statement needs some caveats*

The point is well taken, especially with grasses as noted above, reallocating their stores of carbon which can be another reason why modeled GPP isn't solely tied to changes in transpiration. However, the DCHM strongly links transpiration to GPP and does not account for reallocating carbon storages in other ways. We will use a new paragraph at Line 367-370 that leads into the current paragraph at line 371.

Line 367 (now Section 4.3 Linking Carbon and Water Fluxes, excerpt): **Despite major reductions in infiltration and fluctuations in top layer soil moisture during flash drought onset, modeled root water uptake indicates that plants were still pulling small amounts of water through their roots, preventing them from completely shutting down. With the ability to tap into water stores from deeper layers (Giardina et al., 2023) and small rates of transpiration still occurring, modeled carbon uptake is still maintained (Figure 11a, S15a, S16a).**

*In the  discussion, one of the challenges here is that observations and models suggest differences in plant ability to pull water from deeper layers - The first paragraph blurs these distinctions  - so for example the line  "we did find that even during the*

*peak..plants were still" - is this based on flux towers, or model?. I also note that model and observed in Figure 11 suggest different stories about water use efficiency during the drought. The observed data suggests plants maintain higher water use efficiency, longer, during the drought than the model - this is very interesting - its suggests plants are doing something that the model misses which is informative - but needs to be much more of a focus in the paper.*

The reviewer's comments are well taken.

We now include language in the paragraph above (previous response, Line 367) to indicate that we were referring to model outputs regarding root water uptake. In the larger context, we acknowledge that there may be discrepancies between models and flux tower measurements due to below ground processes that models are missing (see response to Reviewer Comment 3b above).

Figure 11 has been updated (and moved to supplementary material as Figure S24) to average over daytime hours for water use efficiency. Corresponding figures for US-KLS and US-Kon are included in the appendix. Updates to Figure 11 include the results from averaging WUE over the daytime and the addition of 2018. We also updated the legend and y-axis label as well as an extension of the viewing window to include all of October. We now align with tower values better in 2012. However, in 2018 the DCHM underestimation of tower WUE can be attributed to the differences in the 2018 modeled and flux tower measurements of GPP since modeled ET at US-KFS matches well against tower measurements. See the updated Figures 8 and 9 above for reference to GPP and ET, respectively.

[Figure]

**Figure S24. Growing season water-use efficiency (WUE=GPP/ET) from DCHM-V, DCHM-PV (3YR), and AmeriFlux for (a) 2012 flash drought, (b) 2018 drought, and (c) 2019**

**non-drought at US-KFS. Ensemble means shown for DCHM-PV with 2 standard deviations (shaded).**

*Figure 12 - its not so easy to see from this figure that when plant are transpiring more they are more efficient in their water use - Simply graphing transpiration vs WUE would show this much more clearly.*

The reviewer's comments are well taken and appreciated. In fact, our results agree with the point that plants transpiring more are not necessarily more efficient in their water use.

Figure 12 has been updated below. The main point of the new Figure 12 is to show that transpiration as a percentage of overall ET and WUE follow the same transition from normal or above average levels to drought levels from May to July. This suggests that both follow the same pattern and can be used to mark the rapid transition to drought state.

Given the new analysis and figures of transpiration accumulation and average water use efficiency in drought and non-drought years compared with non-drought years we can better observe the trends associated with WUE and transpiration. The new Figure 12 suggests that plants are more efficient in non-drought years. There is an association with decreased T/ET and decreases WUE. However, that does not substantiate the claim that "plants that are transpiring more are more efficient." We no longer like to make this claim.

[Figure]

**Figure 15. Ratio of transpiration to evapotranspiration, T/ET, and water-use efficiency, WUE, for for drought (red) and non-drought (blue) years compared to flash drought (black) for US-KFS, US-KLS, and US-Kon.**

Figures 11 and 12 were updated. Figure 11 now includes a daily average time series at US-KFS for 2018 in addition to 2012 and 2019. Corresponding figures for US-KLS and US-Kon will be included in the supplemental material (Figures S24-26). Figure 12 was updated to compare flash drought T/ET ration and WUE to aggregate monthly averages from drought and non-drought years (now Figure 15). We also added October to the analysis period.

A more apt comment using Figure 15 (Old Figure 12 ) has been added to the discussion section.

**4.3 Linking Carbon and Water Fluxes.**

**Plants are more efficient during non-drought periods, and are less efficient during flash drought onset (Figure 15). Ratios of T/ET also indicate plants that transpire more are more efficient in their water-use. WUE is similar at US-KFS in August-October regardless in drought and non-drought years which might be attributed the site being modeled as a cropland. WUE at all sites started off in 2012 with above average non-drought levels and an increase from April to May. However, from May-July WUE at all sites fell from above non-drought years to more than one standard deviation below drought years. With GPP differences being more substantial than ET between flash drought and non-flash drought periods (Figure 10), subseasonal reductions in WUE can be attributed to the losses in GPP. Reductions in WUE from above  non-drought conditions to below drought conditions (e.g., the 60\%-70\% reduction from May to July in 2012, Figure 15 d,e,f), appear to be a feature of flash drought onset.**

From Figure S27, we can see that changes in transpiration explain a small component, less than 10%, of the variability observed in WUE. Generally when plants have more available water (e.g., 2019), they transpire more, but higher values of WUE can be seen in flash drought and drought years (e.g. 2012, 2018) despite having lower rates of transpiration.

[Figure]

**Figure S27 (appendix). Daily averages of water use efficiency versus transpiration for 2012, 2018, and 2019.**

*Also in figure 12 - some strategic use of color to differentiate wet versus dry years would be helpful here*

We appreciate this comment.

Upon initial submission of the manuscript, the processing editor suggested making some changes and updates to make plots and figures more readable. Changing to figures include updated color schemes, font sizes, line weights, line styles, and markers. Color schemes were generated using a tool Coloring for Colorblindness https://davidmathlogic.com/colorblind/#%23D81B60-%231E88E5-%23FFC107-%23004D40 and figures were tested on Coblis - Color Blind Simulator https://www.color-blindness.com/coblis-color-blindness-simulator/.

*Line 403 - The idea of flash drought responses is intriguing but I think the paper could do much more to support these ideas (e.g that the rapidness of the change is indicative of a flash drought) - Some more strategic comparison of the declines across different type of drought - flash versus "non-flash" (Of course this would require clearly distinguish what a flash drought is from other types of drought - but that seems to be part of the paper's motivation)*

The authors thank the reviewer for this thoughtful comment. A major goal of the authors' responses to the reviewer comments throughout this document has been to refocus analysis to compare results (e.g., GPP, ET, WUE, etc.) from a flash drought year to drought and

non-drought years (see response to Major Comment 1). It is our intent that the rapid changes seen from May to July in 2012 model results of vegetation-atmosphere interactions along with the decreased precipitation and increased atmospheric demand for water highlight the transitional period of flash drought intensification. Furthermore, we hope this emphasizes flash drought as being the rapid development of drought that can be observed through land-atmosphere interactions.

Responses to Anonymous Reviewer 2 for paper: *Unraveling phenological to extreme drought and implications for water and carbon budgets*

The authors greatly appreciate the reviewer taking time to provide feedback on our manuscript. We think that the comments are helpful in shaping the new analysis and clarifying how important mechanisms driving plant controls of water and carbon movements may be affected during flash drought.

In light of new analysis described in our responses below, we propose a new title: ***Unraveling phenological and stomatal responses to flash drought and implications for water and carbon budgets***

We respond to each of the Reviewer's comments below, which are in **bold and italics**. Author responses are in blue with proposed manuscript changes in **bold**. Responses in *italics* are used to reference exact quotes from the author's responses to Reviewer Comment 1 (RC1). New figures that we propose to include have labels N# throughout this document and the number corresponds to how the new figures were introduced starting with RC1. Revised figures have the same figure number as in the original manuscript. Figures used to support claims in this document that are not going into the revised manuscript do not have Figure numbers but we have included figure captions.

*This study investigates vegetation phenological processes during drought and non-drought periods using a set of DCHM-V and DCHM-PV model simulations. The focus is on three sites in Kansas USA, as they experienced extreme drought and pluvial conditions in recent decades, and also have ground-based and satellite-based observations available to compare with model simulations and study observed processes. The modeling experiments are neatly designed, and the investigation is systematic to study sources of uncertainty in vegetation phenology. I however have a number of comments that need to be addressed – please see below.*

*Main comments*

*1) This study uses 2012 and 2019 to exemplify the contrast of vegetation phenology between flash drought vs non-drought years. It would help to add a non-flash/conventional drought year to the study, as the vegetation phenology could differ considerably between flash drought and conventional drought. It would be interesting to see how the evolution of plant phenology, water use, and productivity may differ between the two drought cases.*

The authors appreciate this comment and agree with the reviewer that we should highlight the differences between flash drought and "non-flash/conventional drought". To this end, we propose to add 2018 (conventional drought) as a specific case study to compare against 2012 (flash drought) and 2019 (non-drought). Additionally, we focus components of the analysis to consider average conditions across conventional "drought" and "non-drought" over the course of

the period 2006-2019. This major change is described in our response to Major Comment #1 from Reviewer #1.

Examples of how we compare flash drought to non-flash drought can be seen with the new Figures 10 and 11 above.

*2)    Much of the findings are based on the DCHM-V and DCHM-PV simulations, and are thus subject to the performance of the DCHM and its predictive phenology in simulating observed land surface and vegetation processes. The comparison between the model results and independent observations (e.g., MODIS, AmeriFlux) however shows considerable differences: some of the models vs. observations differences are so substantial that they are much larger than the differences between different model experiments (e.g., Figs.5-9). While these differences could be in part due to the data comparison across different spatial and temporal scales (Section 4.5), they also make one wonder about the performance of the DCHM and its predictive phenology. I suggest the study provides more information/results on the fidelity of the DCHM and its predictive component is simulating basic land surface variables (e.g., soil moisture, evapotranspiration) and observed vegetation phonology related fields (e.g., LAI, FPAR), e.g., in terms of climatology, seasonal cycle and year-to-year variations, to make the model-based findings more convincing. Also see some of my detailed comments below.*

The author's appreciate this reviewer's comment and we agree that more discussion is needed involving the comparison of the results from the DCHM and measurements and observations from AmeriFlux and MODIS.

Major Comment 3 from our responses to RC1, who shared similar feedback, addressed this point. See response above, especially the addition of **Section 4.6.1 Model vs Observations.**

*Detailed comments*

*1)    It would help to briefly discuss the implications of the findings (e.g., based on WET vs DRY vs 3YR) to subseasonal prediction of vegetation.*

Our original intention was to simulate different plant isohydric and anisohydric tendencies following Lowman and Barros (2018) who showed that the data assimilation period can be used to generate phenology model parameters that represent different water use strategies. Following this logic led us to test parameters using WET, DRY, or mixed conditions (3YR) to simulate anisohydric vs isohydric tendencies among the different plants. Our results show that the data assimilation period may not be the only factor to consider when trying to simulate water use strategies. The DCHM predicts stomatal conductance depending on vapor pressure deficit (VPD), light exposure, and soil moisture. High temperatures and low relative humidity lead to increases in VPD. In the model, high VPD leads to very low (or zero) stomatal conductance (Figure 9). With little water available and high VPD, the DCHM-V and the DCHM-PV follow very closely. The DCHM-PV predicts higher stomatal conductance than the DCHM-V when ample

water is available and there are lower values of VPD (Figure 9). This translates to higher GPP predictions in non-drought years (Figures 10, 11).

We suggest the following updates to the discussion section **4.1 Mechanisms Controlling Plant Responses to Drought** and its subsections. We also added implications in section **4.6 Model Performance and Limitations** above.

**4.1.1 Stomatal and Non-stomatal Regulation of Gross Primary Productivity**

**4.1.2 Vapor Pressure Dependence**

**While phenology is an important component to consider when computing changes to transpiration and carbon uptake (Lowman and Barros, 2018; Flack-Prain et al., 2019),, our results indicate that stomatal conductance is also critical for accurately representing these fluxes. Plants adaptively regulate their stomata during periods of water stress (Guo et al., 2020), and some have been demonstrated to maintain open stomata or even increase stomatal conductance under high VPD conditions (Urban et al., 2017). Stomatal conductance shuts down under high VPD in the DCHM (Figure 9), which does not account for the possibility of an adaptive stomatal regulation strategy. Since GPP is directly dependent on stomatal conductance (Farquhar and Sharkey, 1982), DCHM estimates of sub-daily GPP decrease in response to elevated VPD (Figure S22). Moreover, changes in phenological growth state (i.e. LAI) occur across longer (i.e. seasonal) time scales (Katul et al., 2001) than stomatal regulation, which controls carbon and water exchange at sub-daily timescales (Guo et al., 2020).**

**The differences between modeled and observed GPP and ET (Figures 11, 13) suggest that there are mechanisms controlling plant responses to drought stress not accounted for within the DCHM. For example, the DCHM could be too strict in representing the sensitivity of stomatal closure to elevated VPD for the Kansas study sites. There could be plant or climate specific VPD dependence (Grossiord et al., 2020), plants could have access to stores of water not accounted for (Giardina et al., 2023), or both. Guo et al. (2020) showed that isohydricity (i.e. stomatal regulation) exists on a spectrum and that some plants are able to move along that spectrum at sub-daily time-scales with varying environmental conditions, such as higher VPD. Given the high VPD in 2012 at our study sites (Figures S5, S28, S29, S30), we expect the DCHM to estimate low stomatal conductance, and thus low GPP relative to AmeriFlux observations when under atmospheric water stress. Additionally, VPD estimated by the DCHM using the NLDAS-2 Forcing File A atmospheric variables is higher during 2012 and 2018 and lower in 2019 than the AmeriFlux observations (Figure S28), explaining in part the discrepancies between model and AmeriFlux GPP. As stomatal response to increasing VPD and resulting impacts on land-atmosphere water fluxes is more complex than how it is represented in LSMs (Vargas Zeppetello et al., 2023), future modeling studies should focus on how rising VPD drives stomatal closure across different vegetation types (Grossiord et al., 2020).**

[Figure]

**Figure S28. Daily vapor pressure deficit at US-KFS for (a) 2012 - flash drought, (b) 2018 - drought and (c) 2019 - non-drought. The DCHM computes VPD using air temperature and vapor pressure from NLDAS-2 Forcing File A.**

*2)   Noah-LSM: Noah LSM has multiple versions. If the Noah-LSM used in this study refers to the Noah in NLDAS-2, please specify.*

The reviewer's point is well-taken. The Noah-LSM in this study does refer to the Noah model employed in NLDAS-2 (Xia et al., 2012). We will update the soil moisture figure captions and any references to NLDAS-2 soil moisture computed using Noah-LSM in the main body of the manuscript and in the Appendix. We also propose to combine figures from the Appendix so that Figure A1 and A2 become Figure S1 a,b,c (added above) to represent the top layer soil moisture at US-KFS for 2012, 2018, and 2019. Now the legends in the soil moisture figures have **NLDAS-2** instead of Noah-LSM.

*3)   line 259: of gamma => of the growth rate parameter*

This comment is well taken. Following a similar comment from Review 1, we now use the greek letter/symbol gamma once it is defined throughout the remainder of the manuscript rather than going back and forth between gamma and the growth rate parameter.

*4)   Figure 12: May want to increase the thickness of curves for 2012 and 2019 to highlight the results for these two years*

This comment is well taken. We have updated many figures to use thicker lines, varied color, dashed lines, and new marker shapes to help distinguish between simulations/years. Figure 12

has been completely reformatted so the flash drought can be compared to other drought and non-drought periods, as opposed to solely 2019. It is now Figure 15.

5)   *Line 390: (Figure 10 => (Figure 10)*

This reviewer comment is well taken, and we will implement this change.

6)   *Figures A3, A5. Middle and deep layer soil moisture for the flash drought year 2012. How to explain the substantial differences between DCHM-V/DCHM-PV and Noah-LSM? Noah-LSM seems to make more sense as it shows a notable decline after June 2012. In contrast, the soil moisture in DCHM-V/DCHM-PV remains relatively steady throughout 2012 and does not seem to be responsive to the strong precipitation deficits during 2012, which looks odd; this is concerning as any issues in simulating soil moisture would adversely impact the simulation of vegetation and evapotranspiration processes etc. Please also see my second main comment.*

The reviewer's comment is well taken. First, see updates to Figure A3, which will now be S2 and combine 2012, 2018, and 2019 middle layer soil moisture for US-KFS. We respond below by (1) explaining why we see differences between the DCHM and NLDAS-2 soil moisture, and (2) by describing how these differences impact estimates of carbon uptake (GPP) and transpiration (T). We investigate soil moisture, GPP, and T by comparing our results to another modeling study who investigated US-KFS and US-Kon during 2012 and 2018 (Hosseini et al. 2022).

**NOTE: We cannot reproduce the figures referenced from Hosseini et al. (2022) here. Instead, we reference specific figures and panels from their paper for comparison. In reference to soil moisture, see the bottom four panels of Figure 6 in Hosseini et al. (2022). In reference to GPP, see bottom panels of Figure 3 in Hosseini et al. (2022). In reference to transpiration, see the third panel of Figure 5 in Hosseini et al. (2022). In reference to LAI, see the top panels of Figure 6 in Hosseini et al. (2022).

(1) Why we see differences

**NOTE: In the following paragraphs we compare DCHM soil moisture from different layers to other products (SMERGE, NLDAS-2) and model outputs Noah-MP (Hosseini et al., 2022). Layer depths do not directly compare so for reference, we briefly state the various depths used.

The DCHM top layer soil moisture is an average over 0-8 cm, the middle layer is 8-89 cm, and the deep layer is 89-183). Depths were determined from the Kansas Soil Survey (Soil Survey Staff).  In Hosseini et al. (2022) the top layer in Noah-MP is 0-10 cm and the deep layer average soil moisture they present comes from three layers with thicknesses of 30, 60, and 100 cm. Effectively, this is an average over 10-200 cm vs the DCHM which ranges from 8-183 cm . We average the DCHM middle and deep layers for comparison (see Figure below) and convert Noah-MP estimates into volumetric soil water content for comparison. The NLDAS-2 soil moisture depths used for comparison are 0-10 cm, 0-100 cm, 100-200 cm (Xia et al. 2012) to compare against the DCHM top, middle, and deep layers, respectively. In figures of the middle

layer soil moisture, we include comparisons to SMERGE 0-40 cm, computed from "merging" NLDAS and the European Space Agency satellite soil moisture (Tobin et al. 2019).

A first explanation for why we see differences between DCHM modeled soil moisture and NLDAS-2 is that NLDAS-2 soil moisture was estimated from the Noah-LSM without predictive phenology (Xia et al., 2012). However, Hosseini et al. (2022) used various Noah-MP configurations (including with and without predictive phenology) to compute soil moisture, and the DCHM results match well with their soil moisture at US-Kon in 2012 and 2018 (see Figure 6 in Hosseini et al. 2022). Converting units from mm to m^3m^{-3}, we see that Noah-MP predicts a drop in 2012 soil moisture at US-Kon from ~0.35 to 0.28 m^3 m^{-3} from January to September while the DCHM sees a drop of about from ~0.36 to 0.30 m^3 m^{-3}. The Noah-MP model configuration that uses dynamic LAI and vegetation fraction (V3-LD-FD) predicts soil moisture decay from June-September that shows the least steep decline in soil moisture from late June to late August (Hosseini et al. 2022, Figure 6 bottom panel), aligning with results from the DCHM-V and -PV (Figure below and addition SM figures for US-Kon further down - none of which are included in manuscript but are included here for justification).

[Figure]

Figure. DCHM-PV 3YR volumetric soil moisture averaged across middle and deep layers for US-Kon in 2012, 2018. This figure is not included in the manuscript.

It is also important to note that Hosseini et al. (2022) estimates of the top 10 cm of soil moisture match well the magnitude of flux tower soil moisture, fluctuating between ~0.15-0.3 m^3 m^{-3}, between May and July. These findings agree favorably with DCHM top layer soil moisture in 2012 (See below for additional US-Kon SM figures). However, like the DCHM, all model configurations of Noah-MP in Hosseini et al. (2022) estimate lower soil moisture compared to field measurements in the top layer from mid-February to early May 2012 and higher soil moisture from early May through the rest of the year except for some spikes preceding larger

rainfall events. Similarly for top layer soil moisture results from 2018, all of the Hosseini models and the DCHM overestimate soil moisture compared to field observations starting in late April and throughout the end of the year (See below for additional US-Kon SM figures). Thus, the DCHM model results for soil moisture in 2012 and 2018 at KON are in line with what has previously been estimated from different configurations of the Noah-MP that use predictive phenology and differ similarly from the NLDAS-2 dataset and field observations of soil moisture.

A second explanation of the DCHM estimating higher soil moisture than NLDAS-2 might have to do with cascading effects high VPD has on stomatal conductance. In response to high VPD in the DCHM, stomatal conductance shuts down (Figure 9). Therefore plants are not transpiring. Reduced transpiration is directly tied to reduced root water uptake, resulting in the soils retaining comparatively higher levels of moisture. Figure S6 shows that modeled middle and deep layer root water uptake decreases ~50% from May to July 2012 at US-KFS. Within the DCHM, reduced root water uptake (Figure S6) is likely why estimates of soil moisture in the middle and deep layers remain higher compared to SMERGE and NLDAS-2 (using Noah-LSM) soil moisture at US-Kon (US-Kon SM figures below). However, the DCHM and SMERGE agree favorably in 2012 and 2018 throughout most of the growing season at US-KFS. Note that the DCHM matches well middle and deep layer estimates of soil moisture from NLDAS-2 and SMERGE in 2019 when there is ample water available for plant use within the DCHM.

  (2) How these differences impact estimates of GPP and transpiration

GPP

The DCHM estimates low GPP and stomatal conductance rates during the flash drought period in 2012, while eddy covariance data recorded elevated rates of GPP (e.g., Figure 9, 11,S22). The low estimates of GPP and stomatal conductance from the DCHM are directly related to high atmospheric aridity (or VPD) indicating that the DCHM slows carbon and water exchanges under atmospheric water stress, despite sufficient soil moisture to undergo photosynthesis.

Hosseini et al. (2022) report 11-year (2008-2018) averages of GPP for US-Kon and US-KFS using different Noah-MP configurations, MODIS and AmeriFlux data (Figure 3 in Hosseini et al. 2022). In the figure below, we show the same 11-year averages computed from the DCHM-PV. Noah-MP using predictive LAI configurations estimates higher GPP in April (~150-200 gC m^{-2}) and May (~300 gC m^{-2}), than the DCHM by ~100 gC m^{-2} for similar soil moisture during this time (see Figure 6 in Hosseini et al. 2022 and Figure S2 for KFS and figure included here only). Both Noah-MP and the DCHM GPP peak in June and the Noah-MP results fall within one standard deviation of the DCHM in June and July at both sites. However, the DCHM 11-year averages of GPP match well the Apr-Oct averages from flux towers for KFS. Noah-MP includes routines for reallocating carbon to different parts of plants (i.e. stems, roots, etc.) that may account for the higher estimates of GPP compared to the DCHM, which does not include such processes.

[Figure]

[Figure]

Figure. Monthly GPP averages across the same 11-year period (2008-2018) as Hosseini et al. (2022) using ensemble mean estimates from the DCHM-PV 3YR. Error bars represent one standard deviation from the 11-year average.

Transpiration

The maximum daily transpiration rate estimated from the DCHM, which computes transpiration from root water uptake, is 1.25 mm d^{-1} in 2012 and 2018 (Figure below), but the Noah-MP modeled transpiration reach over 2mm d^{-1} in May and June for both 2012 and 2018. July - September rates of transpiration for both the DCHM and Noah-MP (with dynamic LAI) fall to less than 0.5 mm d^{-1}. Peak transpiration in May and June of 2012 before a decrease to lower transpiration rates in July-October is observed in both the DCHM and Noah-MP (see the third panel of Figure 5 Hosseini et al., 2022) although there are differences in magnitude of transpiration, some of which can be attributed to the differences in computed LAI. Like Hosseini et al., (2022), the DCHM estimates two seasonal peaks of transpiration in June and September of 2018. The late season peak seems to align with large increases in late season precipitation.

Some of the discrepancies in transpiration may result from differences in estimated LAI from both models. The DCHM-PV estimates of LAI tend to agree favorably with the timing of green up and seasonal changes compared to MODIS (see RC1 for full Figure 6 showing LAI at all three sites from 2012, 2018, 2019). At US-Kon, the DCHM-PV 3YR shows April LAI less than 1 m^2 m^{-2} (see our Figure 6g below,), but Hosseini et al., (2022) estimates leaf out earlier and with April LAI at ~2.7 m^2 m^{-2} (see top panels of their Figure 6). The uptick in transpiration seen by Hoesseini in September 2012 might also be due to the increase in LAI from 0.2 to 2.0 m^2 m^{-2} that they found at the same time. Meanwhile, the uptick in LAI seen by the DCHM-PV was from 1.0 to 1.2 m^2 m^{-2}.

Overestimating LAI leads to overestimating latent heat fluxes, as transpiration is a component of latent heat. DCHM estimates of latent heat in May and June of 2012 are less than that of flux tower by ~100 W m^{-2} and match tower measurements well when during wet periods, like 2019 (Figure below). In Noah-MP (Niu et al, 2011; Ma et al., 2017, Li et al., 2021) and in the DCHM, transpiration is directly related to root water uptake which depends on canopy (and stomatal) conductance and both models compute canopy conductance using LAI. Soil moisture across the two models was similar, but LAI varied by over 1 m^2 m^{-2} during the growing

season. Thus, LAI and not differences in soil moisture are likely responsible for differences in modeled GPP and transpiration. .

[Figure]

Figure. Daily transpiration averaged over daytime.

[Figure]

Cropped from Figure 6. Leaf area index (LAI) predicted from DCHM-PV for the flash drought year (2012), a drought year (2018), and a non-drought year (2019). Colors indicate the different data assimilation periods (yellow - 3YR (2003-2005), blue - WET (2005), red - DRY (2003)), with corresponding shaded regions representing one standard deviation of model outputs from the 2000 ensemble simulations. The 8-day MODIS MOD15A2H LAI is shown in black markers. The gray shaded region highlights the June to July decrease in FPAR during the 2012 flash drought.

[Figure]

Figure. DCHM estimates of latent heat at US-Kon for 2012, 2018, 2019 compared with AmeriFlux.

[Figure]

**Figure S2** (newly created to combine A3 and A4 and adding 2018**). Middle layer soil moisture at US-KFS for (a) 2012, flash drought, (b) 2018 drought and (c) 2019 a non-drought year using the DCHM-V (black dotted line), the DCHM-PV with two standard deviations (red), SMERGE (green dashed line), NLDAS-2 derived from Noah-LSM (yellow) and Stage IV precipitation on the top and right axes (blue).**

[Figure]

**Figure S6. DCHM-PV 3YR monthly root water uptake totals for drought (red) and non-drought (blue) years compared to flash drought (black) across three soil layers for our three study sites. Monthly sums are computed from the ensemble means of the 2000 Monte Carlo simulations then averaged across drought or non-drought years. Error bars represent one standard deviation across drought and non-drought years, respectively.**

[Figure]

Figure. Top layer soil moisture at US-Kon for (a) 2012, flash drought, (b) 2018 drought and (c) 2019 a non-drought year using the DCHM-V (black dotted line), the DCHM-PV with two standard deviations (red), AmeriFlux (blue dashed line), NLDAS-2 derived from Noah-LSM (yellow) and Stage IV precipitation on the top and right axes (blue).

[Figure]

Figure. Middle layer soil moisture at US-Kon for (a) 2012, flash drought, (b) 2018 drought and (c) 2019 a non-drought year using the DCHM-V (black dotted line), the DCHM-PV with two standard deviations (red), SMERGE (green-dashed line), NLDAS-2 derived from Noah-LSM (yellow) and Stage IV precipitation on the top and right axes (blue).

*7)    Figure A6 is identical to Figure A5 and appears to be incorrect. Please check if it plots the results for 2019.*

The author's appreciate the reviewer pointing this out. We have fixed this mistake and combined into one figure while adding 2018. This mistake also happened with A1 and A2 (see combination above).

[Figure]

**Figure S3** (new and result of combining A5 and A6 with results from 2018**). Deep layer soil moisture at US-KFS for (a) 2012, flash drought, (b) 2018 drought and (c) 2019 a non-drought year using the DCHM-V (black dotted line), the DCHM-PV with two standard deviations (red), NLDAS-2 derived from Noah-LSM (yellow) and Stage IV precipitation on the top and right axes (blue).**

*8) Figure A10: "during 2012"=>"during 2019'?*

The authors thank the reviewer for pointing out this error. We will update the figure caption accordingly. We also propose to provide updated figures with the DCHM averaged over only the daytime hours as mentioned above in response to Major Comment 2. We update the color scheme to be monochromatic grayscale to be more vision friendly. It should be noted that the WET and DRY were identical to the 3YR. This was a bug in the plotting code that we fixed.

An example of one of the new figures is below. With the addition of new figures, A10 became S14.

[Figure]

**Figure S14. MODIS (MOD17A2H) vs DCHM-PV 3YR, WET, and DRY for all three sites during 2019.**

*9) Figure A11a: The difference between Ameriflux and model simulation is striking. The inclusion of Ameriflux appears to cause confusion rather than providing a truthful evaluation of the model results.*

The authors appreciate this comment from the reviewer. The data discrepancies were striking and were the result of an error made when plotting. See Response to Major Comment 3a from the responses to RC1 and Response to Major Comment 2 above.

With updates to how we compute daily averages from model GPP and the use of AmeriFlux FLUXNET, we see that model and AmeriFlux are in better alignment. There is still a striking difference in June and July of 2018 (newly added drought year) that suggests during drought there may be something plants are doing below ground to maintain higher rates of GPP that the DCHM is not capturing. We feel that the use of AmeriFlux FLUXNET in updated figures (including Figure S17) are now more useful in evaluating model performance.

[Figure]

**Figure S17. Daily evapotranspiration, ET, [mm d$^{-1}$], at US-KLS for (a) 2012 flash drought, (b) 2018 drought and (c) 2019 a non-drought year. Two standard deviations are shown for the DCHM-PV simulations. AmeriFlux ET is derived from latent heat measurements and shown as blue dots.**

---

## Author Response (AR2)

23 Jan 2024
**Editor decision: Publish subject to minor revisions (review by editor)**
by Gemma Coxon
Public justification (visible to the public if the article is accepted and published):
Dear authors,

Many thanks for your careful revisions to the paper. Your article was reviewed again by one of the original reviewers who recommended the paper was published as is. I also reviewed your paper and have some suggested minor revisions below that I kindly ask you to address. The paper will then be reviewed once again by myself, but will not be sent to external referees. I am looking forward to receiving the revised version of the manuscript.

Best wishes,
Gemma Coxon

Thank you very much for the careful review. Please find our responses to comments below in blue.

In addition to your comments and suggestions, we also combed through the manuscript to fix any remaining typos and grammatical errors, improve clarity and readability by rewriting sentences as needed, and improving figure/table readability and quality.

L16 – 18. 'Decreases in uncertainty….' This sentence doesn't make sense to me – can you rewrite and clarify?

We have revised this sentence to read:

Model estimates of GPP and ET during flash drought reduce to rates similar to what is observed during the winter indicating that plant function during drought periods is similar to those of dormant months.

L21. 'Frequency and severity of extreme droughts' should be 'The frequency and severity of'

Thank you for the suggestion. This change was made to the manuscript.

L25. I am not sure what you mean by development time. Development of what?

We have updated this sentence to clarify that we meant worsening soil and meteorological conditions. Additionally, we have added Christian et al., 2024 as a more recent reference that further supports this point.

**Work over the last decade has improved methods for identifying flash droughts based on rates of intensification of dry soils and concurrent elevated temperatures and atmospheric aridity (see Christian et al., 2024 and Lisonbee et al., 2021 for a summary of flash drought definitions and indicators).**

L60. 'Vegetation type and growth stage can plan an important…' should be 'Vegetation type and growth stage can play an important…'

Thank you for the suggestion. This change was made to the manuscript.

Figure 8. I find it quite difficult to interpret this graph with the different symbols for every month. Are there clear differences between months? If not, I would just use one shape, or maybe two shapes for Spring and Summer?

The authors appreciate this suggestion as differences were not easily distinguishable by month in this figure. We implemented the suggested two-symbol approach with "early" and "late" growing seasons indicated instead of using a different symbol for each month, and edited the legend accordingly.

[Figure]

**Figure 8. Monthly infiltration accumulation vs average days between precipitation events within a single month for (a) US-KFS, (b) US-KLS, and (c) US-KON. Each shape indicates whether the month occurs in the early growing season (circle: April - July) or late growing season (square: August - October). Colors distinguish flash drought (black) from drought (red) and non-drought (blue) years**

L450-470 and Figure 14. This section should be in the results as you are presenting figures and analysing results.

Thank you for this helpful comment. We moved this text and accompanying figure to the results Section 3.3.3, with the exception of the first sentence L450-451, after the results for

GPP from the modeling work. The figures and sections that followed were renumbered accordingly.

Discussion – the discussion section is currently 8 pages long. I would recommend a critical look at the content of the discussion and consider whether it is all needed as your key messages and discussion points get a little lost.

Thank you for this suggestion. For the initial revision, we worked hard to emphasize the main points about plant responses to flash drought and explain the processes that drive differences between models and observations. In doing so, we may have been redundant in trying to get our key points across to readers. We've since removed some of those redundancies (outlined below), which shortened the discussion by 3 pages and the manuscript by 2 pages.

Since most of Section 4.1.1 was moved to Section 3.3.3, we combined subsections 4.1.1 and 4.1.2 into one subsection 4.1, keeping the original title. Similar changes were made with Sections 4.2 and 4.6, and in combining parts of section 4.5 and 4.6.1. The new Discussion Sections are:
4.1 Mechanisms Controlling Plant Responses to Drought
4.2 Surface and Sub-surface Water Movement
4.3 Linking Carbon and Water Fluxes
4.4 Uncertainty in Vegetation Responses
4.5 Model Performance and Limitations
4.6 Implications for Land-surface Models

Overview of changes to the Discussion sections and justifications:

Section 4.1
- The first sentence and last paragraph remain. L451-470 were moved to make Section 3.3.3 as suggested.

Section 4.2 - We made this one section by combining 2 shorter subsections.
- L510-514 provided overly specific detailed results about infiltration and evaporation which can be generalized as: "Across all three study sites, infiltration exceeds evaporation in the growing season in drought and non-drought years (Figures 7, 10)."
- L521-525 moved to next section at L536
- Excluding L521-525, L 519-531 rewritten to be more concise.

Section 4.3
- Removed two sentences at L537-539 because they were redundant with L651-655, which is now moved to this section from section 4.6.2.

- Rearrange the first two sentences at L542 and remove the next sentence at L543 to help with flow.

Section 4.4 - We cut L562-570 because the details were deemed redundant and/or unnecessary towards describing how uncertainty from ensemble estimates differs during drought and non-drought periods. Specifically, the text was redundant with lines 290-292 and 325-330. This led to rearranging Section 4.4 into two paragraphs to help with flow and readability.
- L571-576 were moved to follow L555 "...uncertainty in phenology shrinks during dry periods."
- The first sentence of the paragraph beginning at L561 is now followed by L555-560.

Section 4.5 - We merged Section 4.5 with Section 4.6.1, using the title from Section 4.6 "Model Performance and Limitations". Much of the discussion in the original Section 4.5 Land Cover Influences was evaluating model performance by comparing outputs to observations, so there was a natural connection.
- We cut L578-585 (except the sentence on L579 beginning "Effects on infiltration…") because it was deemed redundant with results on lines 333-339. L579 was moved to the conclusion at L691.
- L585-586 is better placed in section 4.1 L498 with the discussion of GPP, stomatal conductance and high VPD.
- We cut the first sentence of Section 4.6.1 so that the new section begins with L604 "Our modeling approach permits…" Then L587-600 follows to make the first paragraph of this new section.

Several sentences were removed from Section 4.6.1 because they clouded the point being made comparing models outputs to observations. After removal, some rearranging had to be done to ensure readability (see new organization below). Sentences/lines removed:
- L607-612 - Details do not further contribute to the point.
- L614-615 - Redundant and further justified in L338-339.
- L619-621 - Too much detail.
- L628-630 - This sentence provided too many details of results from Hosseini et al., 2022 that do not enhance the argument comparing how models differ from observations.
- L638-641- Details of soil moisture comparisons do not further contribute to the argument comparing results and observations.

Section 4.6 comprises what remains after editing section 4.6.2.
The following lines were (re)moved
- L646-651 - The points covered here are a bit off topic for Implications for LSMs. Moreover, they continue to draw comparisons between modeling and observations, which has already been discussed at length in the previous section.
- L 651-655 was moved to Section 4.3

- L655-656, which was in direct response to one reviewer comment, has been moved to the previous section before the sentence beginning, "Differences in DCHM-PV and AmeriFlux Gpp cannot be fully attributed to…"
- L658-660 have into the subsequent paragraph about stomatal adaptations at L667.
- L667-679 has been broken into two sentences to enhance readability. And an additional citation (Guo, et al., 2022) has been added to further support our suggestion to improve model representations of adaptive stomatal regulations.
- L671-673, sentence beginning with "Future studies…" moved up to L665 and reworded to enhance readability.
* * *
New references from introduction and discussion.

Christian, J. I., Hobbins, M., Hoell, A., Otkin, J. A., Ford, T. W., Cravens, A. E., ... & Mishra, V. (2024). Flash drought: A state of the science review. *Wiley Interdisciplinary Reviews: Water*, e1714.

Guo, J. S., Bush, S. E., & Hultine, K. R. (2022). Temporal variation in stomatal sensitivity to vapour pressure deficit in western riparian forests. *Functional Ecology*, *36*(7), 1599-1611.